



# CAPRAM reduction towards an operational multiphase halogen and DMS chemistry treatment in the chemistry transport model COSMO-MUSCAT(5.04e)

Erik H. Hoffmann[1], Roland Schrödner[2], Andreas Tilgner[1], Ralf Wolke[2], Hartmut Herrmann[1]

[1] Atmospheric Chemistry Department (ACD), Leibniz Institute for Tropospheric Research (TROPOS), Permoserstr. 15, 04318 Leipzig, Germany
[2] Modeling of Atmospheric Processes Department (MAPD), Leibniz Institute for Tropospheric Research (TROPOS), Permoserstr. 15, 04318 Leipzig, 04318, Germany

*Correspondence to*: Hartmut Herrmann (herrmann@tropos.de)

**Abstract.** A condensed multiphase halogen and dimethyl sulfide (DMS) chemistry mechanism for application in chemical transport models is developed by reducing the CAPRAM DMS module 1.0 (CAPRAM-DM1.0) and the CAPRAM halogen module 3.0 (CAPRAM-HM3.0). The reduction is achieved by determining the main oxidation pathways from analysing the mass fluxes of complex multiphase chemistry simulations with the air parcel model SPACCIM. These simulations are designed to cover both pristine and polluted marine boundary layer conditions. Overall, the reduced DM1.0 contains 32 gas-phase reactions, 5 phase transfers, and 12 aqueous-phase reactions, of which two processes are described as equilibrium reactions. The reduced CAPRAM-HM3.0 contains 199 gas-phase reactions, 23 phase transfers, and 87 aqueous-phase reactions. For the aqueous-phase chemistry, 39 processes are described as chemical equilibrium reactions. A comparison of simulations using the complete DM1.0 and CAPRAM-HM3.0 mechanisms against the reduced ones indicates that the percentage deviations are below 5 % for important inorganic and organic air pollutants and key reactive species under pristine ocean and polluted conditions. The reduced mechanism has been implemented into the chemical transport model COSMO-MUSCAT and tested by performing 2D-simulations under prescribed meteorological conditions that investigate the effect of stable (stratiform cloud) and more unstable weather conditions (convective clouds) on marine multiphase chemistry. The simulated maximum concentrations of HCl are in the range of $10^9$ molecules cm$^{-3}$ and those of BrO are at around $1 \cdot 10^7$ molecules cm$^{-3}$ reproducing the range of ambient measurements. Afterwards, the oxidation pathway of DMS in a cloudy marine atmosphere has been investigated in detail. The simulations demonstrate that clouds have both a direct and an indirect photochemical effect on the multiphase processing of DMS and its oxidation products. The direct photochemical effect is related to in-cloud chemistry that leads to high DMSO oxidation rates and a subsequently enhanced formation of methane sulfonic acid compared to aerosol chemistry. The indirect photochemical effect is characterised by cloud shading, which occurs particularly in the case of stratiform clouds. The lower photolysis rate affects the activation of Br atoms and consequently lowers the formation of BrO radicals. The corresponding DMS oxidation flux is lowered by up to 30 % under thick optical clouds. Moreover, high updraft velocities lead to a strong vertical mixing of DMS into the free troposphere predominantly under cloudy conditions.



Furthermore, HOX photolysis is reduced as well, resulting in higher HOX-driven sulfite oxidation in aerosol particles below stratiform clouds. Altogether, the present model simulations have demonstrated the ability of the reduced mechanism to be applied in studying marine aerosol cloud processing effects in regional models such as COSMO-MUSCAT and can be applied

for more adequate interpretations of complex marine field measurement data, also by other regional models.

## 1 Introduction

In the marine and coastal atmosphere the chemical composition of the gas-phase, particles, and clouds as well as the size-distribution of particles are significantly influenced by emissions of sea spray aerosols (SSA) and volatile organic compounds from the sea surface (Simpson et al., 2015; Farmer et al., 2015; Quinn et al., 2015). Sea salt is an important compound of SSA

(Quinn et al., 2015) and represents a primary source for reactive chlorine and bromine compounds in the troposphere (Saiz-Lopez and von Glasow, 2012; Simpson et al., 2015). For reactive iodine compounds however, emissions of gaseous iodine compounds from the ocean surface dominate (Carpenter et al., 2012; Carpenter et al., 2013; Carpenter and Nightingale, 2015; Saiz-Lopez et al., 2012). Additionally, the ocean is the main source for dimethyl sulfide (DMS), which is the biggest natural atmospheric sulfur source (Andreae, 1990; Lana et al., 2011). The oxidation of DMS is the key to understanding the natural

radiative forcing as it affects both aerosol and cloud condensation nuclei (CCN) concentrations (Charlson et al., 1987). The chemical systems of halogens and DMS interact with each other strongly and are highly influenced by multiphase chemistry (Barnes et al., 2006; Hoffmann et al., 2016; von Glasow and Crutzen, 2004). As oceans cover around 70 % of the Earth's surface (Joshi et al., 2017; Law et al., 2013) and are in strong interaction with densely populated coastal areas (Kummu et al., 2016; von Glasow et al., 2013), this ocean-related atmospheric chemical subsystem is important for both Earth's climate and

air quality.

The chemistry of reactive halogen compounds as well as of DMS is very sensitive to anthropogenic pollution. The advection of $NO_x$ and ozone has strong effects on the activation of reactive halogen compounds (Hoffmann et al., 2019b; Shechner and Tas, 2017; Mahajan et al., 2009b; Mahajan et al., 2009a; McFiggans et al., 2002) and on DMS oxidation (Breider et al., 2010; Barnes et al., 2006; Chen et al., 2018). Moreover, reactive halogen compounds can significantly influence the depletion of

$NO_x$, ozone, $SO_2$, volatile organic compounds (VOCs), and oxidised volatile organic compounds (OVOCs) (von Glasow et al., 2002b, a; Sherwen et al., 2017; Schmidt et al., 2016; Sherwen et al., 2016). Furthermore, in anthropogenically influenced atmospheric environments, the $NO_3$ radical concentration is enhanced (Brown and Stutz, 2012). Thus, $NO_3$ radical-related DMS oxidation is reinforced (Breider et al., 2010; Chen et al., 2018), which influences aerosol sulfate formation and correspondingly leads to an increase of aerosol acidity (Muniz-Unamunzaga et al., 2018). The changed aerosol acidity further

affects the secondary organic aerosol (SOA) formation (Surratt et al., 2010; Surratt et al., 2007; Gaston et al., 2014) as well as the activation of reactive halogen compounds (Keene et al., 1998). Moreover, the ongoing reduction of fossil fuel combustion emissions in some parts of the world will promote the oxidation of DMS as an important contributor to sulfate aerosol



formation even in the Northern Hemisphere (Perraud et al., 2015). Therefore, it is important that chemical transport models (CTMs) treat the crucial multiphase chemistry pathways of both reactive halogen compounds and DMS.

Currently, only a couple of multiphase chemistry mechanisms of halogens and DMS have been developed and applied within CTMs, e.g. EMAC, CAM–MECCA, and GEOS-Chem (Chen et al., 2017; Chen et al., 2018; Jöckel et al., 2016; Long et al., 2014). However, the applied model core of these CTMs does not treat aqueous-phase chemistry of halogens and DMS by default. In CTMs that deal with the chemistry in the marine boundary layer (MBL) and the free troposphere, the activation of reactive halogen compounds and its dependence on aerosol acidity is often described by heterogeneous reactions. The

parameters of these reactions have been determined in laboratory studies for aerosol solutions that are more ideal than they naturally occur, e.g. pure sulfate or sodium chloride/bromide aerosol. Hence, the accuracy of the description of these processes is restricted and it cannot easily be assumed that they are representative under heterogeneous atmospheric conditions (Ammann et al., 2013). The treatment of multiphase chemistry in models allows for more detailed investigations concerning complex sea spray aerosol matrices. However, the level of detail for the implementation of aqueous-phase chemistry into CTMs is limited

because of numerical restrictions, since the implementation of aqueous-phase chemistry usually consumes huge amounts of CPU time. Consequently, mostly only specific small sub-systems are investigated, including a low number of halogen or DMS multiphase chemistry reactions (Chen et al., 2018; Chen et al., 2017). Both discussed aspects, consumption of CPU time and investigating only small sub-systems, highlight that an overall picture of multiphase marine chemistry cannot be drawn by chemical transport modelling yet and might lead to an over- or underestimation of important chemical pathways.

To achieve the goal of treating the multiphase chemistry of DMS and reactive halogen compounds within CTMs, not only the solvation of the high CPU consumption is necessary, but also the development of an adequate condensed multiphase chemistry mechanism dealing with the complexity of these chemical systems. An adequate mechanism does not currently exist and can only be derived by reducing detailed multiphase chemistry mechanisms, because important pathways could otherwise be missed.

In the present study, a reduced multiphase chemistry mechanism describing halogen and DMS chemistry is developed through a manual reduction using box model studies with the CAPRAM halogen module 3.0 (CAPRAM-HM3.0, Hoffmann et al., 2019a) and the CAPRAM DMS module 1.0 (CAPRAM-DM1.0, Hoffmann et al., 2016). Both modules currently contain the most detailed mechanisms dealing with the multiphase chemistry of these chemical systems. The reduced mechanism is implemented into the CTM MUSCAT (MUltiScale Chemistry Aerosol Transport; Wolke et al., 2004; Wolke et al., 2012).

Finally, the reduced mechanism is applied in idealised 2D-simulations with a focus on multiphase DMS oxidation in the MBL and the various effects of clouds essentially on halogens and DMS.



## 2 Reduction of the CAPRAM-DM1.0 and the CAPRAM-HM3.0

### 2.1 Model setup

The reduction of the marine multiphase chemistry modules CAPRAM-DM1.0 and CAPRAM-HM3.0 is achieved through
modelling studies with the air parcel model SPACCIM (SPectral Aerosol Cloud Chemistry Interaction Model, Sehili et al.,
2005; Wolke et al., 2005). SPACCIM is a model framework designed to solve complex multiphase chemistry systems and has
already been used for the development of reduced aqueous-phase chemistry mechanisms (Deguillaume et al., 2010). The
description of the simultaneously occurring chemical and physical processes in tropospheric cloud droplets and aqueous
aerosol particles in SPACCIM is realised by combining a complex size-resolved cloud microphysical model and a detailed
multiphase chemistry model. The standard atmospheric multiphase chemistry is represented by the near-explicit gas-phase
mechanism MCMv3.2 (Jenkin et al., 2003; Saunders et al., 2003) and the near-explicit aqueous-phase mechanism
CAPRAM4.0 (Bräuer et al., 2019).

The goal of reducing the CAPRAM-DM1.0 and CAPRAM-HM3.0 is that both modules can be applied in different marine
atmospheric environments in CTMs. To this end, simulations are carried out under two environmental conditions: (i) pristine
ocean and (ii) polluted coastal area. The simulations run for 48 hours and are equivalent to former simulations of atmospheric
marine environments studied with CAPRAM (Bräuer et al., 2013; Hoffmann et al., 2016; Hoffmann et al., 2019b).

In the simulations dealing the pristine ocean conditions, an air parcel is moved along a predefined trajectory at a 900 hPa
pressure level. The simulations are carried out at different latitudes (15°, 30°, 45°, 60° and 75°) and in different seasons of the
year (summer and winter). The air temperature in the simulations is adjusted accordingly. Furthermore, the simulations are
performed at different relative humidity levels (50 %, 70 %, and 90 %). In the simulations with relative humidity levels of
70 % and 90 %, cloud passages of the air parcel are considered. The cloud occurrence is modelled by uplifting the air parcel
to an 800 hPa pressure level either at noon and midnight or in the early morning. Due to adiabatic cooling, the relative humidity
increases, reaching the critical supersaturation so that a cloud is formed. The uplifting and down lifting of the air parcel requires
half an hour of modelling time in each case. The in-cloud residential time of the air parcel is two hours in the simulations with
cloud occurrences at noon and midnight and three hours in those with cloud occurrences in the early morning. The simulations
with 50 % relative humidity have no cloud passages of the air parcel included. For a detailed description of the emission and
initialisation of chemical species within the pristine ocean scenario, the reader is referred to Bräuer et al. (2013) and Hoffmann
et al. (2016).

The scenario at the polluted coastal area is divided into two sub-simulations at 45° latitude and 70 % relative humidity. The
lower diversity of the simulations compared to the pristine ocean scenario is chosen because previous model studies had
revealed that high $NO_x$ concentrations suppress gas-phase halogen radical cross reactions and lead to a domination of halogen
nitrate and nitryl chloride photolysis in halogen atom activation. The effect of photolysis and temperature change does not
affect these four important halogen activation precursors ($ClNO_2$, $ClNO_3$, $BrNO_3$, and $INO_3$). In the first simulation, the air
parcel represents the composition of a pristine marine environment, which is advected over a polluted coastal urban area. The





second simulation describes an air-sea breeze circulation system. Details on the model configurations of the first simulation
are explicitly given in Hoffmann et al. (2019b) and those of the second simulation in Hoffmann et al. (2019a).

The reduction of both modules is performed by analysing the modelled ten-minute time-resolved source and sink fluxes of key
chemical compounds of marine multiphase chemistry. These cover all DMS oxidation intermediates and, for halogen
chemistry, all $X_y$ (X, $X_2O_2$, $XNO_2$, $XNO_3$, XO, XOO, OXO, XY, HOX, and HX, with X/Y = Cl, Br, or I) species. At first, the

CAPRAM-DM1.0 is reduced and afterwards the CAPRAM-HM3.0. In the following Sect 2.2 and Sect 2.3, the development
of the reduced CAPRAM-DM1.0 and CAPRAM-HM3.0 is described.

The final reduced marine multiphase chemistry mechanism is evaluated through control simulations, which are carried out by
using not only the full but also the reduced marine multiphase chemistry mechanism. The goal of the reduction is that the
concentration of chemical species important for chemical transport modelling, e.g. ozone, sulfate or nitrate, only deviate by

less than 5 % on average over the full modelling time. A subordinate goal is that oxidants and important chemical compounds
of marine multiphase chemistry only diverge by less than 10 % on average. The important chemical compounds of DMS
multiphase oxidation are DMS, dimethyl sulfoxide (DMSO), and methane sulfonic acid (MSA). In the case of halogen
multiphase chemistry, they are the Cl, Br, and I atoms as well as ClO, BrO, and IO and stable halogen compounds, which act
as important reservoir or activation species for halogen radicals, i.e. hypohalogeneous acids, nitryl chloride, and dihalogen

molecules. Lastly, the evolution of the concentration time profile for all species has to match ($R^2 \geq 0.75$).

## 2.2 Development of the reduced multiphase DMS chemistry module

The oxidation of DMS in the tropospheric multiphase system leads to gaseous $SO_2$, sulfuric acid, $DMSO_2$, MSA, dissolved
sulfate, or methane sulfonate through a sequence of steps (Hoffmann et al., 2016; Barnes et al., 2006). To cover the important
intermediate oxidation steps, the reduction consists of six individual ones:

i)  Consideration of main multiphase DMS oxidation pathways only;

ii)  Lumping of simple reactions;

iii)  Application of the pseudo-steady-state approximation;

iv)  Neglect of production/oxidation of dimethyl sulfone ($DMSO_2$) in the aqueous phase;

v)  Lumping of the aqueous-phase oxidation of methane sulfinic acid (MSIA);

vi)  Reduction of oxidation/production pathways of specific chemical compounds unimportant in the gas or aqueous
phase.

In the following, these mechanism reduction steps are outlined in more detail.

## 2.2.1 Main pathways of multiphase DMS oxidation

In the first reduction step, the main pathways of the multiphase oxidation of DMS and its oxidation products are investigated

by analysing the time-resolved source and sink fluxes of all simulations. Main pathways are defined here as chemical
production or loss processes that contribute more than 5 % to the overall average mass flux of the investigated compound. This





analytical approach has proven its applicability in manual mechanism reduction (Deguillaume et al., 2010; Ervens et al., 2003). The present analyses provide the important DMS multiphase reaction pathways within the troposphere, similar to a former CAPRAM study dealing with multiphase DMS chemistry (Hoffmann et al., 2016). This approach determines the subsequent
reduction steps.

### 2.2.2 Lumping of simple reactions

According to the current knowledge, in the gas phase, the oxidation of DMS or its oxidation products through H-abstraction leads to a corresponding peroxyl radical, which can be further oxidised into an alkoxy radical.  Recently, it was suggested that the methylthiomethylperoxyl radical ($CH_3SCH_2O_2$) undergoes a rapid unimolecular H-shift (Wu et al., 2015; Berndt et al.,
2019). The final stable product will be an oxidised organic sulfur compound that is characterised by an aldehyde and an organic hydrogen peroxide functionality. This compound can be oxidised then in both gas and aqueous phase. Currently, the chemistry of this compound is not investigated and therefore not treated in the CAPRAM-DM1.0. However, when more laboratory data are available, further mechanisms have to consider this chemistry to improve the modelling of DMS chemistry and its effects. Classically, the alkoxy radical can then undergo thermal decomposition, a reaction with oxygen, or isomerization (Lightfoot
et al., 1992). However, the current gas-phase chemistry of the CAPRAM-DM1.0 is based on the MCMv3.2, which only treats thermal decomposition for the $CH_3SCH_2O$, i.e. C-C bond scission leading to the release of formaldehyde (HCHO). The analyses of the mass fluxes reveal that decomposition appears immediately after formation. Therefore, further decomposition products are directly incorporated in every reaction in which an alkoxy radical is formed and the corresponding decomposition reaction is deleted (see Eq. R1-3).

$$CH_3SCH_2O_2 + NO \rightarrow CH_3SCH_2O + NO_2 \tag{R1}$$
$$CH_3SCH_2O \rightarrow CH_3S + HCHO \tag{R2}$$
$$\Sigma: CH_3SCH_2O_2 + NO \rightarrow CH_3S + HCHO + NO_2 \tag{R3}$$

Further simple integrated reactions correspond with aqueous-phase reactions, i.e. reactions with water. The reduction of the CAPRAM3.0 has already revealed that peroxyl radical formations in the aqueous phase are not reaction-rate determining steps
(Deguillaume et al., 2010). The same is true if reactions of oxidation intermediates occur with water only. Therefore, such aqueous-phase reactions are deleted and the products are directly implemented on the right-hand side of reaction equations.

### 2.2.3 Application of the pseudo-steady-state-approximation

The oxidation of DMS by the OH radical and the Cl atom occurs not only through H-abstraction but also through the addition of these radicals onto sulfur. The formed DMS adduct is unstable and decomposes back into DMS and the corresponding
radical if it is not stabilised through a reaction with oxygen, which adds to the sulfur (Barnes et al., 2006). It is recommended that the DMS adduct can also react with $NO_x$ or decompose into methane sulfenic acid ($CH_3SOH$) and a methyl ($CH_3$) radical (Barnes et al., 2006; Yin et al., 1990). However, the first reduction step has already revealed that even at polluted coastlines,





with NO$_x$ concentrations above 10 ppb, NO$_x$ related decompositions are not of atmospheric importance. The analysis shows that oxygen is too reactive against DMS adducts.

The pseudo-steady-state-approximation (PSSA) is a fundamental way to deal with such reactive intermediates to derive the overall rate of a chemical reaction (Seinfeld and Pandis, 2006). The same method is applicable on the oxidation of DMSO, whose oxidation by the Cl atom also leads to a DMSO adduct.

$$\frac{d[DMS]}{dt} = -k_1[DMS][X] + k_2[DMS\text{-}X]; (X = OH \text{ or } Cl) \tag{1}$$

$$\frac{d[DMS\text{-}X]}{dt} = k_1[DMS][X] - k_2[DMS\text{-}X] - k_3[DMS\text{-}X][O_2] \tag{2}$$

$\hookdownarrow \frac{d[DMS]}{dt} = -\frac{k_1 k_3 [O_2]}{k_2 + k_3 [O_2]} [DMS][X]$ \hfill (3)

As outlined, apart from the DMSO formation, a considerable amount of CH$_3$SOH is also formed through DMS adduct decomposition. Hence, a PSSA is also effective for this reaction sequence. The implemented rate constant of the DMS adduct decomposition within the CAPRAM-DM1.0 is given temperature independent. The same is true for the oxygen addition reaction. Therefore, under tropospheric conditions both reactions can be aggregated into a first order reaction rate constants

and merged. This merging gives a ratio between decomposition and oxygen addition that is implemented into the overall reaction (see Eq. R4-7).

$$CH_3SCH_3 + OH \rightleftarrows CH_3S(\text{-}OH)CH_3 \tag{R4}$$

$$CH_3S(\text{-}OH)CH_3 + O_2 \rightarrow CH_3S(O)CH_3 + HO_2 \tag{R5}$$

$$CH_3S(\text{-}OH)CH_3 \overset{O_2}{\rightarrow} CH_3SOH + CH_3O_2 \tag{R6}$$

$\Sigma$: $CH_3SCH_3 + OH \overset{O_2}{\rightarrow} 0.9\ CH_3S(O)CH_3 + 0.9\ HO_2 + 0.1\ CH_3SOH + 0.1\ CH_3O_2$ \hfill (R7)

The PSSA is also valid for the methylthiyl radical (CH$_3$S) reaction with oxygen, which leads to the methylthio peroxyl radical (CH$_3$SOO). The CH$_3$SOO has multiple reaction pathways. It can react with NO$_x$, decompose back into CH$_3$S, or SO$_2$ and CH$_3$, or rearrange itself into the sulfonyl radical (CH$_3$SO$_2$). The first reduction step reveals that the reaction with NO$_x$ is negligible. Only a decomposition into SO$_2$ as well as the formation of the sulfonyl radical occur. Therefore, the PSSA is also applied to

these two reaction pathways as well.

$$\frac{d[CH_3S]}{dt} = -k_1[CH_3S][O_2] + k_2[CH_3SOO] \tag{4}$$

$$\frac{d[CH_3SOO]}{dt} = k_1[CH_3S][O_2] - k_2[CH_3SOO] - k_3[CH_3SOO] \tag{5}$$

$\hookdownarrow \frac{d[CH_3S]}{dt} = -\frac{k_1 k_3}{k_2 + k_3} [CH_3S][O_2]$ \hfill (6)

### 2.2.4 Neglect of production/oxidation of DMSO$_2$

An analysis of the sink and source fluxes has revealed that aqueous-phase chemistry contributes a little more than 5 % to DMSO$_2$ formation and oxidation. However, the modelled overall DMSO$_2$ formation and oxidation flux is negligible compared to that of MSIA. Furthermore, DMSO$_2$ has low reactivity towards OH oxidation in the gas ($k_{OH} < 3.0\times10^{-13}$ cm$^3$ molecules$^{-1}$





$s^{-1}$; Falbe-Hansen et al., 2000) and aqueous phase ($k_{OH} = 1.77 \times 10^7$ l mol$^{-1}$ s$^{-1}$; Zhu et al., 2003) . Because of the low measured background gas-phase concentrations of DMSO$_2$ in the single-digit ppt range (Davis et al., 1998; Berresheim et al., 1998), the

gas-phase oxidation of DMSO$_2$ by the OH radical is likely to be suppressed through methane oxidation ($k = 3.5 \times 10^{-15}$ to $6.4 \times 10^{-15}$ cm$^3$ molecules$^{-1}$ s$^{-1}$ in the temperature range of 270 to 298 K). Therefore, dry and wet deposition can be assumed as the major atmospheric removal processes for DMSO$_2$. Consequently, in order to shrink the mechanism DMSO$_2$ production in the aqueous phase is neglected and also the oxidation of DMSO$_2$ in the gas- and aqueous phase.

**2.2.5 Lumping of the aqueous-phase oxidation of MSIA**

In the fifth reduction step, the oxidation of MSIA in the aqueous phase has been simplified. MSIA has a pK$_a$ value of 2.28 (Wudl et al., 1967). Therefore, it occurs in both its non-dissociated and its dissociated form under atmospheric aerosol as well as cloud conditions. The only important oxidant for the non-dissociated form is ozone. The deprotonated MSIA reacts with both the OH and the Cl$_2^-$ radical via an electron transfer reaction into aqueous CH$_3$SO$_2$. The formed CH$_3$SO$_2$ reacts with O$_2$ into the methylsulfonylperoxyl radical (CH$_3$SO$_2$O$_2$) or decomposes into the CH$_3$ radical and dissolved SO$_2$ that is immediately

dissociated into HSO$_3^-$/SO$_3^{2-}$. Because of its high atmospheric abundance, in our study the O$_2$ concentration is modelled to be almost constant within the tropospheric aqueous phase. Furthermore, both reaction rate constants are implemented as temperature-independent. Therefore, a ratio between these reactions can be calculated. The reaction of the CH$_3$SO$_2$O$_2$ with MSIA yields MSA and the methylsulfonylalkoxyl radical (CH$_3$SO$_3$). The later decomposes into the CH$_3$ radical and sulfate. Both reactions occur immediately. Consequently, all reactions of the deprotonated MSIA oxidation can be summarized into

one reaction for each oxidant, covering the overall MSIA loss (see below Eq. R8-13 for MSIA oxidation by the OH radical).

$$CH_3SO_2^- + OH \xrightarrow{O_{2,aq}, H_2O} 0.9\ CH_3SO_{2,aq} + 0.9\ OH^- + 0.1\ CH_3O_{2,aq} + 0.1\ HSO_3^- - 0.1\ H_2O \tag{R8}$$

$$CH_3SO_{2,aq} \xrightarrow{O_{2,aq}} SO_{2,aq} + CH_3O_{2,aq} \tag{R9}$$

$$CH_3SO_{2,aq} + O_{2,aq} \rightarrow CH_3SO_2O_{2,aq} \tag{R10}$$

$$CH_3SO_2^- + CH_3SO_2O_{2,aq} \rightarrow CH_3SO_3^- + CH_3SO_{3,aq} \tag{R11}$$

$$CH_3SO_{3,aq} \xrightarrow{O_{2,aq}} SO_{3,aq} + CH_3O_{2,aq} \tag{R12}$$

$$\Sigma: CH_3SO_2^- + OH \xrightarrow{O_{2,aq}, H_2O}$$
$$CH_3O_{2,aq} + 0.235\ HSO_3^- + 0.765\ CH_3SO_3^- + 0.765\ SO_{3,aq} + 0.9\ OH^- - 0.765\ CH_3SO_2^- - 0.235\ H_2O \tag{R13}$$

**2.2.6 Reduction of oxidation/production pathways of specific chemical compounds less important in gas or aqueous phase**

In the last reduction step, the mechanism is again analysed for residual multiphase chemistry pathways to be combined. These are, for example, reactions of radicals that now treat only one fast reaction sequence and thus are merged into the previous reaction.





## 2.3 Development of the reduced multiphase halogen chemistry module

The goal of reducing the CAPRAM-HM3.0 is to enable the description of key halogen chemistry affecting ozone, $NO_x$, $SO_2$,
VOCs, and OVOCs by a reduced mechanism that almost conserves the concentration time profile of the reactive halogen
compounds listed earlier. The following three reduction steps are applied to achieve this:

    i)     Consideration of chemical production or loss processes that contribute more than 5 % to the overall mass flux of
            halogen compounds only;

    ii)    Lumping of simple reaction sequences;

iii)   Neglect of oxidation/production of specific chemical species modelled to be unimportant in tropospheric gas or
            aqueous-phase chemistry.

In the following, these mechanism reduction steps are outlined in more detail.

### 2.3.1 Main pathways of multiphase halogen chemistry

An analysis of the main pathways is performed in a similar manner as for the multiphase DMS chemistry. However, the
development of a DMS-like stepwise oxidation scheme is impossible, due to important side pathways and interconnections
with other chemical subsystems, e.g. $NO_x$ or $HO_x$ chemistry and various halogen cross interactions (Saiz-Lopez and von
Glasow, 2012). Furthermore, halogen multiphase chemistry is characterised by large differences in aqueous-phase oxidation
within aerosol particles and cloud droplets (Bräuer et al., 2013; Hoffmann et al., 2019b; von Glasow et al., 2002a). Therefore,
an appropriate reduced representation of multiphase halogen chemistry requires the focus of the reduction to be on different
halogen species. Hence, the determination of the main pathways is done for a huge number of halogen compounds covering
key halogen atoms, halogen radicals, halogen nitrates, halogenated organics such as halogenated aldehydes, as well as halogen
oxo-carboxylic acids.

As already modelled in other studies, the analyses revealed that the Cl atom is an important oxidant for VOCs and OVOCs,
e.g. alkanes, non-oxidised aromatic compounds, alcohols, and aldehydes. However, because of limited computational costs,
state-of-the-art chemical mechanisms of CTMs do not contain a high variability of organic compounds as the near-explicit
MCM. In order to still represent the chemistry of important VOCs and OVOCs in CTMs, species of the same compound classes
or of equal reactivity are typically merged into 'lumped' species in condensed mechanisms applied in CTMs (Baklanov et al.,
2014). Because of these limitations, the reduced CAPRAM-HM3.0 has to be linkable with the chemical mechanisms used in
CTMs. Therefore, a first screening on treated VOCs and OVOCs in the mechanisms MOZART4.0 (Schultz et al., 2018),
RACM2 (Goliff et al., 2013), MECCA (Jöckel et al., 2016), GEOS-Chem (Wang et al., 2019),and SAPRC11 (Yan et al., 2019)
has been performed for the main VOCs and OVOCs. As a result, only the Cl atom oxidation of the lumped VOCs and OVOCs
that are treated within all of these mechanisms is considered further. However, this approach results in lower HCl but higher
ClO formation when the reduced version of the CAPRAM-HM3.0 is compared to the non-condensed MCMv3.2/CAPRAM4.0/
CAPRAM-DM1.0/CAPRAM-HM3.0.





### 2.3.2 Lumping of simple reaction sequences

In the gas-phase, halogen atoms react rapidly with $O_2$- and CO-yielding unstable molecules that, as the model simulations show, immediately decompose again. Still, within the CAPRAM-HM3.0, specific oxidation pathways lead to unstable molecules (e.g. the oxidation of halogenated oxidised organics). Consequently, in every reaction in which such an unstable molecule occurs as a product it is replaced by the halogen atom and $O_2$ or CO.

The further processing of halogenated organic peroxyl radicals in the gas phase result in halogenated organic alkoxy radicals. As for DMS, the halogenated organic alkoxy radical decomposition, which is modelled not to be the overall rate-determining step is integrated into these reactions. Overall, the recombination of the halogenated organic peroxyl radicals with other organic peroxyl radicals ($RO_2$) leads exclusively to halogenated carbonyls (see Eq. R14-17).

$$XCH_2O_2 + RO_2 \rightarrow 0.2\ XCHO + 0.2\ XCH_2OH + 0.6\ XCH_2O;\ (X=Cl\ or\ Br) \tag{R14}$$

$$XCH_2OH + OH \xrightarrow{O_2} XCHO + HO_2 + H_2O \tag{R15}$$

$$XCH_2O \xrightarrow{O_2} XCHO + HO_2 \tag{R16}$$

$$\Sigma:\ XCH_2O_2 + RO_2 \xrightarrow{O_2,\ OH} XCHO + 0.8\ HO_2 \tag{R17}$$

If the analysis of the main pathways leads to only one further reaction of a compound being left, this reaction has been screened for two criteria: (i) Does the follow-up reaction occur rapidly and (ii) is the overall concentration of the product so low that it would not be a significant interfering factor for the modelling. If both issues are true, the overall reaction is merged together. For example, the recombination of IO in the aqueous phase leads to iodite ($HIO_2$) which is an intermediate in the conversion between iodide and iodate ($IO_3^-$). It is quickly oxidized into iodate by $H_2O_2$, which is ubiquitous in the marine atmospheric multiphase (Jacob and Klockow, 1992; Benedict et al., 2012; Kim et al., 2007; Yuan and Shiller, 2000), and also has a very low modelled concentration. Overall, the IO recombination together with the oxidation of $HIO_2$ by $H_2O_2$ results in iodate (see Eq. R18-20).

$$IO_{aq} + IO_{aq} + H_2O \rightarrow HOI_{aq} + HIO_{2,\ aq} \tag{R18}$$

$$HIO_{2,\ aq} + H_2O_2 \rightarrow H^+ + IO_3^- + H_2O \tag{R19}$$

$$\Sigma:\ IO_{aq} + IO_{aq} \xrightarrow{H_2O_{2,\ aq}} HOI_{aq} + H^+ + IO_3^- \tag{R20}$$

### 2.3.3 Neglect of oxidation/production of specific chemical species modelled to be less important in tropospheric gas or aqueous-phase chemistry

For the reduction of the halogen chemistry part, less important chemical halogen species are defined as such with low ($< 0.1$ppt for non-radical species) modelled concentrations or high chemical stability ($k_{OH} < 6.4 \cdot 10^{-15}$ cm$^3$ molecules$^{-1}$ s$^{-1}$ that is the $k_{298}$ of methane; Atkinson et al., 2006) that their non-consideration does not affect the concentrations of the target species under conditions in the lower troposphere. Typical species with a rather high chemical stability are chlorinated and brominated organics (e.g. $CH_3Cl$), except bromoform ($CHBr_3$), for which oxidation in the lower troposphere is negligible. Typical species



with low modelled concentrations are oxidised halogenated organics derived from the OH oxidation of methylated halogens (e.g. ICHO or ICIO from CH₃I oxidation), precursors of which are mainly removed through photolysis. As the reduced mechanism is developed to deal with tropospheric multiphase chemistry, the oxidation of such species is not treated within the reduced CAPRAM-HM3.0.

**2.4 Evaluation of the reduction steps**

The evaluation of the performed reduction steps is achieved by comparing simulations with the reduced and with the original CAPRAM-DM1.0 and CAPRAM-HM3.0 added to the multiphase chemistry mechanism MCMv3.2/CAPRAM4.0. The evaluation simulations are carried out for 45° latitude with a relative humidity of 70 % under pristine ocean (Hoffmann et al., 2016) and polluted coastal conditions (Hoffmann et al., 2019a). The scenarios are termed as follows: 'Pristine', 'Breeze' and

'Outflow'. The term 'Pristine' is the abbreviation of the pristine ocean scenario, the term 'Breeze' stands for the scenario of an air-sea breeze circulation system, and the term 'Outflow' represents the scenario of the advection of polluted air masses over a marine environment. The simulations run for 96 hours and include cloud passages between 11 am and 1 pm and between 11 pm and 1 am in the scenarios 'Pristine' and 'Outflow' and between 1 pm and 2 pm in the scenario 'Breeze'. The longer simulation time compared to that of previous simulations has been chosen in order to investigate the effect of a longer

modelling time on concentration divergence.

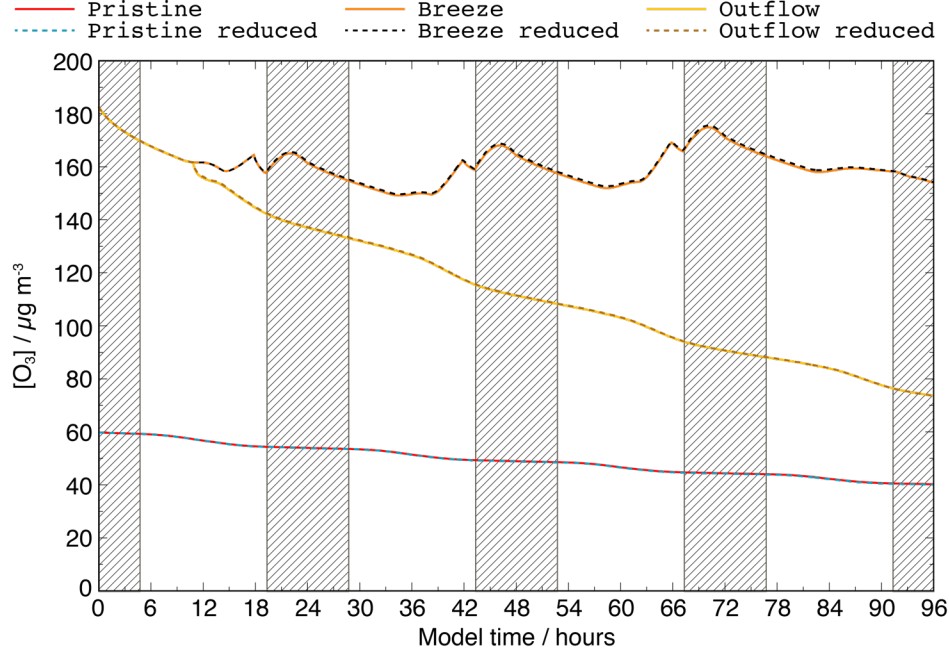

**Figure 1 Modelled concentrations of ozone within the scenarios 'Pristine', 'Breeze', and 'Outflow' compared between the simulations with the full (solid lines) and reduced (dotted lines) CAPRAM-DM1.0 and CAPRAM-HM3 mechanism.**


The investigation of the modelled evolution of the concentration time profile of ozone is shown in Figure 1, revealing an
excellent agreement for all the scenario ($R^2 = 1$). Moreover, the average ozone concentrations diverge by less than 5 %
throughout the entire modelling time. This demonstrates that the reduced mechanism is able to reproduce the modelled ozone
concentrations of the complex mechanism.

The same analysis is performed for other air pollutants and key aerosol compounds important for air quality modelling. These
are NO, $NO_2$, $SO_2$, $HNO_3$, HCl, and DMS in the gas-phase, the dry mass as well as the organic mass of the aerosols together
with nitrate, sulfate, chloride, and methane sulfonate. Furthermore, the analysis is performed for reactive halogen compounds
and the OH, $NO_3$, and $HO_2$ radicals. The average percentage deviation for these chemical species can be found in Table 1.

The main target species, except MSA in the 'Breeze' scenario, do not exceed the 5 % threshold. This is also true for the OH,
$HO_2$, and $NO_3$ radical in the gas and aqueous phases. Even for reactive halogen compounds, the deviation rarely exceeds the
5 % mark.

However, especially in the 'Outflow' scenario, reactive bromine compounds exceed the 10 % threshold. This is caused by
missing brominated organics that trap the bromine from further reaction. For example, 3 ppt of bromine is trapped on average
in brominated alcohols formed through Br atom-related oxidation of alkenes and further $RO_2$ recombination. Regarding the
low concentration and the fact that alcohols are further oxidized into carbonyls, only the formation of brominated carbonyls is
considered in the reduced CAPRAM-HM3.0 to minimize the mechanism. Consequently, the bromine radical is recycled faster
by the following reaction sequence:

$$Br_g + C_2H_{4,g} \rightarrow \rightarrow \rightarrow BrCHO_g + 1.8\,HO_2 + HCHO - CH_3O_2 \tag{R21}$$

$$BrCHO_g \rightleftharpoons BrCHO_{aq} \tag{R22}$$

$$BrCHO_{aq} \rightarrow H^+ + Br^- + CO_{aq} \tag{R23}$$

$$Br_g + O_{3,g} \rightarrow BrO_g + O_{2,g} \tag{R24}$$

$$BrO_g + HO_{2,g} \rightarrow HOBr_g + O_{2,g} \tag{R25}$$

$$HOBr_g \rightleftharpoons HOBr_{aq} \tag{R26}$$

$$HOBr_{aq} + Br^- + H^+ \rightleftharpoons Br_{2,aq} + H_2O \tag{R27}$$

$$Br_{2,aq} \rightleftharpoons Br_{2,g} \tag{R28}$$

$$Br_{2,g} + h\nu \rightarrow Br_g + Br_g \tag{R29}$$

Overall, this increases the modelled concentrations of reactive bromine compounds, particularly in the afternoon after cloud
occurrence and under high alkene as well as low ozone conditions compared to an urban environment. However, the evolution
of the concentration time profile fits very well ($R^2 = 0.98, 0.95, 0.97$, and $0.75$ for Br, BrO, HOBr, and $Br_2$, respectively).
Apart from bromine, larger differences also occur for HOCl in the 'Breeze' and 'Outflow' scenarios. This is related to the
restricted VOC and OVOC oxidations within the reduced CAPRAM-HM3.0 in order to match the condensed gas-phase
chemistry mechanisms implemented in the CTMs. Therefore, the HCl concentration is more than 2 % smaller and more Cl
atoms react with ozone into ClO that reacts quickly with $HO_2$ to yield HOCl. However, as opposed to $Br_2$ formation by HOBr,





the higher HOCl does not lead to a too high modelled $Cl_2$ formation because of the significant higher chloride content in sea spray aerosols compared to bromine.

Finally, the methane sulfonate anion ($MS^-$) is around one quarter lower in the 'Breeze' scenario, which is caused by the $Cl_2^-$ radical oxidation. This reduction actually revealed that the $Cl_2^-$ radical is an important oxidant for oxalic acid and $MS^-$. Consequently, other OVOC oxidations are discarded for treatment in the reduced CAPRAM-HM3.0. The modelled $Cl_2^-$ radical concentrations are around one order of magnitude higher in the 'Breeze' scenario compared to the 'Pristine' and 'Outflow' scenarios, resulting in much higher $MS^-$ oxidation rates. However, the evolution of the $MS^-$ concentration time profile matches very well ($R^2 = 1$) and the relative difference at the end of the modelling is only a deviation of 6 % from the overall concentration.

Overall, the evaluation reveals that the reduced mechanism system is able to reproduce similar results as the full mechanism system. Therefore, it is appropriate for implementation into CTMs. The reduced mechanisms of the CAPRAM-DM1.0 and CAPRAM-HM3.0 will be called CAPRAM-DM1.0red and CAPRAM-HM3.0red in the following text.

The modelling studies also reveal that computational (CPU) time is conserved, especially within the scenario 'Pristine' (see Figure 2). Compared to the base runs, the CPU time is reduced by 16 %, 5 %, and 6 % in the scenarios 'Pristine', 'Breeze', and 'Urban outflow', respectively. Overall, the CPU time reduction is low, but is accounted to the usage of the MCMv3.2 and CAPRAM4.0 that still treats more than 21000 reactions. Furthermore, the calculation of microphysical processes consumes a huge amount of CPU time, too. Consequently, the high CPU time required overlay the CPU time consumption from the reduction. The higher time consumption in the scenario 'Pristine' compared to the scenarios 'Breeze' and 'Urban outflow' is caused by the very low initialised $NO_x$ concentrations. Within the scenarios 'Breeze' and 'Urban outflow', the high initialised $NO_x$ concentrations effectively suppress halogen radical cross-interactions in the gas phase. These are very fast equilibria and thus increase the CPU time. Therefore, under high $NO_x$ conditions, much lower CPU time is required to solve halogen multiphase chemistry processes. However, in the 'Pristine' scenario, the rapid occurring gas-phase cross-interactions of halogen radicals still exist hindering stronger amplified CPU time reductions.

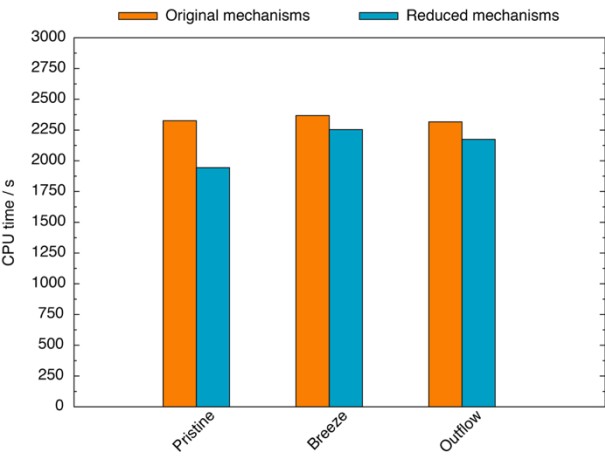


**Figure 2 Required CPU time within the scenarios 'Pristine', 'Breeze', and 'Outflow' considering the original multiphase chemistry mechanism system MCMv3.2/CAPRAM4.0/CAPRAM-DM1.0/CAPRAM-HM3.0 and the multiphase chemistry mechanism system with the CAPRAM-DM1.0red and CAPRAM-HM3.0red. The CPU costs include gas and aqueous-phase chemistry, microphysics, model initialization, and output.**

## 3 First applications in chemistry transport modelling with COSMO-MUSCAT


The state-of the art CTM currently applied at TROPOS is MUSCAT (Wolke et al., 2004; Wolke et al., 2012). It is either coupled to the weather model COSMO (Consortium for Small Scale Modelling; Steppeler et al., 2003; Baldauf et al., 2011) or ICON (ICOsahedral Non-hydrostatic; Zängl et al., 2015), providing all required meteorological data fields (e.g. wind, temperature, relative humidity, liquid water content, and precipitation) to MUSCAT that are necessary to calculate the advection, diffusion, and physico-chemical interaction of particles and trace gases. The emission files of gases and aerosols within MUSCAT are generated by pre-processors and the chemical mechanism is imported from ASCII files, which allows for changes without code recompilation. In terms of dust (Heinold et al., 2007) and sea spray aerosols (Barthel et al., 2019), emissions can also be calculated online.

### 3.1 Implementation of the CAPRAM-DM1.0red and the CAPRAM-HM3.0red into COSMO-MUSCAT


Within the present study, the model framework COSMO-MUSCAT is used, which has recently been extended to be able to treat multiphase chemistry in clouds (Schrödner et al., 2014; Schrödner et al., 2018). The chemistry of DMS and detailed halogen chemistry are still missing. Therefore, the mechanisms CAPRAM-DM1.0red and CAPRAM-HM3.0red are implemented into the atmospheric multiphase chemistry core of COSMO-MUSCAT, in which gas-phase chemistry is described by the MOZART4 mechanism (Schultz et al., 2018) and aqueous-phase chemistry is described by the CAPRAM3.0red mechanism (Deguillaume et al., 2010). MOZART4 treats comprehensive halogen and DMS gas-phase chemistry that is replaced, as well as specific lumped VOCs and OVOCs. As outlined, these lumped species particularly cover the VOCs and OVOCs, where the oxidation by the Cl atom is significant. To link the reduced CAPRAM-HM3.0 to the





MOZART4, the overall rate constants of the Cl atom are derived for the following lumped VOCs: (i) BIGALK, (ii) ALKOH, (iii) $C_2H_5CHO$, (iv) BIGALD1, (v) XYL, and (vi) BZALD. The lumped species $C_2H_5CHO$ represents all aldehydes with more

than three carbon atoms and is newly implemented into MOZART4, which makes it consistent with CAPRAM3.0red. Accordingly, the oxidation pathways are adjusted and the $C_2H_5CHO$ oxidation by the OH radical and the Br atom has also been implemented.

$$C_2H_5CHO + OH/Cl/Br \rightarrow 1.5 \ CH_3C(O)O_2 + H_2O/HCl/HBr \tag{R30}$$

Table 2 provides the implemented reaction rate constants of lumped VOCs. A full mechanistic description of the adjusted

CAPRAM-DM1.0red and CAPRAM-HM3.0red is given in the supplement (Table S2-S10).

The recombination of halogenated peroxyl radicals as described above is adjusted to fit into MOZART4. In the MCM, this recombination is implemented for the sum of all $RO_2$, whereas in MOZART4 it is often only for $CH_3O_2$ and $CH_3C(O)O_2$. Therefore, the ratio of the MCM is applied in the $RO_2$ recombination reaction, but $RO_2$ is considered as $CH_3O_2$. The reaction rate constant is adopted from the corresponding HM2 reaction (Bräuer et al., 2013).

$$XCH_2O_2 + CH_3O_2 \rightarrow XCHO + 1.4 \ HO_2 + 0.8 \ HCHO + 0.2 \ CH_3OH \tag{R31}$$

Currently, the calculation of aqueous-phase chemical processes in MUSCAT is limited to cloudy conditions, i.e. a liquid water content (LWC) of above 0.01 g m$^{-3}$. Furthermore, the chemical equilibria are treated dynamically as forward and backward reactions, which represents a critical challenge for the numerical solver. Deviations from the equilibrium state of rapid phase transfers and dissociations may lead to large chemical fluxes and hence small timesteps, i.e. high computational costs. The

robustness of the numerical integration is particularly affected at phase boundaries between cloud and non-cloud grid cells. A robust numerical integration can be achieved by pre-balancing the aqueous-phase equilibria of $CO_2$, $NH_3$, $HNO_3$, $HCl$, $SO_2$, $H_2SO_4$, and organic acids in cloud-free grid cells at a predefined threshold LWC under cloud conditions (0.01 g m$^{-3}$, abbreviated with 'sub#1'). This approach is similar to the pH calculation parameterisation described by Alexander et al. (2012) but is designed to describe cloud chemistry and its effect on cloud droplet acidity in detail. In order to describe the activation

of reactive halogen compounds by the chemistry in deliquesced aerosols, an additional sub-mechanism is introduced, assuming an LWC of 10 µg m$^{-3}$ in cloud-free grid cells (abbreviated with 'sub#2'). A schematic on how both sub-mechanisms enable the multiphase chemistry treatment in COSMO-MUSCAT can be seen in Figure 3.


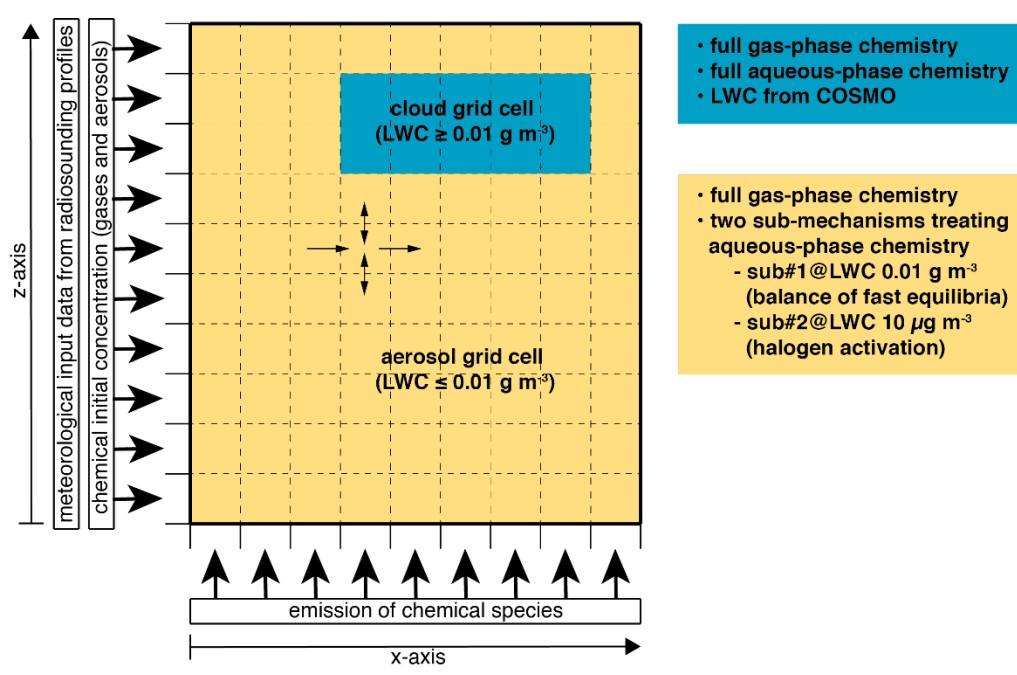

**Figure 3 Schematic representation of the multiphase chemistry treatment in COSMO-MUSCAT in an idealised 2D-simulation.**

The main halogen chemistry reactions that are treated in 'sub#2' are:

$$XY_g \rightleftharpoons XY_{aq} \text{ (with X = Cl, Br or I, and Y = Cl, Br or I)} \tag{R32}$$

$$HOX_g \rightleftharpoons HOX_{aq} \tag{R33}$$

$$XNO_{3,g} \rightleftharpoons XNO_{3,aq} \tag{R34}$$

$$HOX_{aq} + Y^- + H^+ \rightleftharpoons XY_{aq} + H_2O \tag{R35}$$

$$XY_{aq} + Z^- \rightleftharpoons XZ_{aq} + Y^- \text{ (with Z = Cl, Br or I)} \tag{R36}$$

$$HX_{aq} \rightleftharpoons H^+ + X^- \tag{R37}$$

$$HNO_{3,g} \rightleftharpoons HNO_{3,aq} \tag{R38}$$

$$N_2O_{5,g} \rightleftharpoons N_2O_{5,aq} \tag{R39}$$

$$N_2O_{5,aq} \rightarrow NO_2^+ + NO_3^- \tag{R40}$$

$$NO_2^+ + H_2O \rightarrow NO_3^- + 2\,H^+ \tag{R41}$$

$$NO_2^+ + Cl^- \rightleftharpoons ClNO_{2,aq} \tag{R42}$$

$$XNO_{3,aq} + H_2O \rightarrow HOX_{aq} + HNO_{3,aq} \tag{R43}$$

$$HSO_3^- + HOX_{aq} \rightarrow HSO_4^- + X^- + H^+ \tag{R44}$$

$$H_2O_{2,g} \rightleftharpoons H_2O_{2,aq} \tag{R45}$$

$$HSO_3^- + H_2O_2 + H^+ \rightarrow SO_4^{2-} + 2\,H^+ + H_2O \tag{R46}$$



The activation of halogens is in accordance with the heterogeneous reactions used in other CTMs (Badia et al., 2019; Hossaini et al., 2016; Jöckel et al., 2016; Long et al., 2014; Muniz-Unamunzaga et al., 2018; Saiz-Lopez et al., 2014; Wang et al., 2019) to describe the activation of reactive halogen compounds. However, the first of the two defined chemical solution sub-systems is able to treat both pH-dependent processes, including (i) the activation of reactive halogen compounds and (ii) sulfite

formation induced by HOX and $H_2O_2$ in the MBL online. A complete overview of the treated aqueous-phase reactions and phase transfers in both the 'sub#1' and the 'sub#2' mechanism in the CAPRAM-DM1.0red and CAPRAM-HM3.0red is provided in the supplement in Tables S8-S10. Overall, COSMO-MUSCAT now represents the CTM with the most detailed description of marine multiphase chemistry (see Figure 4).

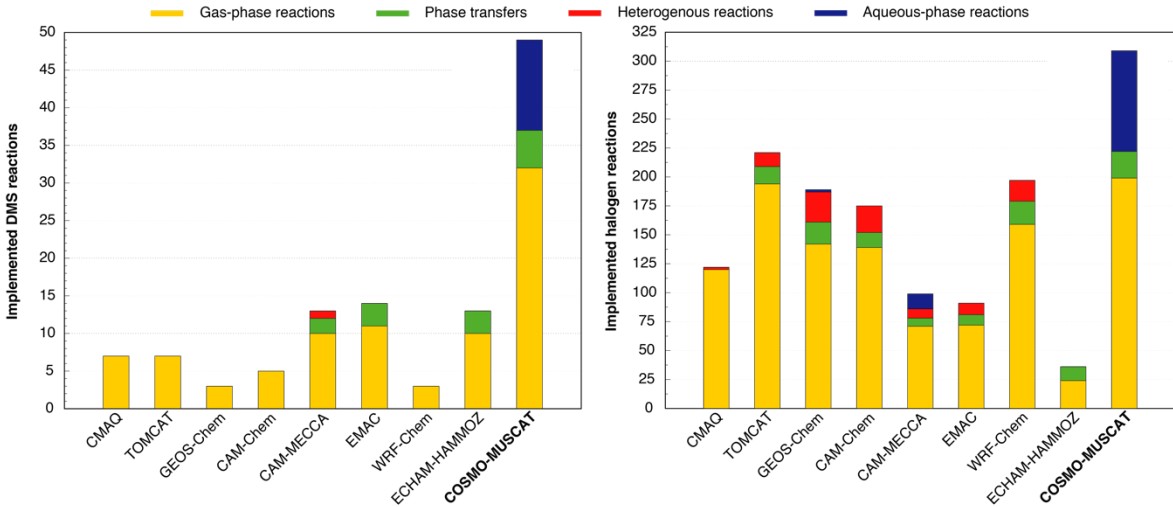

**Figure 4 Comparison of applied tropospheric DMS and halogen chemistry mechanisms within the chemical transport models: CMAQ (Muniz-Unamunzaga et al., 2018), TOMCAT (Hossaini et al., 2016), Geos-Chem (Wang et al., 2019), CAM-Chem (Saiz-Lopez et al., 2014), CAM-MECCA (Long et al., 2014), EMAC (Jöckel et al., 2016), WRF-Chem (Badia et al., 2019), and ECHAM-HAMMOZ (Schultz et al., 2018).**

## 3.2 Verification of the 2D-implementation

The verification of the implementation is carried out by two 2D-simulations (x-z-cross section) dealing with a 'pristine ocean' scenario under two meteorological conditions, a convective and stable atmospheric layer. These conditions will result in modelled convective and stratiform clouds, respectively. Moreover, this scenario is applied in order to investigate DMS oxidation in more detail. The verification of the implementation is performed by investigating the activation of halogen compounds in the MBL and comparing it with available ambient measurements and other model data. 2D-simulations are

preferred over 3D ones because they are based on the same meteorological dynamics but have one lower degree of freedom. As a result, the system under investigation is less computationally expensive. Additionally, 2D-simulations enable a comprehensive understanding of multiphase chemistry in the atmospheric column, including vertical mixing processes.





### 3.2.1 2D model setup

The chemical model setup, i.e. the initialisation, deposition, and emission of trace gases and VOCs as well as the aerosol
composition to describe the atmospheric composition of the pristine ocean, is the same as the one used for reduction. Because
of the lumped species within MOZART4, specific VOC emissions are merged together. The emission rates are provided in the
supplement (Table S1). Initialised concentrations of gas-phase and aerosol compounds represent ground values and are
distributed vertically as constant mass mixing ratios within the model domain of MUSCAT. These values are used as constant
boundary conditions on the left-hand side of the model domain (see Figure 3).

Two different meteorological cases are simulated: One with a high diversity of clouds and a strong vertical wind velocity
(further called 'unstable meteorological condition') and another with a stable cloud cover at the top of the marine boundary
layer and a weak vertical wind velocity (further called 'stable meteorological condition'). These two different meteorological
'pristine ocean' simulations are chosen to evaluate the numerical robustness of the model. The meteorological scenarios are
initialised using radiosonde profiles ('unstable meteorological condition' station: Camborne Observations, station identifier:
3808 on 12Z 21.06.2016 and 'stable meteorological condition' station: GVAC Sal Observations, station identifier: 8594 on
12Z 12.06.2017), which can be read and processed by COSMO by default and are considered constant meteorological
boundary conditions on the left-hand side of the model domain (see Figure 3).

The model domain spans 400 horizontal grid cells with a resolution of 1.11 km per grid cell and 100 vertical levels with a
resolution of 100 m. Whereas COSMO is run on the full domain, only the inner 200 horizontal columns and lowermost 15
vertical levels are used for the multiphase chemistry simulations with MUSCAT. The smaller distribution for MUSCAT is
used because the interaction is sufficient to describe the multiphase chemistry in the marine boundary layer (MBL).
Furthermore, the height of the MBL is often lower than 1000 m (Norris, 1998; Carrillo et al., 2016), which enables an
investigation of the interactions between the MBL and the free troposphere and significantly saves computation time compared
to the full domain. Therefore, this resolution can capture almost all essential chemical processes as well as the distribution of
important species in the marine troposphere. Overall, the modelling domain for multiphase chemistry encompasses 3000 grid
cells.

### 3.2.2 Comparison with measurements

*HCl gas-phase concentration*

Firstly, the modelled concentration range of gaseous HCl are compared to actual measurements. In Figure 5, it can be seen
that very high HCl concentrations are modelled, especially in the 'stable meteorological condition' simulation above the MBL.
These are a result of the constant vertical distributed mass mixing ratios of the initial values. Since the evaluation focuses on
the activation of HCl within the MBL, these values can be neglected. The modelled values below 1000 m outside of clouds
are in the order of $10^9$ molecules cm$^{-3}$ after 12 hours of modelling time and thus in the range of both other modelled (Hossaini
et al., 2016; Wang et al., 2019) and measured values within the marine pristine boundary layer (e.g., Keene and Savoie, 1999;



Sander et al., 2013; Pszenny et al., 2004). The higher LWC of clouds results in strongly reduced HCl gas-phase concentrations due to the phase partitioning shift from the gas to the aqueous phase. Further chemical cloud processing increases aerosol acidity, yielding higher HCl gas-phase concentrations behind the cloud in wind direction (see Figure 5).

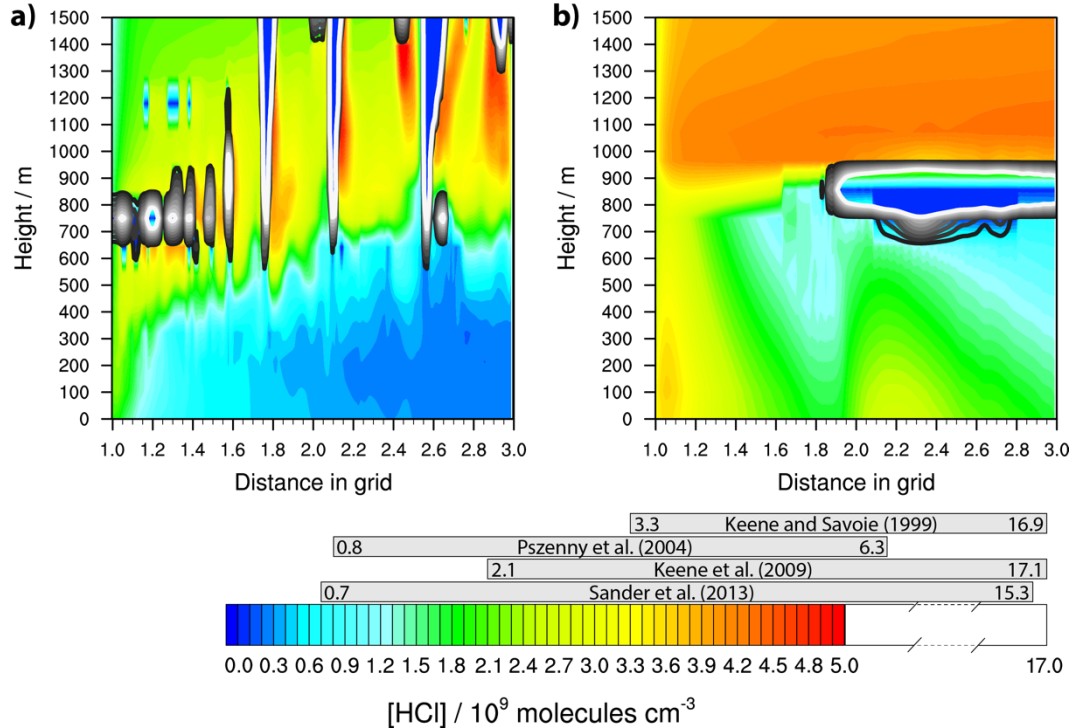

**Figure 5 Simulated concentrations of HCl in the gas phase by COSMO-MUSCAT (a) in the 'unstable meteorological condition'**
**simulation with convective clouds and (b) in the 'stable meteorological condition' simulation with stratiform clouds after 12 hours of modelling time. The grey bars represent measured values in most likely pristine marine environments (see Table S11 for further details). The black contour lines represent the simulated clouds. The denser the crowding, the higher the simulated liquid water content of the clouds (with a max of 0.9 g m$^{-3}$ and 0.3 g m$^{-3}$ in the 'unstable meteorological condition' and 'stable meteorological condition' simulations, respectively).**

*BrO gas-phase concentration*

Contrary to HCl, the measured concentrations of HBr over the pristine ocean are missing. However, a high number of measured gas-phase BrO concentrations in the MBL are available (Saiz-Lopez and von Glasow, 2012; Simpson et al., 2015). Therefore, as a second step, the modelled gas-phase BrO concentration range is compared to measurements. In Figure 6, it can be seen that the modelled values outside of the cloud grid cells are in the range between $10^6$ to $10^7$ molecules cm$^{-3}$ after 12 hours of

modelling time. Thus, they are in the range of other modelled (Zhu et al., 2019) as well as measured values within the pristine MBL (e.g., Leser et al., 2003; Read et al., 2008; Chen et al., 2016).





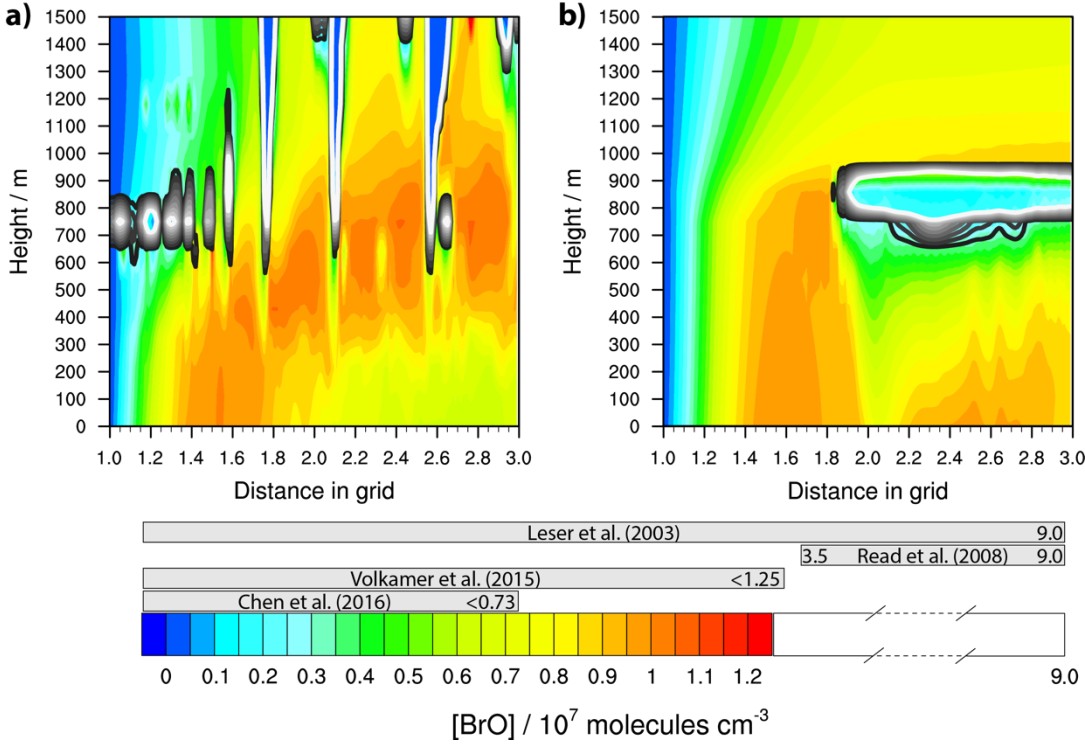

**Figure 6 Simulated concentrations of bromine monoxide in the gas phase by COSMO-MUSCAT (a) in the 'unstable meteorological condition' simulation with convective clouds and (b) in the 'stable meteorological condition' simulation with stratiform clouds after 12 hours of modelling time. The grey bars represent measured values in most likely pristine marine environments (see Table S11 for further details). The black contour lines represent the simulated clouds. The denser the crowding, the higher the simulated liquid water content of the clouds.**

Apart from that, many field studies suffered the problem that the BrO concentration in the MBL was always below the detection limit (Sander et al., 2003). The measured BrO in the MBL in a recent measurement study was also always below the detection limit of 0.5 pptv, i.e. around $1.2 \cdot 10^7$ molecules cm$^{-3}$ (Volkamer et al., 2015). Hence, the modelled BrO concentrations should be adequate. Since the activation of reactive bromine is highly related to that of chlorine, the mechanism is able to represent the activation of reactive halogen compounds within the MBL. A comparison of reactive iodine compounds with measurements is not performed, because the concentration range is highly sensitive to the initialised emission values of molecular iodine and iodinated organics from the sea surface and can thus be stated as uncertain.

Overall, the new marine multiphase chemistry model can represent marine aerosol chemistry and linked halogen activation under consideration of meteorological dynamics and shows a good agreement to other field as well as model data. Thus, it is applicable for further detailed 3D studies.



### 3.3 Results of pristine ocean scenarios

### 3.3.1 Vertical wind and DMS distribution

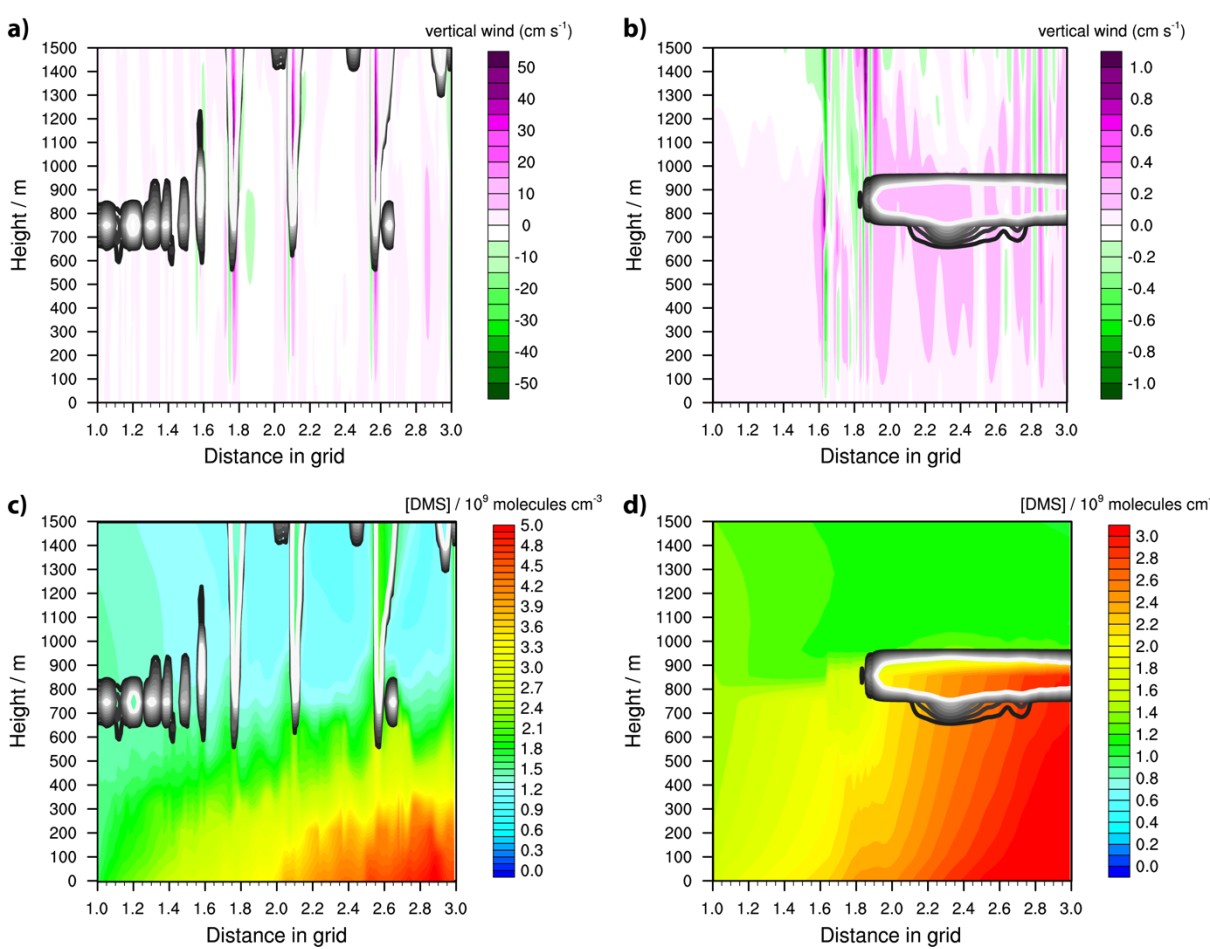

**Figure 7 Modelled vertical winds (cm s⁻¹) (a) in the 'unstable meteorological condition' simulation with convective clouds and (b) in the 'stable meteorological condition' simulation with stratiform clouds after 12 hours of modelling time. The black contour lines represent the simulated clouds. The denser the crowding, the higher the simulated liquid water content of the clouds. Further, the modelled concentrations of DMS in the gas phase (10⁹ molecules cm⁻³) are shown (c) in the 'unstable meteorological condition' simulation with convective clouds and (d) in the 'stable meteorological condition' simulation with stratiform clouds after 12 hours of modelling time.**

Both scenarios are further applied to investigate the multiphase oxidation pathways of DMS in a cloudy marine atmosphere in detail. In Figure 7, the modelled distribution of clouds and the strength of the vertical wind field as well as the modelled DMS concentration distribution for both simulations is shown after 12 hours of modelling time. In the 'unstable meteorological condition' simulation, the clouds extend up to a height of more than 2000 m, whereas in the 'stable meteorological condition' simulation, the top of the cloud is capped below an inversion layer at around 1000 m. Furthermore, the vertical winds are much stronger in the 'unstable meteorological condition' simulation. Because of the strong vertical winds, gas-phase DMS





concentrations of around $2 \cdot 10^9$ molecules $cm^{-3}$ are transported into the lower free troposphere. The strong inversion and low magnitude of the vertical wind speed in the 'stable meteorological condition' simulation hinders effective DMS transportation

555  into the free troposphere. Therefore, above the MBL, the DMS concentration is at around $1 \cdot 10^9$ molecules $cm^{-3}$, which is consistent with the initialised background DMS concentration. Below the inversion, DMS concentrations are more homogeneously distributed, which is different to the 'unstable meteorological condition' simulation, with a stronger variability related to the vertical wind field, i.e. the peaking of DMS concentrations into higher vertical levels because of strong updrafts (cp. Figure 7).

560  **3.3.2 Vertical DMSO distribution**

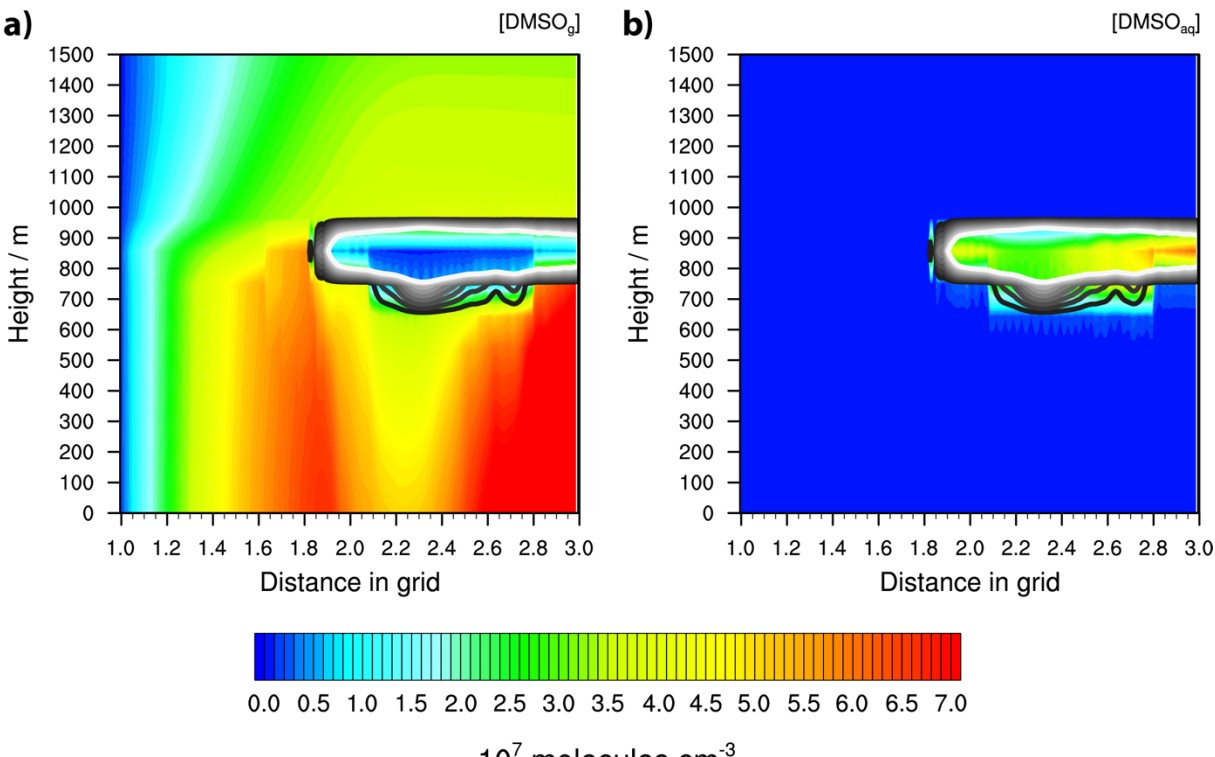

**Figure 8 Simulated concentrations of DMSO (a) in the gas phase and (b) in the aqueous phase under the 'stable meteorological condition' simulation with stratiform clouds after 12 hours of modelling time. The black contour lines represent the simulated clouds. The denser the crowding, the higher the simulated liquid water content of the clouds.**

565  If such homogeneously distributed DMS concentrations are modelled as in the 'stable meteorological condition' simulation in a clear sky atmosphere, a similar concentration distribution for the first stable DMS oxidation products will be modelled. Hence, the concentration distribution of DMSO is a good indicator to investigate the effect of clouds on DMS oxidation. In Figure 8, the distribution of DMSO in the gas and aqueous phases in the 'stable meteorological condition' simulation after 12 hours of modelling time is shown.





The stratiform clouds have a very high influence on the DMSO concentration in both the gas and the aqueous phase. The spatial gas-phase DMSO concentration distribution differs from the DMS concentration. Below the optically thickest clouds, the gas-phase DMSO concentration is significantly reduced, whereas above the cloud it slightly increases. In the cloud grid cells, the gas-phase DMSO concentration is reduced significantly because of the uptake into the aqueous phase. The reduced gas-phase concentrations below the cloud cannot be explained by vertical or horizontal transportation because, as can be seen in Figure 7, an updraft would result in observable concentration peaks in the higher vertical levels. Therefore, the gas-phase DMSO formation in the MBL is somehow influenced by the cloud above and it is necessary to investigate the cloud-induced effect on crucial DMS oxidants in the pristine MBL.

### 3.3.3 Effects of stratiform clouds on DMS oxidation

Within the pristine MBL, the BrO radical is a primary DMS oxidant that forms DMSO (Barnes et al., 2006; Breider et al., 2010; Hoffmann et al., 2016; von Glasow and Crutzen, 2004; Chen et al., 2018). This radical is formed through reaction of $O_3$ by the Br atom that is activated by multiphase chemistry. The following are important pathways of activation in a clear sky (von Glasow and Crutzen, 2004; von Glasow et al., 2002b):

$$BrCl_g + h\nu \rightarrow Br_g + Cl_g \tag{R47}$$

$$Br_g + O_{3,g} \rightarrow BrO_g + O_{2,g} \tag{R48}$$

$$BrO_g + DMS_g \rightarrow DMSO_g + Br \tag{R49}$$

$$BrO_g + HO_{2,g} \rightarrow HOBr_g + O_{2,g} \tag{R50}$$

$$HOBr_g + h\nu \rightarrow Br_g + OH_g \tag{R51}$$

$$HOBr_g \rightleftharpoons HOBr_{aq} \tag{R52}$$

$$HOBr_{aq} + Br^- + H^+ \rightleftharpoons BrCl_{aq} + H_2O \tag{R53}$$

$$BrCl_{aq} \rightleftharpoons BrCl_g \tag{R54}$$

In the pristine marine boundary layer, two competing pathways determine the main fate of the BrO radical through (i) a reaction with DMS and (ii) a reaction with HO2. The oxidation of DMS leads to DMSO and the Br atom, so that a cycle is established that continuously depletes $O_3$ and forms DMSO as long as DMS is emitted or ozone is available. This cycle is disturbed by the reaction of BrO with HO2, yielding HOBr, which can be photolyzed back into the Br atom again or converted by multiphase chemistry into BrCl or Br2. Overall, the photolysis of HOBr and particularly BrCl determine the DMS to DMSO conversion. Clouds suppress the photolysis of BrCl and HOBr due to the reflection of incoming solar radiation. The thicker the cloud, the lower the radiation flux below. Consequently, Br atom activation below the cloud is hindered, affecting the BrO concentration and thus the reaction rate of BrO with DMS that yields DMSO (see Figure 9). Due to a longer lifetime and corresponding horizontal advection, the DMSO concentration profile is shifted to the right compared to the BrO one.

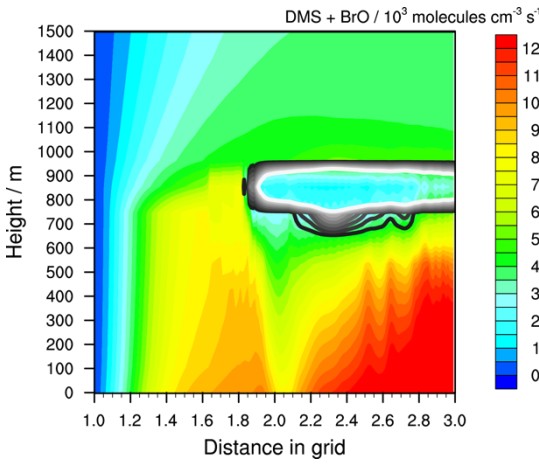


**Figure 9 Simulated oxidation rate of DMS by BrO in the 'stable meteorological condition' simulation with stratiform clouds after 12 hours of modelling time. The black contour lines represent the simulated clouds. The denser the crowding, the higher the simulated liquid water content of the clouds.**

The photolysis of BrCl compounds is highly sensitive to cloud shading and thus has a high impact on the formation of reactive

bromine and the linked DMS oxidation. Furthermore, model studies suggest that BrCl photolysis is an important contributor to Cl atom activation in the MBL (Wang et al., 2019; von Glasow et al., 2002b). Hence, the outlined model results reveal that the shading effect of clouds is also very important for the atmospheric Cl atom concentration budget, affecting the atmospheric oxidation capacity within the MBL.

### 3.3.4 The formation of MSA and aqueous sulfate

DMSO is rapidly oxidised into MSIA and thus a similar profile is modelled. However, because of its high reactivity in the gas and aqueous phases as well as its high solubility, MSIA is rapidly oxidised into methane sulfonate ($MS^-$) in both the aerosol and the cloud phases. There, $O_3$ is the preferred oxidant in the aerosol phase, whereas OH is in cloud droplets. In the next time step, the $MS^-$ formed in-cloud can be transported towards the ground by downdrafts. However, as can be seen from comparing the DMS concentrations in Figure 7 with the DMSO concentrations in Figure 8, the up- and downdrafts in the 'stable

meteorological condition' simulations have little effect on the concentration distribution in height. The strongest effect relates to the advection from the left-hand to the right-hand side of the model domain and continuous emission from the surface. In the grid cells before cloud occurrence, the DMSO concentration is high and consequently the $MS^-$ formation is as well. Due to the advection of the stable $MS^-$ to the right-hand side of the model domain, the spatial DMSO profile is not modelled. The high modelled chemical fluxes in cloud droplets indicate the highest $MS^-$ concentrations to be within and below the cloud grid

cells (see Figure 10).





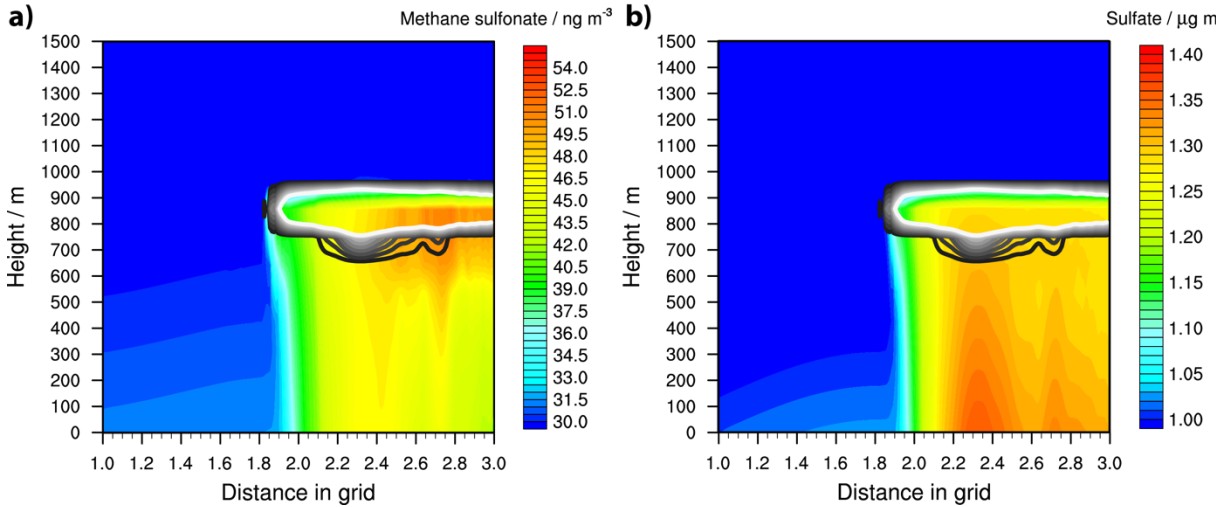

**Figure 10 Simulated aqueous-phase concentrations of (a) methane sulfonate and (b) sulfate in the 'stable meteorological condition' simulation with stratiform clouds after 12 hours of modelling time. The initial background concentration of methane sulfonate is at about 30 ng m$^{-3}$ and that of sulfate at 1 µg m$^{-3}$.**

As for MSA, the formation of sulfate is enhanced in the grid cells directly or indirectly affected by clouds. At the beginning cloud edge, the lower photolysis rates increase the $SO_2$ oxidation into sulfate by HOX and $H_2O_2$. Hence, a stronger HOX-related (especially HOI) reactive $SO_2$ uptake on the aerosols is modelled. The uptake is highest under the optically thickest modelled clouds, resulting in the highest modelled sulfate concentrations being there. Therefore, the modelled spatial concentration is contrary to that of DMSO.

## 4 Conclusion and Outlook

A reduced multiphase chemistry mechanism of DMS and reactive halogen compounds is developed through the reduction of the near-explicit multiphase chemistry mechanisms CAPRAM-DM1.0 and CAPRAM-HM3.0. Simulations that compare the reduced with the original mechanisms revealed that the reduced mechanisms are able to reproduce the concentrations and time evolutions of main air pollutants as well as key reactive halogen compounds. Additionally, CPU time in the box model

simulations is reduced by 16 %, 5 %, and 6 %, depending on the model scenario. Afterwards, the reduced mechanisms are implemented into the chemistry transport model COSMO-MUSCAT. The implementation is evaluated by idealised 2D-simulations of an atmospheric pristine ocean environment. The verification has proven that the reduced marine multiphase chemistry mechanism can represent marine aerosol chemistry and linked halogen activation as it matches measured field concentrations, e.g. HCl and BrO.

Afterwards, 2D-simulations of a pristine ocean scenario are carried out, investigating the effect of stable (stratiform cloud) and more unstable weather conditions (convective clouds) on multiphase DMS oxidation. The simulations reveal that clouds have both strong direct and indirect photochemical effects on the oxidation and vertical distribution of DMS in the marine

atmosphere. Firstly, locally high updraft velocities in the unstable scenario result in a fast transportation of DMS from the
marine boundary layer into the free troposphere. Hence, the transportation and further oxidation of DMS can be an important
source of $SO_2$ within the free troposphere, particularly in the Southern Ocean region that is less affected by anthropogenic
pollution. Secondly, clouds enhance the formation of MSA via the DMS addition channel. The formed DMSO is effectively
consumed by cloud droplets where it is rapidly oxidised into MSA. Thirdly, the shading of clouds has a high impact on the
photolysis of dihalogens that are the main contributor to Cl and Br atom activation. Hence, a much lower oxidation of DMS
into DMSO occurs below stratiform clouds. In contrast, the lower HOX photolysis induces stronger sulfate formations. The
results indicate that clouds strongly affect the oxidation of DMS directly because of enhanced aqueous-phase oxidation into
MSA and indirectly by suppressing the DMSO formation due to lower halogen atom activation. In total, a strong possible
effect on the atmospheric oxidation capacity within the MBL of the pristine marine boundary layer is assumed.

Overall, the 2D-simulations demonstrate the capability of COSMO-MUSCAT to now cover the multiphase chemistry in
marine influenced atmospheric environments. This allows for deeper investigations of multiphase chemistry in a wide range
of temporal and spatial resolutions together with transportation and microphysical processes in the future. In a next step, the
mechanism will be applied in simulations with COSMO-MUSCAT to model measurement campaigns at the Cape Verde
Atmospheric Observatory (Carpenter et al., 2010), supporting the interpretation of the measurement data and enabling further
model/mechanism validations. Furthermore, the reduced mechanism is designed in such a way that new findings in DMS or
halogen chemistry can easily be implemented, e.g. improved understanding of the multiphase chemistry of the unimolecular
H-shift of $CH_3SCH_2O_2$.

**Code and Data availability**

The code for the COSMO model is available according to the Software License Agreement by Deutscher Wetterdienst (German
Weather Service, http://cosmo-model.org). The source code of MUSCAT and SPACCIM, external parameters, and applied
mechanisms are archived on a local Git server and can be obtained by request through Ralf Wolke (wolke@tropos.de). Access
to the model code used in the paper has been granted to the editor.

**Author contribution**

EHH, AT, and RW did the model development on SPACCIM. EHH, AT, and HH designed the SPACCIM modelling work.
EHH performed the SPACCIM simulations. EHH, AT, and HH analysed the SPACCIM model results. EHH, RS, and RW did
the model development on COSMO-MUSCAT. EHH, RS, and RW designed the COSMO-MUSCAT modelling work. EHH
performed the COSMO-MUSCAT simulations. EHH analysed the COSMO-MUSCAT model results. EHH, RS, AT, and HH
wrote the paper.



**Competing interests**

The authors declare that they have no conflict of interest.

*Special issue statement.* This article is part of the special issue "Simulation chambers as tools in atmospheric research (AMT/ACP/GMD inter-journal SI)". It is not associated with a conference.

*Acknowledgements*

E.H.H. thanks the Ph.D. scholarship program of the German Federal Environmental Foundation (Deutsche Bundesstiftung Umwelt, DBU, AZ: 2016/424) for its financial support. This work has received funding from the European Union's Horizon 680 2020 research and innovation program through the EUROCHAMP-2020 Infrastructure Activity under grant agreement no. 730997. This work was also supported by the EU Marie Skłodowska-Curie Actions, (690958-MARSU-RISE-2015).

*Financial support*

This research has been supported by the European Commission (grant no. EUROCHAMP-2020 (730997)).

This work was also supported by the EU Marie Skłodowska-Curie Actions, (690958-MARSU-RISE-2015).

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




Table 1  Average percentage deviations [%] of some inorganic and organic target compounds between the simulations with the full and reduced CAPRAM-DM1.0 and CAPRAM-HM3 mechanisms (deviations calculated throughout the full SPACCIM simulation). Exceedances of the threshold are marked in red.

| Species | 'Pristine' | | 'Breeze' | | 'Outflow' | |
|---|---|---|---|---|---|---|
| **Gas phase** | Average | R² | Average | R² | Average | R² |
| *5% treshold* | | | | | | |
| O₃ | -0.2% | 1.00 | 0.3% | 1.00 | 0.1% | 1.00 |
| NO | -3.9% | 1.00 | 0.4% | 1.00 | -0.8% | 1.00 |
| NO₂ | -2.1% | 1.00 | 0.7% | 1.00 | -0.1% | 1.00 |
| SO₂ | 1.5% | 1.00 | 1.3% | 1.00 | 0.6% | 1.00 |
| HNO₃ | 1.2% | 1.00 | -0.5% | 1.00 | -1.0% | 1.00 |
| DMS | -4.2% | 1.00 | -1.6% | 1.00 | -3.2% | 1.00 |
| HCl | 0.6% | 1.00 | -2.1% | 1.00 | -0.6% | 1.00 |
| *10% treshold* | | | | | | |
| DMSO | 6.1% | 1.00 | -1.5% | 1.00 | 10.6% | 0.99 |
| OH | 0.0% | 1.00 | 0.0% | 1.00 | 0.1% | 1.00 |
| HO₂ | -0.5% | 1.00 | 0.1% | 1.00 | -0.1% | 1.00 |
| NO₃ | -0.5% | 1.00 | 0.6% | 1.00 | 0.0% | 1.00 |
| H₂O₂ | 0.0% | 1.00 | -3.3% | 1.00 | -2.4% | 1.00 |
| Cl | 6.4% | 1.00 | 5.1% | 1.00 | 0.8% | 1.00 |
| Br | 9.8% | 1.00 | 4.2% | 1.00 | 29.0% | 0.98 |
| I | -0.6% | 1.00 | -6.1% | 0.99 | -1.2% | 0.99 |
| ClO | 7.4% | 1.00 | 7.9% | 1.00 | 2.0% | 1.00 |
| BrO | 9.7% | 0.99 | 4.2% | 1.00 | 24.5% | 0.95 |
| IO | -1.8% | 1.00 | -3.1% | 0.99 | -0.9% | 1.00 |
| HOCl | 5.0% | 1.00 | 37.5% | 0.96 | 17.8% | 1.00 |
| HOBr | 8.9% | 1.00 | 10.2% | 1.00 | 27.7% | 0.97 |
| HOI | -2.4% | 1.00 | 3.5% | 0.98 | -3.4% | 0.98 |
| Cl₂ | 0.4% | 1.00 | 5.6% | 0.97 | -2.4% | 0.97 |
| Br₂ | 36.0% | 0.96 | -1.1% | 0.93 | 24.0% | 0.75 |
| ClNO₂ | -0.9% | 1.00 | 0.0% | 1.00 | 0.3% | 1.00 |
| **Aqueous phase** | Average | R² | Average | R² | Average | R² |
| *5% treshold* | | | | | | |
| OrgMass | 1.1% | 1.00 | -0.2% | 1.00 | 0.0% | 1.00 |
| DryMass | 0.2% | 1.00 | -0.3% | 1.00 | 0.1% | 1.00 |
| H⁺ | 0.2% | 1.00 | -1.2% | 1.00 | -0.2% | 1.00 |
| Sulfate | 1.1% | 1.00 | -0.6% | 1.00 | -0.1% | 1.00 |
| Nitrate | 0.0% | 1.00 | 0.8% | 1.00 | 1.7% | 1.00 |
| Cl⁻ | -0.4% | 1.00 | 0.0% | 1.00 | 0.7% | 1.00 |
| Methane sulfonate | 2.5% | 1.00 | -18.7% | 1.00 | 2.1% | 1.00 |
| *10% treshold* | | | | | | |
| OH_aq | -1.9% | 1.00 | 2.0% | 0.99 | 2.1% | 0.99 |
| HO₂, aq | 0.2% | 1.00 | 0.7% | 1.00 | 0.1% | 1.00 |
| O₂⁻aq | -0.1% | 1.00 | 2.8% | 1.00 | 3.0% | 1.00 |




**Table 2**  **Description of the lumped MOZART4 species and the corresponding kinetic reaction rate constants.**

| Species | Comment | k | Comment on k |
|---------|---------|---|--------------|
| BIGALK | Alkanes with C $\geq$ 4 | $2.05\times10^{-10}$ | same as for butane |
| ALKOH | Alcohols with C $\geq$ 3 | $2.7\times10^{-11}e^{525}/_T$ | same as for propanol |
| $C_2H_5CHO$ | Aldehydes with C $\geq$ 3 | $4.9\times10^{-12}e^{405}/_T$ | OH; same as for propionaldehyde |
| | | $1.3\times10^{-10}$ | Cl; same as for propionaldehyde |
| | | $5.75\times10^{-11}e^{-610}/_T$ | Br; same as for propionaldehyde |
| BIGALD1 | Unsaturated dialdehyde | $1.35\times10^{-10}$ | same as for 2-butenedial |
| XYL | Lumped xylenes | $1.4\times10^{-10}$ | mean value of o-, p- and m-xylene |
| BZALD | Lumped aromatic aldehydes | $1.0\times10^{-10}$ | same as for benzaldehyde |