# Peer review of "CAPRAM reduction towards an operational multiphase halogen and DMS chemistry treatment in the chemistry transport model COSMO-MUSCAT(5.04e)"

_Geoscientific Model Development, 2019_

## Referee Comment (RC1) · Anonymous Referee #1 · 25 Feb 2020

Hoffmann et al. developed simplified multiphase halogen and DMS chemistry schemes that consumed less CPU time and could be implemented into the chemical transport models. Along with the development of observation techniques, halogen chemistry is getting more attention in the atmospheric chemistry community during recent years. DMS chemistry is critical for the climate through formation of sulfate aerosols and clouds. Both halogen and DMS chemistry involves many chemical species and re-actions and thus are difficult to model, especially the multiphase chemistry parts that are generally not included in CTMs. The chemistry schemes developed in this stud will

benefit the atmospheric chemistry community. It is within the scope of GMD. However, I think the manuscript can be improved through more discussion about the results (some unclear scientific reasoning), doing sensitivity tests, and adding references to the reaction coefficients used in this study. I recommend it to be accepted after some revisions.

General comments

1. In the Model Setup section, Lines 105-110, it states that the simulations were performed at 48 hours, different latitudes, different seasons, and different relative humidity levels. But I was not able to find where the results for all these simulations are in the manuscript. Please clarify this.

2. The CPU time evaluation was shown for the box model with the new chemistry schemes. It will be worth showing the CPU time evaluation for the 2-D modeling before and after using the new schemes.

3. References should be added or clarified to all the coefficients shown in the tables in the Supplement.

4. There should be discussions about wet scavenging of reactive halogens by clouds when explaining the cloud impacts on halogen and sulfur chemistry.

5. It will be useful to have a section discussing the main uncertainties of the new chemistry schemes (e.g. reaction coefficients).

Other comments

1. Page 3, Line 32: What is the HOX-driven sulfite oxidation? Was it included in the model?

2. Line 120: Please give references.

3. Fig.1: Please clarify shading in the figure. Nighttime?

4. Lines 362-363: Please clarify this sentence.

[Figure]

5. Line 489: 200 horizontal columns – does it mean 222 km?

6. Fig. 5: Please clarify the x-axis "Distance in grid". What does it mean and what unit.
7. Line 515, BrO gas-phase concentration section: Please explain the differences of BrO in two scenarios shown in Fig. 6.

8. Sect. 3.3.2 Vertical DMSO distribution: It will be useful to show plots of DMSO production and loss rates when discussing the DMSO profiles.

9. Line 595: Why particularly BrCl? What about Br2?

10. Lines 598-599: Please clarify more why DMSO concentration profile is shifted to the right compared to the BrO one.

11. Line 612: Please explain "O3 is the preferred oxidant in the aerosol phase whereas OH is in the cloud droplets"?

12. Line 617: Please explain "In the grid cells before cloud occurrence, the DMSO concentration is high and consequently the MS- formation is as well".

13. Line 625: Please clarify "As for MSA, the formation of sulfate is enhanced in the grid cells directly or indirectly affected by clouds".

14. Line 649: How much does the HOBr+HSO3- in clouds affect the bromine budget through converting HOBr into Br-?
* * *

---

## Referee Comment (RC2) · Anonymous Referee #2 · 26 Feb 2020

The authors have developed a reduced version of two chemical mechanisms that consider multiphase reactions of dimethylsulfide and reactive halogen species by identifying the primary pathways through which key atmospheric products are formed. The goal of this work is to be able to account for the combined effects of these important multiphase mechanisms in large-scale models for which more comprehensive mechanisms are computationally infeasible. The reduced mechanism is evaluated against a detailed scheme including the two "pre-reduction" versions of the mechanisms under several atmospherically relevant sets of conditions. The authors use the reduced

mechanism to draw conclusions on the various factors contributing to DMS oxidation in cloudy environments and reveal strong direct and indirect effects of clouds on this process.

The work is novel in that it provides a new means to account for the combined effects of multiphase dimetylsulfide oxidation and reactive halogen chemistry in large-scale models. The methodology by which the reduction is performed is well reasoned and follows established approaches for such reductions. The evaluation of the reduced mechanism is convincing and the conclusions drawn from the results of incorporating these chemical processes in a chemical transport model are well argued. It is recommended that this article be published in GMD after consideration of a few comments.

The only major comment is that the grammar in certain sections of the manuscript makes interpretation of the arguments difficult at times. I would recommend that the manuscript be edited for grammar by a native English speaker, or someone similarly fluent in English. Some examples (but not all cases) of such passages are included the following comments.

Lines 80–81: "not only the solvation of the high CPU consumption is necessary"

"Solvation" refers to solvent–solute interactions. Also, it is not entirely clear what is meant here. Is not the point of reduced mechanisms to reduce CPU consumption? Does this refer to efforts to develop more efficient numerical solvers for chemical systems?

Line 83–84: "An adequate mechanism does not currently exist and can only be derived by reducing detailed multiphase chemistry mechanisms."

I'm not sure that I agree that this is the "only" means to generate such a mechanism. For example, have there not been some machine learning-based techniques applied to simulating atmospheric chemical species transformations based purely on observations? It seems these types of approaches could lead to similarly predictive models,

but may shed less light on the underlying chemistry.

Line 91 "the various effects of clouds essentially on halogens and DMS."

It is not clear what the word 'essentially' means here.

Lines 103–104: "The goal of reducing the CAPRAM-DM1.0 and CAPRAM-HM3.0 is that both modules can be applied in different marine atmospheric environments in CTMs."

This is somewhat unclear. It seems that the results of this work are a single combined reduced mechanism. However, this sentence makes it seem like the goal is to develop two separate reduced mechanisms, and that a CTM can choose which to apply (possibly together or separately?) to particular grid cells. If this is the case, maybe this could also be included in the introduction when the goals of the work are first stated.

Lines 132–140

I find it somewhat unclear how the 'importance' of a chemical species is determined here and would prefer a slightly more specific description of why certain species are included in the evaluation. Do you expect that the choice of which species to include in the evaluation, and at what acceptable level of accuracy, would have a large effect on which chemical pathways end up in the reduced mechanism?

Line 268: "As already modelled in other studies, the analyses revealed that the Cl atom is an important oxidant for VOCs and OVOCs"

References for these other studies should be included.

Lines 274–277: "Therefore, a first screening on treated VOCs and OVOCs in the mechanisms MOZART4.0 (Schultz et al., 2018), 275 RACM2 (Goliff et al., 2013), MECCA (JoÌLckel et al., 2016), GEOS-Chem (Wang et al., 2019),and SAPRC11 (Yan et al., 2019) has been performed for the main VOCs and OVOCs. As a result, only the Cl atom oxidation of the lumped VOCs and OVOCs that are treated within all of these

mechanisms is considered further."

This could be clarified. Do you first determine which organic species are included in the various lumped species in each model? Do the lumped species in the various models comprise the same sets of actual organic species? How do you determine the rate at which Cl reacts with a lumped species? If one mechanism excludes a specific organic species, are its reactions with Cl automatically excluded from the reduced mechanism? I am not clear on why this is necessary.

Line 317–319: "The evaluation simulations are carried out for $45°$ latitude with a relative humidity of 70 % under pristine ocean (Hoffmann et al., 2016) and polluted coastal conditions (Hoffmann et al., 2019a)."

In the previous section, several latitudes and relative humidities are modeled. Why are these not used in the evaluation of the reduced model? Can you provide an argument that these scenarios are sufficient to evaluate the reduced mechanism?

Line 378–379: "Consequently, the high CPU time required overlay the CPU time consumption from the reduction."

This should be clarified.

Line 560 Figure 8

These is an unusually square feature in Fig. 8b for aqueous-phase DMSO directly under the cloud. Is this a result of the way cloud grid-cells and aerosol grid-cells are treated in the model (Fig. 3)?

Line 617–618: "Due to the advection of the stable MS- to the right-hand side of the model domain, the spatial DMSO profile is not modelled."

Is this not shown in Figure 8?

---

## Author Comment (AC1) · 15 Apr 2020

**Response to Anonymous Reviewer #1**

Hoffmann et al. developed simplified multiphase halogen and DMS chemistry schemes that consumed less CPU time and could be implemented into the chemical transport models. Along with the development of observation techniques, halogen chemistry is getting more attention in the atmospheric chemistry community during recent years. DMS chemistry is critical for the climate through formation of sulfate aerosols and clouds. Both halogen and DMS chemistry involves many chemical species and reactions and thus are difficult to model, especially the multiphase chemistry parts that are generally not included in CTMs. The chemistry schemes developed in this stud will benefit the atmospheric chemistry community. It is within the scope of GMD. However, I think the manuscript can be improved through more discussion about the results (some unclear scientific reasoning), doing sensitivity tests, and adding references to the reaction coefficients used in this study. I recommend it to be accepted after some revisions.

We thank the anonymous reviewer 1 for the recommendation and constructive comments to improve the manuscript. The comments of the reviewer are carefully addressed point by point in the sections below. The answers to the reviewer comments are marked as blue text.

Because of the comment of reviewer 2 the whole manuscript has been checked for grammar by an English language professional at the C2 level. For further information, please see the manuscript version with highlighted changes.

Besides, due to a personal comment from the community regarding that ECHAM-HAMMOZ and EMAC are chemistry climate models, we have deleted ECHAM-HAMMOZ from Figure 4 and added to the Figure caption that EMAC can be run as a CTM. Please, see the changes below:

[Figure]

**Figure 4**      **Comparison of applied tropospheric DMS and halogen chemistry mechanisms within the chemical transport models: CMAQ (Muniz-Unamunzaga et al., 2018), TOMCAT (Hossaini et al., 2016), Geos-Chem (Wang et al., 2019), CAM-Chem (Saiz-Lopez et al., 2014), CAM-MECCA (Long et al., 2014), EMAC (Jöckel et al., 2016), and WRF-Chem (Badia et al., 2019). EMAC is a chemistry climate model that can be run as a CTM.**

General comments

1. In the Model Setup section, Lines 105-110, it states that the simulations were performed at 48 hours, different latitudes, different seasons, and different relative humidity levels. But I was not able to find where the results for all these simulations are in the manuscript. Please clarify this.

As outlined in the main manuscript, we analysed the modelled sink and source fluxes in detail to receive the reduced mechanisms. For the analysis, we calculated the mean contribution of specific pathways directly from the model output, but did not analyse all concentration time-profiles. In the opinion of the authors, the implementation of such results will not help to better understand the manuscript, but might lead to more confusion. Furthermore, in the term of the coastal simulations, the most important model outcomes are already published (see Hoffmann et al., 2019a; Hoffmann et al., 2019b). Thus, it is not necessary that these results are again included in the present manuscript. However, in terms of completeness, we have included modelled concentration time profiles of target compounds from the pristine ocean simulations into the supplement and linked to the previous published papers covering the results for the coastal simulations, which reads as follows:

Lines 123-124
'The modelled concentration time profiles of specific important trace gases and aerosol compounds within the pristine boundary layer are given in Fig. S1 to S10 in the supplement.'

Lines 132-134
'Details on the model configurations of the first simulation are explicitly given in Hoffmann et al. (2019b) and those of the second simulation in Hoffmann et al. (2019a). The main model results of both simulations do not differ from the present work.'

An example for the simulation at 45° latitude is given below.

[Figure]

**Figure S5 Modelled concentration time-profile of key compounds within the pristine marine boundary layer for the summer simulations at 45° latitude. Red: simulation at rel. humidity of 50% (red). Orange: simulation at relative humidity of 70% and cloud occurrence at early morning and evening of the first model day. Dark green: simulation at relative humidity of 70% and cloud occurrence at noon and midnight. Blue: simulation at relative humidity of 90% and cloud occurrence at noon and midnight.**

Besides, we have found a wrong statement regarding the simulation description for the simulations with rel. humidity of 70% and cloud occurrence at early morning. In these simulations cloud occurred at the first model day at early morning and evening. The sentence has been rewritten and reads now as follows:

Lines 119-121
'The in-cloud residential time of the air parcel is two hours in the simulations with cloud occurrences at noon and midnight and three hours in those with cloud occurrences in the early morning and evening of the first model day.'

2. The CPU time evaluation was shown for the box model with the new chemistry schemes. It will be worth showing the CPU time evaluation for the 2-D modeling before and after using the new schemes.

Before this study, COSMO-MUSCAT did not treat multiphase chemistry of reactive halogen species and DMS. To investigate the additional CPU costs with regard to the DMS and halogen chemistry, we have

performed a simulation without CAPRAM-DM1.0red and CAPRAM-HM3.0red. The CPU time increases by 171% (from 18,420 s to 31,432 s) and 180% (from 52,815 s to 94,896 s) when the simulations are performed with CAPRAM-DM1.0red and CAPRAM-HM3.0red in the 'stable meteorological conditions' and 'unstable meteorological conditions' simulations, respectively.

[Figure]

*Figure 1  CPU time when the simulations are performed with and without added CAPRAM-DM1.0red and CAPRAM-HM3.0red.*

3. References should be added or clarified to all the coefficients shown in the tables in the Supplement.

References have been added (see Tables S2 to S10).

4. There should be discussions about wet scavenging of reactive halogens by clouds when explaining the cloud impacts on halogen and sulfur chemistry.

Thanks for this suggestion which will improve the manuscript. The current COSMO-MUSCAT(5.04e) treats wet scavenging of gases by a first-order scavenging rate without underlaying aqueous-phase chemistry. However, a more detailed scheme is implemented for fog droplets, only. The modelled clouds do not precipitate and, thus, this effect would not be of interest in this study. The implementation is a necessary approach that is planned in the future. We have added a discussion on the potential impacts into the section conclusion and outlook which reads as follows:

Lines 718-721
'The important effect of wet scavenging by clouds was not investigated as the current COSMO-MUSCAT(5.04e) did not implemented it in detail, but represented it using a first-order scavenging rate.

Future studies aim to implement a more precise scheme. Since the clouds modelled in this study are not known to precipitate, the propagated error should be small.'

5. It will be useful to have a section discussing the main uncertainties of the new chemistry schemes (e.g. reaction coefficients).

We agree with the reviewer that the uncertainties have to be discussed. However, we have to mention that the new reduced chemistry mechanism is focused on the CAPRAM-DM1.0 and CAPRAM-HM3.0. In the original publication, a discussion of the uncertainties of the main reaction rate constants and products is already given. Furthermore, uncertainties and their discussion are given in the original literature in which the kinetics were determined, too. Therefore, in the first place, we did not include such a section. However, due to the application of the PSSA for the DMS oxidation some uncertainties could exist, because of the changes in the oxygen concentration with height. These were checked and found to be rather small.

A short section that discusses the uncertainties is now added to the revised manuscript and reads as follows:

Lines 421-439
**2.5 Uncertainties of the new chemistry scheme**
'The downsizing of the CAPRAM-HM3.0 and CAPRAM-DM1.0 solely considering the most important reactions, has led to two new reduced mechanisms, which consist of reactions that sensitively impact the model outcome. Hence, the uncertainty of these reactions can be crucial for the model results. A discussion of the uncertainties of the mechanism development has already been performed in the previous papers describing the mechanism development of the CAPRAM-HM2.0 (Bräuer et al., 2013), CAPRAM-HM3.0 (Hoffmann et al., 2019a; Hoffmann et al., 2019b) and CAPRAM-DM1.0 (Hoffmann et al., 2016) as well as in the cited laboratory work. That's why only a short discussion is given here.

For the oxidation of DMS in the gas-phase, most of the rate constants are based on recommended values of the IUPAC database (http://iupac.pole-ether.fr/) or JPL panel (Burkholder et al., 2015). Nevertheless, the application of the PSSA has modified the oxidation pathways, in particular the OH-addition reaction for DMS. The incorporation of the oxygen concentration might increase the uncertainty influencing the DMSO formation rate. As oxygen is in excess under tropospheric conditions and the oxygen concentration is treated in the new derived reaction rate constant specifically, minor changes are expectable. By contrast, no recommended values are available for the aqueous-phase reaction rate constants and can hence be stated as more uncertain (Hoffmann et al., 2016). Further laboratory work is required to minimize their uncertainties.

Regarding the CAPRAM-HM3.0red, certain gas-phase reaction rate constants of the halogen chemistry are based on recommended values (Atkinson et al., 2006, 2007; Atkinson et al., 2008; Burkholder et al., 2015). However, for the oxidation of VOCs/OVOCs by the Cl or Br atom often only one reaction rate constant has been measured by laboratory studies. This is also true for many aqueous-phase chemistry reactions. The highest uncertainties are related to iodine chemistry. Here, often reaction rate constants, which might be of high atmospheric significance are based on estimations, only.'

Other comments

1.  Page 3, Line 32: What is the HOX-driven sulfite oxidation? Was it included in the model?

The term HOX refers to hypohalous acids. The model includes HOCl, HOBr, and HOI. These hypohalous acids are very reactive compounds and are known to oxidise effectively sulfite into sulfate in the aqueous phase (Chen et al., 2017; Pechtl and von Glasow, 2007; von Glasow et al., 2002). This chemistry is included. For a better clarification, we have added a description what X means, which reads as follows:

Lines 32-33
'The photolysis of hypohalous acids (HOX, X = Br, Cl, or I) is reduced as well, resulting in higher HOX-driven sulfite to sulfate oxidation in aerosol particles below stratiform clouds.'

2.  Line 120: Please give references.

We have added some references to our previous model studies and additionally cite three literature reviews on the topic. These are:
Hoffmann et al. (2019a)
Hoffmann et al. (2019b)
Faxon and Allen (2013)
Saiz-Lopez and von Glasow (2012)
von Glasow et al. (2013)

Lines 125-129
'The lower diversity of the simulations compared to the pristine ocean scenario is chosen because previous model studies had revealed that high $NO_x$ concentrations suppress gas-phase halogen radical cross reactions and lead to a domination of halogen nitrate and nitryl chloride photolysis in halogen atom activation (Hoffmann et al., 2019a; Hoffmann et al., 2019b; Faxon and Allen, 2013; Saiz-Lopez and von Glasow, 2012; von Glasow et al., 2013).'

3.  Fig.1: Please clarify shading in the figure. Nighttime?

Yes, the shading in the figure is representing the nighttime. We have added a description into the Figure caption, which reads as:

Lines 358-359
'Grey shaded periods denote the night periods.'

4.  Lines 362-363: Please clarify this sentence.

This sentence was included to outline that the difference for the HOCl concentrations are not a driving factor for $Cl_2$ activation, because different to bromide, the chloride concentrations in sea spray aerosol are still in excess. We have rephrased the sentence in terms of clarity, i.e. It reads now as follows:

Lines 392-394

'As opposed to $Br_2$ formation by HOBr, the higher HOCl does not necessarily lead to a higher modelled $Cl_2$ formation, which is related to the significant higher chloride content in sea spray aerosols compared to bromine. It so concludes that the enhanced HOCl seems not to be a driving factor for $Cl_2$ formation under pristine ocean conditions.'

5. Line 489: 200 horizontal columns – does it mean 222 km?

Yes, it means 222 km. We have added the value in brackets and did this also for the vertical levels. The text reads now as follows:

Lines 537-538

'Whereas COSMO is run on the full domain, only the inner 200 horizontal grid cells (overall 222 km) and lowermost 15 vertical levels (overall 1500 m) are used for the multiphase chemistry simulations with MUSCAT.'

6. Fig. 5: Please clarify the x-axis "Distance in grid". What does it mean and what unit.

This term refers to the model grid resolution, which means the distance from 1 to 2 is 111 km. We added the following sentence to the corresponding figure captions and changed the x-axis in the figures to 'Grid cell number / $10^{2}$' for clarification. Furthermore, because of the grammar checking we have rephrased the sentence describing the clouds modelled in the 2D simulations. The caption text for all 2D simulation reads as follows:

Line 559

'The x-axis represents the innermost horizontal grid cells divided by 100.'

Lines 561-562

'The black line corresponds to a liquid water content of 0.01 g m$^{-3}$ and the white line to 0.1 g m$^{-3}$. The area framed by the white line includes LWC above 0.1 g m$^{-3}$.'

7. Line 515, BrO gas-phase concentration section: Please explain the differences of BrO in two scenarios shown in Fig. 6.

A discussion of the differences has been added, which reads as follows:

Lines 569-576

'Apart from that, the vertical distribution significantly differs between both simulations resulting into distinct spatial pattern. At the left-hand side of the model domain, the BrO concentration is similar, which is related to the activation of reactive bromine species from the initialised marine aerosols. However, when clouds are formed the profiles change. This is related to the high differences in the vertical wind field (see Fig. 7a and b). Because of the stronger updrafts in the 'unstable meteorological condition' simulation, the reactive halogen compounds are advected towards higher altitudes compared to the slow vertical winds in the 'stable meteorological condition' simulation. A second remarkable difference is the much lower BrO concentration at the right-hand side of the model domain

in the 'stable meteorological condition' simulation. This effect is more explicitly discussed in Sect. 3.3.3.'

8. Sect. 3.3.2 Vertical DMSO distribution: It will be useful to show plots of DMSO production and loss rates when discussing the DMSO profiles.

A figure with the modelled DMSO production and loss rates separated into gas and aqueous-phase reactions has been added to the supplement (see Fig. S 11). The link in the manuscript reads as follow:

Lines 626-627

'Furthermore, the overall modelled DMSO production and loss rates separated into gas and aqueous-phase reactions were added to the supplement (see Fig. S11).'

[Figure]

**Figure S11 Modelled formation rate of DMSO in (a) the gas phase and (c) the aqueous phase together with the modelled oxidation rate in (b) the gas phase and (d) the aqueous phase in the 'stable meteorological condition' simulation with stratiform clouds after 12 hours of modelling time. The x-axis represents the innermost horizontal grid cells divided by 100.**

9. Line 595: Why particularly BrCl? What about Br2?

Because of the much higher concentration of Cl$^-$ in the modelled aerosol particles, the HOBr is modelled to preferably react with Cl$^-$ to form BrCl. The modelled BrCl concentrations under the cloud

are one order of magnitude higher than the $Br_2$ concentrations. However, because of the higher photolysis rate constant, $Br_2$ is enhanced under the cloud and thus this also important. We have added $Br_2$ to the text and deleted the word "particularly". The text reads now as follows:

Lines 651-654
'This cycle is disturbed by the reaction of BrO with $HO_2$, yielding HOBr, which can be photolyzed back into the Br atom again or converted by multiphase chemistry into BrCl or $Br_2$. Overall, the photolysis of HOBr, $Br_2$ and BrCl determine the DMS to DMSO conversion. Clouds suppress the photolysis of $Br_2$, BrCl and HOBr due to the reflection of incoming solar radiation.'

10. Lines 598-599: Please clarify more why DMSO concentration profile is shifted to the right compared to the BrO one.

A more detailed explanation is given in the revised manuscript now, which reads as follows:

Lines 656-659
'Due to a longer lifetime against further oxidation and corresponding horizontal advection, the DMSO concentration profile is shifted towards the right compared to BrO. The lowest oxidation flux between DMS and BrO is modelled between grid cell 2.0 and 2.15. The effect on DMSO concentration is modelled between grid cell 2.1 to 2.4.'

11. Line 612: Please explain "O3 is the preferred oxidant in the aerosol phase whereas OH is in the cloud droplets"?

These statements were derived from the results of our previous modelling study Hoffmann et al. (2016) and focused on MSIA oxidation. We have added the reference and rephrased the corresponding text, which reads now as follows:

Lines 671-674
'DMSO is rapidly oxidised into MSIA and thus a similar MSIA profile is modelled. As MSIA is highly reactive in the gas and aqueous phases as well as highly soluble, it is rapidly oxidised into methane sulfonate ($MS^-$) in both the aerosol and the cloud phases. There, $O_3$ is the preferred oxidant in the aerosol phase, whereas in cloud droplets it is the OH radical (Hoffmann et al., 2016).'

12. Line 617: Please explain "In the grid cells before cloud occurrence, the DMSO concentration is high and consequently the MS- formation is as well".

This sentence is related to the rapid oxidation of DMSO in gas and aqueous phase. The formed MSIA is rapidly taken up by sea salt aerosols and is fast oxidised to $MS^-$ and sulfate. We have changed the sentence to make it clearer. The sentence reads now as follows:

'In the grid cells left of the cloud, the DMSO concentration is high and consequently the aerosol particle chemistry of DMSO and of the subsequent oxidation product MSIA leads to a sharp increase of MS-formation at the grid cells below the left cloud edge (see Fig. 10a).'

13. Line 625: Please clarify "As for MSA, the formation of sulfate is enhanced in the grid cells directly or indirectly affected by clouds".

The sentence refers to the similar concentration profiles of MSA and sulfate at the left cloud edge. The sentence has been rewritten for more clarity and reads now as follows:

'Also, the concentration of sulfate (see Fig. 10b) is enhanced in the grid cells at the left cloud edge, but because of different reasons.'

14. Line 649: How much does the HOBr+HSO3- in clouds affect the bromine budget through converting HOBr into Br-?

The concentration of $Br^-$ is increased inside the cloud droplets by up to one order of magnitude. An additional text discussing this issue has been added and a corresponding figure has been places into the supplement. The additional text in manuscript reads as follows:

[revised manuscript text omitted]

---

## Author Comment (AC2) · 15 Apr 2020

**Response to Anonymous Reviewer #2**

The authors have developed a reduced version of two chemical mechanisms that consider multiphase reactions of dimethylsulfide and reactive halogen species by identifying the primary pathways through which key atmospheric products are formed. The goal of this work is to be able to account for the combined effects of these important multiphase mechanisms in large-scale models for which more comprehensive mechanisms are computationally infeasible. The reduced mechanism is evaluated against a detailed scheme including the two "pre-reduction" versions of the mechanisms under several atmospherically relevant sets of conditions. The authors use the reduced mechanism to draw conclusions on the various factors contributing to DMS oxidation in cloudy environments and reveal strong direct and indirect effects of clouds on this process.

The work is novel in that it provides a new means to account for the combined effects of multiphase dimetylsulfide oxidation and reactive halogen chemistry in large-scale models. The methodology by which the reduction is performed is well reasoned and follows established approaches for such reductions. The evaluation of the reduced mechanism is convincing and the conclusions drawn from the results of incorporating these chemical processes in a chemical transport model are well argued. It is recommended that this article be published in GMD after consideration of a few comments.

The only major comment is that the grammar in certain sections of the manuscript makes interpretation of the arguments difficult at times. I would recommend that the manuscript be edited for grammar by a native English speaker, or someone similarly fluent in English. Some examples (but not all cases) of such passages are included the following comments.

We thank the reviewer for the very positive comments, recommendations and suggestions to improve the manuscript. The comments of the reviewer are carefully addressed point by point in the sections below. The answers to the reviewer comments are marked as blue text. According to the reviewer's comment, the manuscript has been thoroughly checked for grammar by ourselves again. Furthermore, the manuscript has been checked by an English language professional at the C2 level. The corresponding changes are provided in the manuscript version with tracked changes. The further discussion of the comments can be seen below.

Because of the manuscript checking we have rephrased the figure caption describing the clouds modelled in the 2D simulations. The caption text for all 2D simulation reads as follows:

Lines 561-562
'The black line corresponds to a liquid water content of 0.01 g m$^{-3}$ and the white line to 0.1 g m$^{-3}$. The area framed by the white line includes LWC above 0.1 g m$^{-3}$.'

Besides, due to a personal comment from the community regarding that ECHAM-HAMMOZ and EMAC are chemistry climate models, we have deleted ECHAM-HAMMOZ from Figure 4 and added to the Figure caption that EMAC can be run as a CTM. Please, see the changes below:

[Figure]

**Figure 4**    Comparison of applied tropospheric DMS and halogen chemistry mechanisms within the chemical transport models: CMAQ (Muniz-Unamunzaga et al., 2018), TOMCAT (Hossaini et al., 2016), Geos-Chem (Wang et al., 2019), CAM-Chem (Saiz-Lopez et al., 2014), CAM-MECCA (Long et al., 2014), EMAC (Jöckel et al., 2016), and WRF-Chem (Badia et al., 2019). EMAC is a chemistry climate model that can be run as a CTM.

Lines 80–81: "not only the solvation of the high CPU consumption is necessary"
"Solvation" refers to solvent–solute interactions. Also, it is not entirely clear what is meant here. Is not the point of reduced mechanisms to reduce CPU consumption? Does this refer to efforts to develop more efficient numerical solvers for chemical systems?

The sentence intends to say that besides reduction of CPU consumption due to effective numerical solutions in CTMs an appropriate mechanistic description is important, too. We have changed "solvation" to "solution". The text passage now reads:

Line 83
'not only a solution for the high CPU consumption is necessary'

Line 83–84: "An adequate mechanism does not currently exist and can only be derived by reducing detailed multiphase chemistry mechanisms."
I'm not sure that I agree that this is the "only" means to generate such a mechanism. For example, have there not been some machine learning-based techniques applied to simulating atmospheric chemical species transformations based purely on observations? It seems these types of approaches could lead to similarly predictive models, but may shed less light on the underlying chemistry.

We agree with the reviewer that a machine learning approach could lead to a simple predictive model. However, for such an approach to get there a high amount of measurement data is required and a prediction might only be suitable for the current applied location. Thus, the underlaid chemistry is still required, as otherwise a huge number of parameterisations must be included slowing down CPU performance. Furthermore, a simple machine learning approach from observations will not help to understand the pathways responsible for the formation, as it will rely on statistical correlation, nor will

be able to take deposition into account. For example, rain will reduce the sulfate and nitrate concentration, but not necessarily the DMS and $NO_x$ concentration.

Hence, only the reduction of comprehensive mechanisms by sophisticated model studies is able to generate such a detailed but reduced mechanism. To make the statement clearer, we have rephrased the sentence in the revised manuscript. It now reads:

Lines 84-86
'Currently, an adequate mechanism does not exist and can be derived by reducing detailed multiphase chemistry mechanisms, because important chemical pathways could otherwise be missed resulting in a misinterpretation of field data.'

Line 91 "the various effects of clouds essentially on halogens and DMS."
It is not clear what the word 'essentially' means here.

We have deleted the word "essentially" as it is not really necessary at this point.

Lines 103–104: "The goal of reducing the CAPRAM-DM1.0 and CAPRAM-HM3.0 is that both modules can be applied in different marine atmospheric environments in CTMs."
This is somewhat unclear. It seems that the results of this work are a single combined reduced mechanism. However, this sentence makes it seem like the goal is to develop two separate reduced mechanisms, and that a CTM can choose which to apply (possibly together or separately?) to particular grid cells. If this is the case, maybe this could also be included in the introduction when the goals of the work are first stated.

Yes, the overall result of this work is a single combined reduced mechanism as it was currently not available for COSMO-MUSCAT. However, as can be seen from the tables in the supplement, the reduction is designed that also only one reduced mechanism or specific reactions could be implemented into CTMs. Because of this, interested modellers could use still their own core mechanism, but can add the marine mechanism module or insights developed in this study, too. This goal was added into the introduction. The new added sentences read as follows:

Lines 90-95
'During the reduction procedure, two mechanisms are derived, which are afterwards combined into a single one. The combined reduced mechanism is implemented into the CTM MUSCAT (MUltiScale Chemistry Aerosol Transport; Wolke et al., 2004; Wolke et al., 2012), which now treats detailed marine multiphase chemistry. Finally, the combined reduced mechanism is applied in idealised 2D simulations with a focus on multiphase DMS oxidation in the MBL and the various effects of clouds on halogens and DMS.'

Lines 132–140
I find it somewhat unclear how the 'importance' of a chemical species is determined here and would prefer a slightly more specific description of why certain species are included in the evaluation. Do you expect that the choice of which species to include in the evaluation, and at what acceptable level of accuracy, would have a large effect on which chemical pathways end up in the reduced mechanism?

Our 5% goal has been determined for the known 'classical' air pollutants, ozone, $SO_2$, $NO_x$ and PM mass. For the DMS chemistry, we have chosen the most important stable DMS oxidation products DMSO and MSA that are often incorporated in higher scale models. For the reactive halogen chemistry, compounds were chosen that are relevant to understand the gas-phase chemistry of reactive halogens as outlined in review articles. Given the limitations of available measurement techniques, especially for the high reactive halogen compounds, in our opinion 10% deviation is acceptable for these species. Indeed, reducing the goal for example down to 2% will increase the number of reactions that have to be considered, but will not necessarily improve the predictions for air pollutants or greenhouse gases such as ozone. Furthermore, an appropriate reduction is not possible anymore. We have included a more specific description, why these compounds are chosen in the revised manuscript which is given in the following.

Lines 142-158
'The goal of the mechanism reduction firstly is that the modelled concentration of chemical species that are classically treated as important in CTMs, e.g. ozone, $SO_2$, $NO_x$, sulfate or nitrate, only deviate from the modelled concentration obtained from the complete scheme by less than 5 % on average over the full modelling time. Secondly, concentrations of oxidants and important chemical compounds of marine multiphase chemistry should only differ by less than 10 % on average. Important chemical compounds of DMS multiphase oxidation are DMS, dimethyl sulfoxide (DMSO), and methane sulfonic acid (MSA), which represent the key stable compounds from DMS oxidation. Dimethyl sulfone ($DMSO_2$) is not considered, because, according to current scientific knowledge, the oxidation of $DMSO_2$ is negligible under atmospheric conditions of the pristine ocean. Additionally, the deviation of the concentration of methane sulfinic acid (MSIA) is not a reduction criterion. MSIA is very reactive, so that even slight changes will immediately result in differences of the MSA and sulfate concentrations. In the case of halogen multiphase chemistry, important species for the mechanism reduction are the Cl, Br, and I atoms as well as the ClO, BrO, and IO radical and stable halogen compounds, which can act as important reservoir or activation species for halogen radicals, i.e. hypohalous acids, nitryl chloride, and dihalogen molecules. These halogen reservoir/activation species are of high importance as strong changes in their budget will obviously affect the overall oxidation processes in the MBL. For other halogen radicals, it has been shown by previous studies (e.g. see Saiz-Lopez et al. 2012 and Simpson et al., 2015) that these are rapidly converted into the above-mentioned compounds and thus strong concentration changes will show up in the concentration of X atoms or XO radicals. Lastly, the pattern of the concentration time profile for all species has to match between the reduced and the full mechanism ($R^2 \geq 0.75$).'

Line 268: "As already modelled in other studies, the analyses revealed that the Cl atom is an important oxidant for VOCs and OVOCs"
References for these other studies should be included.

References have been added. These are:
Hoffmann et al. (2019a)
Sherwen et al. (2016)
Xue et al. (2015)
Pechtl and von Glasow (2007)

Lines 290-292

'As already modelled in other studies, the analyses revealed that the Cl atom is an important oxidant for VOCs and OVOCs, e.g. alkanes, non-oxidised aromatic compounds, alcohols, and aldehydes (e.g. Hoffmann et al., 2019a; Sherwen et al., 2016; Xue et al., 2015; Pechtl and von Glasow, 2007).'

Lines 274–277: "Therefore, a first screening on treated VOCs and OVOCs in the mechanisms MOZART4.0 (Schultz et al., 2018), 275 RACM2 (Goliff et al., 2013), MECCA (Jöckel et al., 2016), GEOS-Chem (Wang et al., 2019), and SAPRC11 (Yan et al., 2019) has been performed for the main VOCs and OVOCs. As a result, only the Cl atom oxidation of the lumped VOCs and OVOCs that are treated within all of these mechanisms is considered further."
This could be clarified. Do you first determine which organic species are included in the various lumped species in each model? Do the lumped species in the various models comprise the same sets of actual organic species? How do you determine the rate at which Cl reacts with a lumped species? If one mechanism excludes a specific organic species, are its reactions with Cl automatically excluded from the reduced mechanism?
I am not clear on why this is necessary.

First, the mechanisms were screened for lumped species. Then, the lumped species were analysed on included organic species. Obviously, the models did not comprise the same sets, but mostly the same VOC/OVOC compound classes, e.g. all mechanisms contain a lumped alkane, aldehyde, ketone and aromatics compound. Therefore, the lumped species implemented in our mechanism are also present in the other mechanisms. As we considered the implementation to the new MOZART mechanism, our mechanism contains a lot of VOC/OVOC oxidations. This set can be applied to the other mechanisms, but has to be adjusted if certain species are missing. The chosen reaction rate constant is based on the first lumped product. As example, for aldehydes the lumped species is C2H5CHO and the first implemented aldehyde is $CH_3CH_2CHO$. Here, the reaction rate constant k of $CH_3CH_2CHO$ with Cl is chosen. As the reaction rate constants are getting higher with longer chain length, this approach is suitable, as it gives the lower limit of the reaction rate constant. Thus, no overestimation will occur.
The statement has been added to better understand the model results from the simulation comparisons. The text now reads as given below.

Lines 292-307

'In order to restrict computational costs, chemical mechanisms in state-of-the-art CTM applications do not contain a high number of organic compounds as the near-explicit MCM. In order to still represent the chemistry of important VOCs and OVOCs in CTMs, species of the same compound classes or of equal reactivity are typically merged into 'lumped' species in condensed mechanisms applied in CTMs (Baklanov et al., 2014). Based on these limitations, the reduced CAPRAM-HM3.0 has to be linkable with the chemical mechanisms used in CTMs. A first screening on treated VOCs and OVOCs in the gas-phase chemical mechanisms MOZART4.0 (Schultz et al., 2018), RACM2 (Goliff et al., 2013), MECCA (Jöckel et al., 2016), GEOS-Chem (Wang et al., 2019), and SAPRC11 (Yan et al., 2019) has been performed for the main VOCs and OVOCs for this purpose. It has been shown that most of the mechanisms contain the same set of primary VOC/OVOC compound classes, for example, aldehydes and alcohols are often treated up to a carbon number of two. As outlined in Sect. 3, the gas-phase mechanism MOZART4 is chosen for further modelling with COSMO-MUSCAT. As a result, only the Cl atom oxidation of the lumped VOCs and OVOCs that are treated within MOZART4 are considered

further. These sets can be applied to the other mechanisms, but have to be adjusted if species are missing. The chosen reaction rate constant is based on the first lumped product. As the k's are higher with longer carbon chain, this approach is suitable, as it gives a lower limit of the reaction rate constant. Thus, no overestimation will occur. However, when the simulations with the reduced version of the CAPRAM-HM3.0 are compared to the simulations with the non-condensed CAPRAM-HM3.0 this approach results in lower HCl but higher ClO formation.'

Line 317–319: "The evaluation simulations are carried out for 45 latitude with a relative humidity of 70 % under pristine ocean (Hoffmann et al., 2016) and polluted coastal conditions (Hoffmann et al., 2019a)."
In the previous section, several latitudes and relative humidities are modeled. Why are these not used in the evaluation of the reduced model? Can you provide an argument that these scenarios are sufficient to evaluate the reduced mechanism?

Within the analysis of the different simulations it turned out, that reactions that were not important under a specific condition, for example 15° latitude summer, are more important under another specific condition, for example 75° latitude winter, and vice versa. When such case occurred, the corresponding reaction is considered in the reduced mechanism to make it applicable to all regimes. Therefore, it is enough to perform only this meteorological setup. We have outlined this specific mechanism development process in more detail, which reads as follows:

Lines 176-177
'To provide a reduced mechanism applicable for a wide range of conditions, a chemical reaction is included in the reduced scheme in any case if the reaction is important under a single simulation condition.'

Lines 288-289
'Again, as for the CAPRAM-DM1.0 reduction, a chemical reaction is included in the reduced scheme in any case if the reaction is important under a single simulation condition.'

Lines 352-355
'The evaluation by these three simulation cases is appropriate, because both reduced mechanisms contain reactions that were both important and not important under the different performed simulations. Thus, other possible evaluation simulations would also treat reactions that are not necessary under specific conditions. Regardless of the simulation setup a similar performance is expected as a consequence.'

Line 378–379: "Consequently, the high CPU time required overlay the CPU time consumption from the reduction."
This should be clarified.

We have rephrased the sentence for clarity. It now reads:

Lines 408-409

'Therefore, the still high CPU time is caused by requirements of the standard multiphase chemistry mechanism. These high requirements cover the reduction of CPU time achieved by the reduction efforts.'

Line 560 Figure 8

These is an unusually square feature in Fig. 8b for aqueous-phase DMSO directly under the cloud. Is this a result of the way cloud grid-cells and aerosol grid-cells are treated in the model (Fig. 3)?

It is related to the treatment and the interpolation of the grid cells by the used graphic program (ncl). If the resolution of the grid cells would be increased, this feature might not be seen. However, this would increase our CPU time and result into a spatial resolution below 1 km that is normally not covered by regional CTMs.

Line 617–618: "Due to the advection of the stable MS- to the right-hand side of the model domain, the spatial DMSO profile is not modelled."
Is this not shown in Figure 8?

Yes, the DMSO profile is shown in Figure 8. Thus, we have linked on that figure. The corresponding sentence reads now as follows:

Lines 680-681
'Due to the advection of the stable MS$^-$ to the right-hand side of the model domain, the spatial profiles of DMSO (Fig. 8) and MS$^-$ differ.'

**References**

Hoffmann, E. H., Tilgner, A., Wolke, R., and Herrmann, H.: Enhanced chlorine and bromine atom activation by hydrolysis of halogen nitrates from marine aerosols at polluted coastal areas, Environ. Sci. Technol., 53, 771-778, https://doi.org/10.1021/acs.est.8b05165, 2019b.

Pechtl, S., and von Glasow, R.: Reactive chlorine in the marine boundary layer in the outflow of polluted continental air: A model study, Geophys. Res. Lett., 34, https://doi.org/10.1029/2007gl029761, 2007.

Sherwen, T., Schmidt, J. A., Evans, M. J., Carpenter, L. J., Großmann, K., Eastham, S. D., Jacob, D. J., Dix, B., Koenig, T. K., Sinreich, R., Ortega, I., Volkamer, R., Saiz-Lopez, A., Prados-Roman, C., Mahajan, A. S., and Ordóñez, C.: Global impacts of tropospheric halogens (Cl, Br, I) on oxidants and composition in GEOS-Chem, Atmos. Chem. Phys., 16, 12239-12271, https://doi.org/10.5194/acp-16-12239-2016, 2016.

Xue, L. K., Saunders, S. M., Wang, T., Gao, R., Wang, X. F., Zhang, Q. Z., and Wang, W. X.: Development of a chlorine chemistry module for the Master Chemical Mechanism, Geosci. Model Dev., 8, 3151-3162, https://doi.org/10.5194/gmd-8-3151-2015, 2015.

---

## Author Comment (AC3) · 15 Apr 2020

[revised manuscript text omitted]

16        Figures:     12
17        Tables:     11

19      ── rel. hum. = 50%  ── rel. hum. = 70% 1st day  ── rel. hum. = 70%  ── rel. hum. = 90%

**20 Figure S1 Modelled concentration time-profile of key compounds within the pristine marine boundary**
**21 layer for the summer simulations at 15° latitude. Red: simulation at rel. humidity of 50% (red).**
**22 Orange: simulation at relative humidity of 70% and cloud occurrence at early morning and**
**23 evening of the first model day. Dark green: simulation at relative humidity of 70% and cloud**
**24 occurrence at noon and midnight. Blue: simulation at relative humidity of 90% and cloud**
**25 occurrence at noon and midnight.**

[Figure]

**Figure S2**    **Modelled concentration time-profile of key compounds within the pristine marine boundary layer for the winter simulations at 15° latitude. Red: simulation at rel. humidity of 50% (red). Orange: simulation at relative humidity of 70% and cloud occurrence at early morning and evening of the first model day. Dark green: simulation at relative humidity of 70% and cloud occurrence at noon and midnight. Blue: simulation at relative humidity of 90% and cloud occurrence at noon and midnight.**

[Figure]

**Figure S3**    **Modelled concentration time-profile of key compounds within the pristine marine boundary layer for the summer simulations at 30° latitude. Red: simulation at rel. humidity of 50% (red). Orange: simulation at relative humidity of 70% and cloud occurrence at early morning and evening of the first model day. Dark green: simulation at relative humidity of 70% and cloud occurrence at noon and midnight. Blue: simulation at relative humidity of 90% and cloud occurrence at noon and midnight.**

[Figure]

**Figure S4**   **Modelled concentration time-profile of key compounds within the pristine marine boundary layer for the winter simulations at 30° latitude. Red: simulation at rel. humidity of 50% (red). Orange: simulation at relative humidity of 70% and cloud occurrence at early morning and evening of the first model day. Dark green: simulation at relative humidity of 70% and cloud occurrence at noon and midnight. Blue: simulation at relative humidity of 90% and cloud occurrence at noon and midnight.**

[Figure]

**Figure S5**    **Modelled concentration time-profile of key compounds within the pristine marine boundary**
**layer for the summer simulations at 45° latitude. Red: simulation at rel. humidity of 50% (red).**
**Orange: simulation at relative humidity of 70% and cloud occurrence at early morning and**
**evening of the first model day. Dark green: simulation at relative humidity of 70% and cloud**
**occurrence at noon and midnight. Blue: simulation at relative humidity of 90% and cloud**
**occurrence at noon and midnight.**

53
54
55
56

[Figure]

**Figure S6**   Modelled concentration time-profile of key compounds within the pristine marine boundary layer for the winter simulations at 45° latitude. Red: simulation at rel. humidity of 50% (red). Orange: simulation at relative humidity of 70% and cloud occurrence at early morning and evening of the first model day. Dark green: simulation at relative humidity of 70% and cloud occurrence at noon and midnight. Blue: simulation at relative humidity of 90% and cloud occurrence at noon and midnight.

[Figure]

**Figure S7    Modelled concentration time-profile of key compounds within the pristine marine boundary layer for the summer simulations at 60° latitude. Red: simulation at rel. humidity of 50% (red). Orange: simulation at relative humidity of 70% and cloud occurrence at early morning and evening of the first model day. Dark green: simulation at relative humidity of 70% and cloud occurrence at noon and midnight. Blue: simulation at relative humidity of 90% and cloud occurrence at noon and midnight.**

[Figure]

**Figure S8** **Modelled concentration time-profile of key compounds within the pristine marine boundary layer for the winter simulations at 60° latitude. Red: simulation at rel. humidity of 50% (red). Orange: simulation at relative humidity of 70% and cloud occurrence at early morning and evening of the first model day. Dark green: simulation at relative humidity of 70% and cloud occurrence at noon and midnight. Blue: simulation at relative humidity of 90% and cloud occurrence at noon and midnight.**

82

[Figure]

83

**Figure S9** **Modelled concentration time-profile of key compounds within the pristine marine boundary layer for the summer simulations at 75° latitude. Red: simulation at rel. humidity of 50% (red). Orange: simulation at relative humidity of 70% and cloud occurrence at early morning and evening of the first model day. Dark green: simulation at relative humidity of 70% and cloud occurrence at noon and midnight. Blue: simulation at relative humidity of 90% and cloud occurrence at noon and midnight.**

[Figure]

91

**Figure S10**   **Modelled concentration time-profile of key compounds within the pristine marine boundary**
**layer for the winter simulations at 75° latitude. Red: simulation at rel. humidity of 50% (red).**
**Orange: simulation at relative humidity of 70% and cloud occurrence at early morning and**
**evening of the first model day. Dark green: simulation at relative humidity of 70% and cloud**
**occurrence at noon and midnight. Blue: simulation at relative humidity of 90% and cloud**
**occurrence at noon and midnight.**

98

[Figure]

**Figure S11  Modelled formation rate of DMSO in (a) the gas phase and (c) the aqueous phase together with the modelled oxidation rate in (b) the gas phase and (d) the aqueous phase in the 'stable meteorological condition' simulation with stratiform clouds after 12 hours of modelling time. The x-axis represents the innermost horizontal grid cells divided by 100. The black contour lines represent the simulated clouds. The black line corresponds to a liquid water content of 0.01 g m$^{-3}$ and the white line to 0.1 g m$^{-3}$. The area framed by the white line includes LWC above 0.1 g m$^{-3}$.**

[Figure]

109
**Figure S12** **Simulated aqueous-phase concentration of bromide in the 'stable meteorological condition'**
110
**simulation with stratiform clouds after 12 hours of modelling time. The x-axis represents the**
111
**innermost horizontal grid cells divided by 100. The black contour lines represent the simulated**
112
**clouds. The black line corresponds to a liquid water content of 0.01 g m$^{-3}$ and the white line to 0.1**
113
**g m$^{-3}$. The area framed by the white line includes LWC above 0.1 g m$^{-3}$. The initial background**
114
**concentration is at about 16 ng m$^{-3}$.**
115

**Table S1** Implemented dry deposition, initial concentrations, and emission rates of chemical species for the open ocean simulation with COSMO-MUSCAT. Details on the dry deposition velocities are given in the previous CAPRAM studies Bräuer et al. (2013), Hoffmann et al. (2016) and Hoffmann et al. (2019a). Details on the initial concentrations and emission rates are given in Bräuer et al. (2013) and Hoffmann et al. (2016). In the term of $I_2$ and HOI, emission rates are derived from Prados-Roman et al. (2015). Aerosol initial concentrations are calculated from the SPACCIM simulations and were provided in the previous CAPRAM study Bräuer et al. (2013).

| Specie | Dry deposition / $s^{-1}$ | Initial concentration / molecules $cm^{-3}$ | Emission rates / mol $m^{-2}$ $s^{-1}$ | Aerosol initial concentration / mol $m^{-3}$ |
|---|---|---|---|---|
| $NH_3$ | $1.0\cdot10^{-2}$ | $1.28\cdot10^{9}$ | $7.589\cdot10^{-10}$ | |
| NO | $2.0\cdot10^{-4}$ | $2.50\cdot10^{8}$ | $4.151\cdot10^{-12}$ | |
| $NO_2$ | $2.0\cdot10^{-4}$ | $5.00\cdot10^{8}$ | | |
| $NO_3$ | $1.0\cdot10^{-2}$ | | | |
| $N_2O_5$ | $1.0\cdot10^{-2}$ | | | |
| HONO | | $2.50\cdot10^{8}$ | | |
| $HNO_3$ | $7.0\cdot10^{-3}$ | $2.00\cdot10^{9}$ | | |
| $HO_2NO_2$ | $5.0\cdot10^{-3}$ | | | |
| $O_3$ | $1.5\cdot10^{-3}$ | $7.50\cdot10^{11}$ | | |
| CO | $1.0\cdot10^{-3}$ | $4.25\cdot10^{12}$ | $1.416\cdot10^{-9}$ | |
| $CO_2$ | | $1.02\cdot10^{16}$ | | |
| $SO_2$ | $8.7\cdot10^{-3}$ | $2.55\cdot10^{9}$ | | |
| SULF | $1.0\cdot10^{-2}$ | | | |
| $H_2$ | | $1.28\cdot10^{13}$ | | |
| $H_2O_2$ | $5.0\cdot10^{-3}$ | $1.50\cdot10^{10}$ | | |
| $CH_4$ | | $4.50\cdot10^{13}$ | $2.923\cdot10^{-11}$ | |
| $C_2H_6$ | | $1.28\cdot10^{10}$ | $1.661\cdot10^{-13}$ | |
| $C_3H_8$ | | $2.31\cdot10^{10}$ | $3.321\cdot10^{-13}$ | |
| $C_2H_2$ | | $2.42\cdot10^{9}$ | $1.661\cdot10^{-13}$ | |
| $C_2H_4$ | | $2.55\cdot10^{9}$ | $3.985\cdot10^{-12}$ | |
| $C_3H_6$ | | | $1.661\cdot10^{-12}$ | |
| BIGENE | | $9.50\cdot10^{8}$ | | |
| HCHO | $5.0\cdot10^{-3}$ | $5.00\cdot10^{9}$ | $2.956\cdot10^{-14}$ | |
| $CH_3CHO$ | | $1.40\cdot10^{8}$ | $1.513\cdot10^{-10}$ | |
| $C_2H_5CHO$ | | $5.13\cdot10^{9}$ | $9.083\cdot10^{-11}$ | |
| HYAC | | $3.83\cdot10^{8}$ | $4.151\cdot10^{-12}$ | |

| Specie | Dry deposition / $s^{-1}$ | Initial concentration / molecules $cm^{-3}$ | Emission rates / mol $m^{-2}$ $s^{-1}$ | Aerosol initial concentration / mol $m^{-3}$ |
|---|---|---|---|---|
| $CH_3COCH_3$ | | $1.10 \cdot 10^{10}$ | $6.320 \cdot 10^{-12}$ | |
| MEK | | $6.89 \cdot 10^{8}$ | $7.124 \cdot 10^{-16}$ | |
| GLYOXAL | | $2.55 \cdot 10^{8}$ | | |
| $CH_3COCHO$ | | $2.55 \cdot 10^{8}$ | | |
| $CH_3OOH$ | $2.5 \cdot 10^{-3}$ | $5.00 \cdot 10^{9}$ | | |
| $CH_3CH_2OOH$ | | $2.55 \cdot 10^{9}$ | | |
| $CH_3COOOH$ | | $2.55 \cdot 10^{7}$ | | |
| PAN | $1.0 \cdot 10^{-4}$ | $2.50 \cdot 10^{8}$ | | |
| $CH_3OH$ | $1.0 \cdot 10^{-2}$ | $1.40 \cdot 10^{10}$ | $9.797 \cdot 10^{-16}$ | |
| $CH_3CH_2OH$ | $5.0 \cdot 10^{-3}$ | $2.00 \cdot 10^{9}$ | $1.015 \cdot 10^{-11}$ | |
| HCOOH | $1.0 \cdot 10^{-2}$ | $6.25 \cdot 10^{9}$ | | |
| $CH_3COOH$ | | $5.00 \cdot 10^{9}$ | $1.278 \cdot 10^{-12}$ | |
| $C_5H_8$ | | $1.28 \cdot 10^{9}$ | $2.341 \cdot 10^{-12}$ | |
| APIN | | $4.53 \cdot 10^{8}$ | $2.541 \cdot 10^{-14}$ | |
| BPIN | | $3.02 \cdot 10^{8}$ | | |
| $CHBr_3$ | | $3.83 \cdot 10^{7}$ | $2.225 \cdot 10^{-13}$ | |
| $C_3H_7I$ | | $1.63 \cdot 10^{7}$ | $8.170 \cdot 10^{-15}$ | |
| $CH_2I_2$ | | $2.55 \cdot 10^{5}$ | $1.876 \cdot 10^{-13}$ | |
| $CH_3I$ | | $2.04 \cdot 10^{7}$ | $2.458 \cdot 10^{-13}$ | |
| $CH_2ClI$ | | $2.55 \cdot 10^{5}$ | $1.524 \cdot 10^{-13}$ | |
| $CH_2BrI$ | | $8.93 \cdot 10^{5}$ | $8.751 \cdot 10^{-14}$ | |
| HCl | $2.0 \cdot 10^{-2}$ | $2.50 \cdot 10^{9}$ | | |
| HOCl | $2.0 \cdot 10^{-3}$ | | | |
| $ClNO_2$ | $1.0 \cdot 10^{-2}$ | | | |
| $ClNO_3$ | $1.0 \cdot 10^{-2}$ | | | |
| HBr | $2.0 \cdot 10^{-2}$ | | | |
| HOBr | $1.6 \cdot 10^{-3}$ | | | |
| $BrNO_2$ | $1.0 \cdot 10^{-2}$ | | | |
| $BrNO_3$ | $5.0 \cdot 10^{-3}$ | | | |
| $I_2$ | | | $1.744 \cdot 10^{-14}$ | |
| HOI | $1.0 \cdot 10^{-2}$ | | $3.321 \cdot 10^{-13}$ | |

| Specie | Dry deposition / $s^{-1}$ | Initial concentration / molecules $cm^{-3}$ | Emission rates / mol $m^{-2}$ $s^{-1}$ | Aerosol initial concentration / mol $m^{-3}$ |
|---|---|---|---|---|
| $INO_3$ | $1.0 \cdot 10^{-2}$ | | | |
| $I_2O_2$ | $1.0 \cdot 10^{-2}$ | | | |
| $I_2O_3$ | $1.0 \cdot 10^{-2}$ | | | |
| $I_2O_4$ | $1.0 \cdot 10^{-2}$ | | | |
| DMS | | $1.53 \cdot 10^9$ | $1.026 \cdot 10^{-10}$ | |
| DMSO | $5.0 \cdot 10^{-3}$ | | | |
| $DMSO_2$ | $5.0 \cdot 10^{-3}$ | | | |
| MSA | $5.0 \cdot 10^{-3}$ | | | |
| $SO_4^{2-}$ | | | | $1.05 \cdot 10^{-8}$ |
| $NO_3^-$ | | | | $2.05 \cdot 10^{-9}$ |
| $Cl^-$ | | | | $9.76 \cdot 10^{-8}$ |
| $Br^-$ | | | | $2.14 \cdot 10^{-10}$ |
| $NH_4^+$ | | | | $5.72 \cdot 10^{-9}$ |
| $Mn^{3+}$ | | | | $3.93 \cdot 10^{-15}$ |
| $Fe^{3+}$ | | | | $4.80 \cdot 10^{-15}$ |
| $Cu^{2+}$ | | | | $1.72 \cdot 10^{-13}$ |
| $HC_2O_4^-$ | | | | $3.94 \cdot 10^{-11}$ |
| MSA | | | | $3.26 \cdot 10^{-10}$ |
| $H^+$ | | | | $1.00 \cdot 10^{-11}$ |

**Table S2**     **Implemented gas-phase reactions in the CAPRAM-DM1.0red.**

| Nr. | Reaction | Rate constant[a] | Reference |
|---|---|---|---|
| D1 | DMS + OH → CH$_3$SCH$_2$O$_2$ - O$_2$ | k = 1.12·10$^{-11}$ exp(-250/T) | IUPAC, Atkinson et al. (2004) |
| D2 | DMS + OH → 0.9 DMSO + 0.9 HO$_2$ + 0.1 CH$_3$SOH + 0.1 CH$_3$O$_2$ − O$_2$ | **(1)** | see description at the table end |
| D3 | DMS + NO$_3$ → CH$_3$SCH$_2$O$_2$ - O$_2$ | k = 1.90·10$^{-13}$ exp(520/T) | IUPAC, Atkinson et al. (2004) |
| D4 | DMS + Cl → 0.82 CH$_3$SCH$_2$O$_2$ + 0.82 HCl + 0.18 DMSO + 0.18 ClO − O$_2$ | k = 1.88·10$^{-10}$ | IUPAC, Urbanski and Wine (1999) |
| D5 | DMS + ClO → 0.73 Cl + 0.73 DMSO + 0.27 HOCl + 0.27 CH$_3$SCH$_2$O$_2$ − 0.27 O$_2$ | k = 1.70·10$^{-15}$ exp(340/T) | IUPAC |
| D6 | DMS + BrO → DMSO + Br | k = 1.50·10$^{-14}$ exp(1000/T) | IUPAC |
| D7 | DMS + Cl$_2$ → CH$_3$SCH$_2$Cl + HCl | k = 3.40·10$^{-14}$ | Dyke et al. (2005) |
| D8 | DMS + IO → DMSO + I | k = 3.30·10$^{-13}$ exp(-925/T) | IUPAC |
| D9 | CH$_3$SCH$_2$O$_2$ + HO$_2$ → CH$_3$SCH$_2$OOH + O$_2$ | k = 1.13·10$^{-13}$ exp(1300/T) | MCMv3.2, Rickard et al. (21.10.2013) |
| D10 | CH$_3$SCH$_2$O$_2$ + NO → CH$_3$S + HCHO + NO$_2$ | k = 4.90·10$^{-12}$ exp(260/T) | MCMv3.2, Rickard et al. (21.10.2013) |
| D11 | CH$_3$SCH$_2$O$_2$ + NO$_3$ → CH$_3$S + HCHO + NO$_2$ + O$_2$ | k = 2.30·10$^{-12}$ | MCMv3.2, Rickard et al. (21.10.2013) |
| D12 | CH$_3$SCH$_2$O$_2$ + CH$_3$O$_2$ → 0.89 CH$_3$S + 0.89 HCHO + 0.11 CH$_3$SCHO + O$_2$ | k = 5.00·10$^{-13}$ exp(400/T) | In accordance to MCMv3.2 RO2 reaction |
| D13 | CH$_3$SCH$_2$Cl + OH → CH$_3$SOH + ClCH$_2$O$_2$ - O$_2$ | k = 2.50·10$^{-12}$ | Shallcross et al. (2006) |
| D14 | CH$_3$SCH$_2$OOH + OH → CH$_3$SCHO + OH + H$_2$O | k = 7.03·10$^{-11}$ | MCMv3.2, Rickard et al. (21.10.2013) |
| D15 | CH$_3$SCHO + OH → CH$_3$S + CO + H$_2$O | k = 1.11·10$^{-11}$ | MCMv3.2, Rickard et al. (21.10.2013) |
| D16 | DMSO + OH → MSIA + CH$_3$O$_2$ - O$_2$ | k = 6.10·10$^{-12}$ exp(800/T) | MCMv3.2, Rickard et al. (21.10.2013) |
| D17 | DMSO + NO$_3$ → DMSO$_2$ + NO$_2$ | k = 2.90·10$^{-13}$ | Sander et al. (2006) |
| D18 | DMSO + Cl → 0.43 DMSO$_2$ + 0.43 ClO + 0.57 CH$_3$SO + 0.57 HCHO + HCl - 0.43 O$_2$ | k = 1.45·10$^{-11}$ | Falbe-Hansen et al. (2000); Nicovich et al. (2006); Kleissas et al. (2007) |
| D19 | DMSO + BrO → CH$_3$SO$_2$CH$_3$ + Br | k = 1.00·10$^{-14}$ | Ballesteros et al. (2002) |
| D20 | CH$_3$SOH + OH → CH$_3$SO + H$_2$O | k = 5.00·10$^{-11}$ | Lucas and Prinn (2002a) |
| D21 | CH$_3$S + O$_3$ → CH$_3$SO + O$_2$ | k = 1.15·10$^{-12}$ exp(430/T) | MCMv3.2, Rickard et al. (21.10.2013) |
| D22 | CH$_3$S + O$_2$ → CH$_3$O$_2$ + SO$_2$ - O$_2$ | **(2)** | see description at the table end |
| D23 | CH$_3$S + O$_2$ → CH$_3$SO$_2$ | **(3)** | see description at the table end |
| D24 | MSIA + OH → CH$_3$O$_2$ + SO$_2$ + H$_2$O - O$_2$ | k = 9.00·10$^{-11}$ | MCMv3.2, Rickard et al. (21.10.2013) |
| D25 | CH$_3$SO + O$_3$ → CH$_3$O$_2$ + SO$_2$ | k = 4.00·10$^{-13}$ | MCMv3.2, Rickard et al. (21.10.2013) |
| D26 | CH$_3$SO$_2$ + O$_3$ → CH$_3$SO$_3$ + O$_2$ | k = 3.00·10$^{-13}$ | MCMv3.2, Rickard et al. (21.10.2013) |

| Nr. | Reaction | Rate constant[a] | Reference |
|---|---|---|---|
| D27 | $CH_3SO_2 \rightarrow CH_3O_2 + SO_2 - O_2$ | $k = 5.00 \cdot 10^{+13} \exp(-9673/T)$ | MCMv3.2, Rickard et al. (21.10.2013) |
| D28 | $CH_3SO_3 + HO_2 \rightarrow MSA + O_2$ | $k = 5.00 \cdot 10^{-11}$ | MCMv3.2, Rickard et al. (21.10.2013) |
| D29 | $CH_3SO_3 \rightarrow CH_3O_2 + SULF - H_2O - O_2$ | $k = 5.00 \cdot 10^{+13} \exp(-9946/T)$ | MCMv3.2, Rickard et al. (21.10.2013) |

**Photolysis reactions**

| | | | |
|---|---|---|---|
| D30 | $CH_3SCH_2OOH \rightarrow CH_3S + HCHO + OH$ | $J = 7.649 \cdot 10^{-06} \cos(\chi)^{0.682} \exp(-0.279/\cos(\chi))$ | MCMv3.2, Rickard et al. (21.10.2013) |
| D31 | $CH_3SCHO \rightarrow CH_3S + CO + HO_2 - O_2$ | $J = 2.792 \cdot 10^{-05} \cos(\chi)^{0.805} \exp(-0.338/\cos(\chi))$ | MCMv3.2, Rickard et al. (21.10.2013) |
| D32 | $CH_3SCH_2Cl \rightarrow CH_3S + ClCH_2O_2 - O_2$ | $J = 1.458 \cdot 10^{-04} \cos(\chi)^{0.314} \exp(-0.641/\cos(\chi))$ | Hoffmann et al. (2016) |

(a) $k^{2nd}$ in $cm^3$ molecules$^{-1}$ s$^{-1}$; $k^{1st}$ in s$^{-1}$; $J$ in s$^{-1}$;

(1) $k = \frac{k1 \times k3}{k2 + k3}$ with $k1 = \frac{9.5 \times 10^{-39} \times [O_2] \times e^{5270/T}}{1 + 7.5 \times 10^{-29} \times [O_2] \times e^{5610/T}}$; $k2 = \frac{2.05 \times 10^{-14} \times [O_2] \times e^{2674/T}}{\left(1 + 5.5 \times 10^{-31} \times [O_2] \times e^{7460/T}\right) \times T}$; rate calculated after Atkinson et al. (2004); Lucas and Prinn (2002b); Sander et al. (2006)

(2) $k = \frac{k1}{1 + k2}$ with $k1 = 1.92 \times 10^{-10} \times e^{-5730/T}$; $k2 = 1.60 \times 10^6 \times e^{-7310/T}$; rate calculated after MCMv3.2, Rickard et al. (21.10.2013)

(3) $k = \frac{k1}{1 + k2}$ with $k1 = 3.43 \times 10^{-27} \times e^{-5140/T}$; $k2 = 2.86 \times 10^{-11} \times e^{-3560/T}$; rate calculated after Campolongo et al. (1999); Turnipseed et al. (1993); (Rickard et al., 21.10.2013)

**Table S3    Implemented phase transfers in the CAPRAM-DM1.0red**

② reactions that run in the cloud mode 'sub#1', ③ reactions that run in the aerosol mode 'sub#2'

| Species | $K_{H\,(298\,K)}$[a] | $-\Delta H/R$[b] | Reference | $\alpha$ | Reference | $D_{g\,(298\,K)}$[c] | Reference |
|---|---|---|---|---|---|---|---|
| D33③ DMS | 0.56 | 4480 | Campolongo et al. (1999) | 0.001 | Zhu et al. (2006) | $1.08 \cdot 10^{-5}$ | Fuller et al. (1966) |
| D34③ DMSO | $1.00 \cdot 10^7$ | 2580 | Campolongo et al. (1999) | 0.1 | De Bruyn et al. (1994) | $1.01 \cdot 10^{-5}$ | Fuller et al. (1966) |
| D35② $DMSO_2$ | $1.00 \cdot 10^7$ | 5390 | Campolongo et al. (1999) | 0.1 | De Bruyn et al. (1994) | $9.55 \cdot 10^{-6}$ | Fuller et al. (1966) |
| D36② MSIA | $1.00 \cdot 10^8$ | 1760 | between $DMSO_2$ and MSA | 0.1 | as for MSAa | $1.11 \cdot 10^{-5}$ | Fuller et al. (1966) |
| D37② MSA | $5.09 \cdot 10^{13}$ | 1760 | Campolongo et al. (1999) | 0.1 | De Bruyn et al. (1994) | $1.04 \cdot 10^{-5}$ | Fuller et al. (1966) |

(a) in M atm$^{-1}$; (b) in K; (c) in $m^2$ s$^{-1}$

**Table S4    Implemented aqueous-phase reactions in the CAPRAM-DM1.0red**

② reactions that run in the cloud mode 'sub#1', ③ reactions that run in the aerosol mode 'sub#2'

| Nr. | Reaction | Rate constant[a] | Reference |
|------|----------|------------------|-----------|
| D38 | $DMS + O_3 \rightarrow DMSO + O_2$ | $k = 8.61 \cdot 10^{+08} exp(-2600/T)$ | Gershenzon et al. (2001) |
| D39 | $DMSO + OH \rightarrow MSIA + CH_3$ | $k = 6.65 \cdot 10^{+09} exp(-1270/T)$ | Zhu et al. (2003a) |
| D40③ | $DMSO + SO_4^- \rightarrow MSIA + CH_3 + H^+ + SO_4^{2-}$ | $k = 2.97 \cdot 10^{+09} exp(-1440/T)$ | Zhu et al. (2003b) |
| D41③ | $DMSO + Cl_2^- \rightarrow MSIA + HCl + CH_3 + Cl^- - H_2O$ | $k = 1.60 \cdot 10^{+07}$ | Zhu (2004) |
| D42② | $MSIA + O_3 \rightarrow MSA + O_2$ | $k = 3.50 \cdot 10^{+07}$ | Herrmann and Zellner (1997) |
| D43 | $MSI^- + OH \rightarrow CH_3 + 0.135\ SO_2 + 0.765\ MS^- + 0.765\ SO_3 - 0.765\ MSI^- + 0.9\ OH^- + 0.1\ HSO_3^-$ | $k = 1.20 \cdot 10^{+10}$ | Bardouki et al. (2002) |
| D44③ | $MSI^- + Cl_2^- \rightarrow CH_3 + 0.15\ SO_2 + 0.85\ MS^- + 0.85\ SO_3 - 0.85\ MSI^- + 2\ Cl^-$ | $k = 8.00 \cdot 10^{+08}$ | Zhu et al. (2005) |
| D45② | $MSI^- + O_3 \rightarrow CH_3SO_3^- + O_2$ | $k = 2.00 \cdot 10^{+06}$ | Flyunt et al. (2001) |
| D46 | $MS^- + OH \rightarrow HCHO + SO_3^- + H_2O - 0.5\ O_2$ | $k = 1.29 \cdot 10^{+07} exp(-2630/T)$ | Zhu et al. (2003a) |
| D47② | $MS^- + Cl_2^- \rightarrow CH_3 + SO_3 + 2\ Cl^-$ | $k = 3.89 \cdot 10^{+03}$ | Zhu (2004) |

(a) $k^{2nd}$ in $l^3\ mol^{-1}\ s^{-1}$

**Table S5    Implemented aqueous-phase equilibria in the CAPRAM-DM1.0red**

② reactions that run in the cloud mode 'sub#1', ③ reactions that run in the aerosol mode 'sub#2'

| | Equilibrium | K[a] | $k_{f, 298}$[b] | $E_A/R$[c] | $k_{b, 298}$[b] | $E_A/R$[c] | Reference |
|---|-------------|------|------------------|------------|------------------|------------|-----------|
| D48② | $MSIA \rightleftharpoons MSI^- + H^+$ | $5.0 \cdot 10^{-03}$ | $2.50 \cdot 10^{08}$ | | $5.00 \cdot 10^{10}$ | | Wudl et al. (1967) |
| D49② | $MSA \rightleftharpoons MS^- + H^+$ | 73 | $3.65 \cdot 10^{12}$ | | $5.00 \cdot 10^{10}$ | | Clarke and Woodward (1966) |

(a) in $M^{m-n}$, n order of reaction of forward reaction, m order of reaction of backward reaction; (b) $k_{298}^{2nd}$ in $l^1\ mol^{-1}\ s^{-1}$, $k_{298}^{1st}$ in $s^{-1}$; (c) in K

**Table S6**    **Implemented gas-phase reactions in the CAPRAM-HM3.0red**

| Nr. | Reaction | Rate constant[a] | Comment |
|---|---|---|---|
| H1 | $Cl + O_3 \rightarrow ClO$ | $k = 2.80\cdot10^{-11}exp(-250/T)$ | Atkinson et al. (2007) |
| H2 | $ClO + HO_2 \rightarrow HOCl$ | $k = 2.20\cdot10^{-12}exp(340/T)$ | Atkinson et al. (2007) |
| H3 | $HCl + OH \rightarrow Cl$ | $k = 1.70\cdot10^{-12}exp(-230/T)$ | Atkinson et al. (2007) |
| H4 | $ClO + NO \rightarrow Cl + NO_2$ | $k = 6.20\cdot10^{-12}exp(295/T)$ | Atkinson et al. (2007) |
| H5 | $Cl + NO_2 \rightarrow ClNO_2$ | TROE | Sander et al. (2006) |
| H6 | $ClO + NO_2 \rightarrow ClNO_3$ | TROE | Atkinson et al. (2007) |
| H7 | $ClNO_3 \rightarrow ClO + NO_2$ | $k = [M]*2.75\cdot10^{-6}exp(11438/T)$ | Anderson and Fahey (1990) |
| H8 | $Cl + CH_4 \rightarrow CH_3O_2 + HCl$ | $k = 6.60\cdot10^{-12}exp(-1240/T)$ | IUPAC, Atkinson et al. (2006) |
| H9 | $Cl + C_2H_6 \rightarrow C_2H_5O_2 + HCl$ | $k = 8.30\cdot10^{-11}exp(-100/T)$ | MCMv3.2, Rickard et al. (21.10.2013) |
| H10 | $Cl + C_3H_8 \rightarrow C_3H_7O_2 + HCl$ | $k = 1.40\cdot10^{-10}$ | IUPAC, Atkinson et al. (2006) |
| H11 | $Cl + BIGALKANE \rightarrow ALKO2 + HCl$ | $k = 2.05\cdot10^{-10}$ | IUPAC, Atkinson et al. (2006) |
| H12 | $Cl + CH_3OH \rightarrow HCHO + HO_2 + HCl$ | $k = 7.10\cdot10^{-11}exp(-75/T)$ | IUPAC, Atkinson et al. (2006) |
| H13 | $Cl + C_2H_5OH \rightarrow 0.92\ CH_3CHO + 0.92\ HO_2 + 0.08\ EO2 + HCl$ | $k = 6.05\cdot10^{-11}exp(155/T)$ | IUPAC, Atkinson et al. (2006) |
| H14 | $Cl + ALKOH \rightarrow 1.25\ MEK + HO_2 + HCl$ | $k = 2.70\cdot10^{-11}exp(525/T)$ | IUPAC, Atkinson et al. (2006) |
| H15 | $Cl + CH_3OOH \rightarrow HCl + 0.6\ CH_3O_2 + 0.4\ HCHO + 0.4\ OH$ | $k = 5.90\cdot10^{-11}$ | IUPAC, Atkinson et al. (2006) |
| H16 | $Cl + C_2H_5OOH \rightarrow HCl + CH_3CHO + OH$ | $k = 1.07\cdot10^{-10}$ | Wallington et al. (1989) |
| H17 | $ClO + CH_3O_2 \rightarrow Cl + O_2 + HCHO + HO_2$ | $k = 1.80\cdot10^{-11}exp(-600/T)$ | Burkholder et al. (2015) |
| H18 | $Cl + HCHO \rightarrow HCl + CO + HO_2$ | $k = 8.10\cdot10^{-11}exp(-34/T)$ | IUPAC, Atkinson et al. (2006) |
| H19 | $Cl + CH_3CHO \rightarrow HCl + CH_3CO_3$ | $k = 8.00\cdot10^{-11}$ | IUPAC, Atkinson et al. (2006) |
| H20 | $Cl + C_2H_5CHO \rightarrow HCl + 1.5\ CH_3CO_3$ | $k = 1.30\cdot10^{-10}$ | IUPAC, Atkinson et al. (2006) |
| H21 | $Cl + HYAC \rightarrow HCl + MGLY + HO_2$ | $k = 5.70\cdot10^{-11}$ | Orlando et al. (1999) |
| H22 | $Cl + CH_3COCHO \rightarrow HCl + CH_3CO_3 + CO$ | $k = 4.80\cdot10^{-11}$ | Green et al. (1990) |
| H23 | $Cl + GLYOXAL \rightarrow HCl + 2.0\ CO + HO_2$ | $k = 3.80\cdot10^{-11}$ | Niki et al. (1985) |
| H24 | $Cl + MEK \rightarrow HCl + MEKO2$ | $k = 3.05\cdot10^{-11}exp(80/T)$ | IUPAC, Atkinson et al. (2006) |
| H25 | $Cl + MACR \rightarrow 0.2\ MACRO2 + 0.8\ CC(O[O])(CCl)C=O + 0.2\ HCl$ | $k = 2.55\cdot10^{-10}$ | Rate constant average Canosa-Mas et al. (2001), Wang et al. (2002), Orlando et al. (2003) & Kaiser et al. (2010), |
| H26 | $CC(O[O])(CCl)C=O + HO_2 \rightarrow CH_3COCH_2Cl + CO + HO_2 + OH$ | $k = 1.00\cdot10^{-11}$ | Hasson et al. (2012) |
| H27 | $CC(O[O])(CCl)C=O + NO \rightarrow CH_3COCH_2Cl + CO + HO_2 + NO_2$ | $k = 1.17\cdot10^{-11}$ | Hsin and Elrod (2007) |

| Nr. | Reaction | Rate constant[a] | Comment |
|---|---|---|---|
| H28 | CC(O[O])(CCl)C=O + $CH_3O_2$ → $CH_3COCH_2Cl$ + CO + $HO_2$ + HCHO | k = $1.00 \cdot 10^{-12}$ | Hasson et al. (2012) |
| H29 | CC(O[O])(CCl)C=O + $CH_3CO_3$ → $CH_3COCH_2Cl$ + CO + $HO_2$ + $CH_3O_2$ | k = $1.00 \cdot 10^{-11}$ | estimated |
| H30 | OH + CC(OO)(CCl)C=O → $CH_3COCH_2Cl$ + CO + OH | k = $3.77 \cdot 10^{-11}$ | estimated |
| H31 | Cl + MVK → CC(=O)C(O[O])CCl | k = $2.10 \cdot 10^{-10}$ | Canosa-Mas et al. (2001) |
| H32 | CC(=O)C(O[O])CCl + $HO_2$ → CC(=O)C(OO)CCl | k = $1.82 \cdot 10^{-13}exp(1300/T)$ | MCMv3.2, Rickard et al. (21.10.2013) |
| H33 | CC(=O)C(O[O])CCl + NO → $ClCH_2CHO$ + $NO_2$ + $CH_3CO_3$ | k = $2.70 \cdot 10^{-12}exp(360/T)$ | MCMv3.2, Rickard et al. (21.10.2013) |
| H34 | CC(=O)C(O[O])CCl+ $NO_3$ → $ClCH_2CHO$ + $NO_2$ + $CH_3CO_3$ | k = $2.30 \cdot 10^{-12}$ | MCMv3.2, Rickard et al. (21.10.2013) |
| H35 | CC(=O)C(O[O])CCl + $CH_3O_2$ → $ClCH_2CHO$ + $CH_3CO_3$ + HCHO | k = $1.00 \cdot 10^{-12}$ | estimated |
| H36 | CC(=O)C(O[O])CCl + $CH_3CO_3$ → $ClCH_2CHO$ + $CH_3CO_3$ + $CH_3O_2$ | k = $1.00 \cdot 10^{-11}$ | estimated |
| H37 | OH + CC(=O)C(OO)CCl → $ClCH_2CHO$ + $CH_3CO_3$ + OH | k = $3.95 \cdot 10^{-11}$ | after MVKOOH in MCMv3.2, Rickard et al. (21.10.2013) |
| H38 | Cl + BIGALD1 → MALO2 + $HO_2$ + HCl | k = $1.35 \cdot 10^{-10}$ | Martín et al. (2013) |
| H39 | Cl + TOL → HCl + TOLO2 | k = $6.20 \cdot 10^{-11}$ | Wang et al. (2005) |
| H40 | Cl + XYL → HCl + XYLNO2 | k = $1.40 \cdot 10^{-10}$ | Wang et al. (2005) |
| H41 | Cl + BZALD → HCl + ACBZO2 | k = $1.00 \cdot 10^{-10}$ | Thiault et al. (2002) |
| H42 | Cl + GLYALD → HCl + $HOCH_2CO_3$ | k = $7.00 \cdot 10^{-11}$ | Niki et al. (1987) |
| H43 | Cl + $CH_3COCH_3$ → HCl + $CH_3COCH_2O_2$ | k = $3.20 \cdot 10^{-11}exp(-815/T)$ | Atkinson et al. (2006) |
| H44 | Cl + $C_2H_2$ → 0.26 ClCHO + 0.21 Cl + 0.53 HCl + 0.21 GLYOXAL + 1.32 CO + 0.79 $HO_2$ | TROE | Atkinson et al. (2006) |
| H45 | Cl + $C_2H_4$ → $ClCH_2CH_2O_2$ | TROE | Atkinson et al. (2006) |
| H46 | $ClCH_2CH_2O_2$ + $HO_2$ → $ClCH_2CH_2OOH$ | k = $3.30 \cdot 10^{-13}exp(820/T)$ | MCMv3.2, Rickard et al. (21.10.2013) |
| H47 | $ClCH_2CH_2O_2$ + NO → $ClCH_2CHO$ + $HO_2$ + $NO_2$ | k = $3.24 \cdot 10^{-12}exp(360/T)$ | Atkinson et al. (2008) |
| H48 | $ClCH_2CH_2O_2$ + $NO_3$ → $ClCH_2CHO$ + $HO_2$ + $NO_2$ | k = $2.30 \cdot 10^{-12}$ | MCMv3.2, Rickard et al. (21.10.2013) |
| H49 | $ClCH_2CH_2O_2$ + $CH_3O_2$ → $ClCH_2CHO$ + 0.8 HCHO + 0.2 $CH_3OH$ + 1.4 $HO_2$ | k = $2.00 \cdot 10^{-12}$ | estimated |
| H50 | $ClCH_2CHO$ + $NO_3$ → $ClCH_2CO_3$ + $HNO_3$ | k = $1.40 \cdot 10^{-12}exp(-1860/T)$ | MCMv3.2, Rickard et al. (21.10.2013) |
| H51 | $ClCH_2CHO$ + OH → $ClCH_2CO_3$ + $H_2O$ | k = $2.09 \cdot 10^{-11}$ | Atkinson et al. (2008) |
| H52 | $ClCH_2CO_3$ + $HO_2$ → 0.44 $ClCH_2O_2$ + 0.44 OH + 0.15 $ClCH_2COOH$ + 0.15 $O_3$ + 0.41 $ClCH_2C(O)OOH$ | k = $5.20 \cdot 10^{-13}exp(980/T)$ | MCMv3.2, Rickard et al. (21.10.2013) |
| H53 | $ClCH_2CO_3$ + NO → $ClCH_2O_2$ + $NO_2$ | k = $7.50 \cdot 10^{-12}exp(290/T)$ | MCMv3.2, Rickard et al. (21.10.2013) |
| H54 | $ClCH_2CO_3$ + $NO_2$ → ClPAN | TROE | MCMv3.2, Rickard et al. (21.10.2013) |
| H55 | $ClCH_2CO_3$ + $NO_3$ → $ClCH_2O_2$ + $NO_2$ | k = $4.00 \cdot 10^{-12}$ | MCMv3.2, Rickard et al. (21.10.2013) |

| Nr. | Reaction | Rate constant[a] | Comment |
|---|---|---|---|
| H56 | $ClCH_2CO_3 + CH_3O_2 \rightarrow 0.7\ ClCH_2O_2 + 0.3\ ClCH_2COOH + 0.7\ HO_2 + HCHO$ | $k = 1.00 \cdot 10^{-11}$ | estimated |
| H57 | $ClCH_2COOH + OH \rightarrow ClCH_2O_2$ | $k = 1.90 \cdot 10^{-12} exp(190/T)$ | MCMv3.2, Rickard et al. (21.10.2013) |
| H58 | $ClCH_2C(O)OOH + OH \rightarrow ClCH_2O_2$ | $k = 4.29 \cdot 10^{-12}$ | MCMv3.2, Rickard et al. (21.10.2013) |
| H59 | $ClPAN + OH \rightarrow ClCHO + CO + NO_2$ | $k = 6.26 \cdot 10^{-13}$ | MCMv3.2, Rickard et al. (21.10.2013) |
| H60 | $ClPAN \rightarrow ClCH_2CO_3 + NO_2$ | TROE | MCMv3.2, Rickard et al. (21.10.2013) |
| H61 | $ClCH_2O_2 + HO_2 \rightarrow 0.3\ ClCH_2OOH + 0.7\ ClCHO$ | $k = 3.20 \cdot 10^{-13} exp(820/T)$ | MCMv3.2, Rickard et al. (21.10.2013) |
| H62 | $ClCH_2O_2 + NO \rightarrow ClCHO + HO_2 + NO_2$ | $k = 4.05 \cdot 10^{-12} exp(360/T)$ | MCMv3.2, Rickard et al. (21.10.2013) |
| H63 | $ClCH_2O_2 + NO_3 \rightarrow ClCHO + HO_2 + NO_2$ | $k = 2.30 \cdot 10^{-12}$ | MCMv3.2, Rickard et al. (21.10.2013) |
| H64 | $ClCH_2O_2 + CH_3O_2 \rightarrow 1.4\ HO_2 + ClCHO + 0.8\ HCHO + 0.2\ CH_3OH$ | $k = 2.50 \cdot 10^{-12}$ | estimated |
| H65 | $Cl + C_3H_6 \rightarrow 0.4\ CH_3CH(O_2)CH_2Cl + 0.5\ CH_3CH(Cl)CH_2O_2 + 0.1\ HYAC$ | $k = 1.43 \cdot 10^{-14} exp(2886/T)$ | Atkinson et al. (2006) |
| H66 | $CH_3CH(O_2)CH_2Cl + NO \rightarrow CH_3COCH_2Cl + HO_2 + NO_2$ | $k = 2.70 \cdot 10^{-12} exp(360/T)$ | Atkinson et al. (2008) |
| H67 | $CH_3CH(Cl)CH_2O_2 + NO \rightarrow CH_3CH(Cl)CHO + NO_2 + HO_2$ | $k = 2.70 \cdot 10^{-12} exp(360/T)$ | MCMv3.2, Rickard et al. (21.10.2013) |
| H68 | $CH_3CH(O_2)CH_2Cl + CH_3O_2 \rightarrow CH_3COCH_2Cl + 0.8\ HCHO + 0.2\ CH_3OH + 1.4\ HO_2$ | $k = 4.00 \cdot 10^{-14}$ | estimated |
| H69 | $CH_3CH(Cl)CH_2O_2 + CH_3O_2 \rightarrow CH_3CH(Cl)CHO + 0.8\ HCHO + 0.2\ CH_3OH + 1.4\ HO_2$ | $k = 6.48 \cdot 10^{-13}$ | estimated |
| H70 | $CH_3COCH_2Cl + OH \rightarrow CH_3COCHClO_2$ | $k = 3.68 \cdot 10^{-13}$ | Atkinson et al. (2008) |
| H71 | $CH_3COCHClO_2 + HO_2 \rightarrow CH_3COCHClOOH$ | $k = 3.30 \cdot 10^{-13} exp(820/T)$ | MCMv3.2, Rickard et al. (21.10.2013) |
| H72 | $CH_3COCHClO_2 + NO \rightarrow ClCHO + CH_3CO_3 + NO_2$ | $k = 2.70 \cdot 10^{-12} exp(360/T)$ | Atkinson et al. (2008) |
| H73 | $CH_3COCHClO_2 + NO_3 \rightarrow ClCHO + CH_3CO_3 + NO_2$ | $k = 2.30 \cdot 10^{-12}$ | MCMv3.2, Rickard et al. (21.10.2013) |
| H74 | $CH_3COCHClO_2 + CH_3O_2 \rightarrow ClCHO + CH_3CO_3 + 0.8\ HCHO + 0.2\ CH_3OH + HO_2$ | $k = 2.00 \cdot 10^{-12}$ | estimated |
| H75 | $CH_3COCHClOOH + OH \rightarrow CH_3COCHClO_2$ | $k = 8.34 \cdot 10^{-12}$ | MCMv3.2, Rickard et al. (21.10.2013) |
| H76 | $ClCHO + NO_3 \rightarrow CO + Cl + HNO_3$ | $k = 1.40 \cdot 10^{-12} exp(-1860/T)$ | Atkinson et al. (2008) |
| H77 | $ClCHO + OH \rightarrow CO + Cl + H_2O$ | $k = 6.12 \cdot 10^{-12}$ | Atkinson et al. (2008) |
| H78 | $CH_3CH(Cl)CHO + OH \rightarrow CH_3CH(Cl)C(O)O_2$ | $k = 4.90 \cdot 10^{-12} exp(405/T)$ | MCMv3.2, Rickard et al. (21.10.2013) |
| H79 | $CH_3CH(Cl)CHO + NO_3 \rightarrow CH_3CH(Cl)C(O)O_2 + HNO_3$ | $k = 3.24 \cdot 10^{-12} exp(-1860/T)$ | MCMv3.2, Rickard et al. (21.10.2013) |
| H80 | $CH_3CH(Cl)C(O)O_2 + HO_2 \rightarrow 0.15\ CH_3CH(Cl)COOH + 0.15\ O_3 + 0.41\ CH_3CH(Cl)C(O)OOH + 0.44\ CH_3CH(Cl)O_2 + 0.44\ OH$ | $k = 5.20 \cdot 10^{-13} exp(980/T)$ | MCMv3.2, Rickard et al. (21.10.2013) |
| H81 | $CH_3CH(Cl)C(O)O_2 + NO \rightarrow CH_3CH(Cl)O_2 + NO_2$ | $k = 7.50 \cdot 10^{-12} exp(290/T)$ | MCMv3.2, Rickard et al. (21.10.2013) |
| H82 | $CH_3CH(Cl)CO_3 + NO_2 \rightarrow CH_3ClPAN$ | TROE | MCMv3.2, Rickard et al. (21.10.2013) |

| Nr. | Reaction | Rate constant[a] | Comment |
|---|---|---|---|
| H83 | $CH_3ClPAN \rightarrow CH_3CH(Cl)CO_3 + NO_2$ | TROE | MCMv3.2, Rickard et al. (21.10.2013) |
| H84 | $CH_3CH(Cl)C(O)O_2 + NO_3 \rightarrow CH_3CH(Cl)O_2 + NO_2$ | $k = 4.00 \cdot 10^{-12}$ | MCMv3.2, Rickard et al. (21.10.2013) |
| H85 | $CH_3CH(Cl)C(O)O_2 + CH_3O_2 \rightarrow 0.3\ CH_3CH(Cl)COOH + 0.7\ CH_3CH(Cl)O_2 +$ $HCHO + HO_2$ | $k = 1.00 \cdot 10^{-11}$ | estimated |
| H86 | $CC(Cl)C(=O)OO + OH \rightarrow CC(Cl)C(=O)O[O]$ | $k = 4.42 \cdot 10^{-12}$ | MCMv3.2, Rickard et al. (21.10.2013) |
| H87 | $CH_3CH(Cl)COOH + OH \rightarrow CH_3CH(Cl)O_2$ | $k = 1.20 \cdot 10^{-12}$ | MCMv3.2, Rickard et al. (21.10.2013) |
| H88 | $CH_3CH(Cl)O_2 + HO_2 \rightarrow CH_3CH(Cl)OOH$ | $k = 3.30 \cdot 10^{-13} exp(820/T)$ | MCMv3.2, Rickard et al. (21.10.2013) |
| H89 | $CH_3CH(Cl)O_2 + NO \rightarrow CH_3CHO + Cl + NO_2$ | $k = 4.05 \cdot 10^{-12} exp(360/T)$ | MCMv3.2, Rickard et al. (21.10.2013) |
| H90 | $CH_3CH(Cl)O_2 + NO_3 \rightarrow CH_3CHO + Cl + NO_2$ | $k = 2.30 \cdot 10^{-12}$ | MCMv3.2, Rickard et al. (21.10.2013) |
| H91 | $CH_3CH(Cl)O_2 + CH_3O_2 \rightarrow 0.6\ CH_3CHO + 0.6\ Cl + 0.4\ CH_3C(O)Cl + 0.8\ HCHO$ $+ 0.2\ CH_3OH + 0.8\ HO_2$ | $k = 2.65 \cdot 10^{-12}$ | estimated |
| H92 | $CH_3CH(Cl)OOH + OH \rightarrow CH_3CH(Cl)O_2 + H_2O$ | $k = 1.90 \cdot 10^{-12} exp(190/T)$ | MCMv3.2, Rickard et al. (21.10.2013) |
| H93 | $CH_3CH(Cl)OOH + OH \rightarrow CH_3C(O)Cl + OH + H_2O$ | $k = 9.95 \cdot 10^{-12}$ | MCMv3.2, Rickard et al. (21.10.2013) |
| H94 | $CH_3C(O)Cl + OH \rightarrow ClCOCH_2O_2 + H_2O$ | $k = 3.88 \cdot 10^{-14}$ | MCMv3.2, Rickard et al. (21.10.2013) |
| H95 | $ClCOCH_2O_2 + HO_2 \rightarrow ClCOCH_2OOH$ | $k = 3.30 \cdot 10^{-13} exp(820/T)$ | MCMv3.2, Rickard et al. (21.10.2013) |
| H96 | $ClCOCH_2O_2 + NO \rightarrow HCHO + Cl + CO + NO_2$ | $k = 3.24 \cdot 10^{-12} exp(360/T)$ | MCMv3.2, Rickard et al. (21.10.2013) |
| H97 | $ClCOCH_2O_2 + NO_3 \rightarrow HCHO + Cl + CO + NO_2$ | $k = 2.30 \cdot 10^{-12}$ | MCMv3.2, Rickard et al. (21.10.2013) |
| H98 | $ClCOCH_2O_2 + CH_3O_2 \rightarrow 2\ HCHO + Cl + CO + HO_2$ | $k = 2.00 \cdot 10^{-12}$ | MCMv3.2, Rickard et al. (21.10.2013) |
| H99 | $Br + O_3 \rightarrow BrO$ | $k = 1.70 \cdot 10^{-11} exp(-800/T)$ | Atkinson et al. (2007) |
| H100 | $BrO + HO_2 \rightarrow HOBr$ | $k = 4.50 \cdot 10^{-12} exp(-500/T)$ | Atkinson et al. (2007) |
| H101 | $BrO + BrO \rightarrow 1.7\ Br + 0.15\ Br_2$ | $k = 1.60 \cdot 10^{-12} exp(-210/T)$ | Atkinson et al. (2007) |
| H102 | $Br + NO_2 \rightarrow BrNO_2$ | TROE | Atkinson et al. (2007) |
| H103 | $BrO + NO \rightarrow Br + NO_2$ | $k = 8.70 \cdot 10^{-12} exp(-260/T)$ | Atkinson et al. (2007) |
| H104 | $BrO + NO_2 \rightarrow BrNO_3$ | TROE | Atkinson et al. (2007) |
| H105 | $BrNO_3 \rightarrow BrO + NO_2$ | $k = 2.79 \cdot 10^{13} exp(-12360/T)$ | Orlando and Tyndall (1996) |
| H106 | $Br + BrNO_3 \rightarrow Br_2 + NO_3$ | $k = 4.90 \cdot 10^{-11}$ | Orlando and Tyndall (1996) |
| H107 | $BrO + ClO \rightarrow 0.95\ Br + 0.5\ OClO + 0.45\ Cl + 0.05\ BrCl$ | $k = 7.32 \cdot 10^{-12} exp(-200/T)$ | Summation A-Factor Burkholder et al. (2015) |
| H108 | $BrO + CH_3O_2 \rightarrow$ $0.25\ Br + 0.25\ HCHO + 0.25\ HO_2 + 0.75\ HOBr + 0.75\ HCOOH$ | $k = 4.10 \cdot 10^{-13} exp(-800/T)$ | Bräuer et al. (2013) |
| H109 | $Br + C_2H_2 \rightarrow$ $0.17\ BrCHO + 0.09\ Br + 0.74\ HBr + 0.09\ GLYOXAL + 1.65\ CO + 0.91\ HO_2$ | $k = 6.35 \cdot 10^{-15} exp(-440/T)$ | Atkinson et al. (2006) |

| Nr. | Reaction | Rate constant[a] | Comment |
|---|---|---|---|
| H110 | $Br + HCHO \rightarrow HBr + CO + HO_2$ | $k = 1.70 \cdot 10^{-11} exp(-800/T)$ | Sander et al. (2006) |
| H111 | $BrO + HCHO \rightarrow HOBr + CO + HO_2$ | $k = 1.50 \cdot 10^{-14}$ | Hansen et al. (1999) |
| H112 | $Br + CH_3CHO \rightarrow HBr + CH_3CO_3$ | $k = 1.80 \cdot 10^{-11} exp(-460/T)$ | Atkinson et al. (2006) |
| H113 | $Br + C_2H_5CHO \rightarrow HBr + 1.5\ CH_3CO_3$ | $k = 5.75 \cdot 10^{-11} exp(-610/T)$ | Ramacher et al. (2000) |
| H114 | $Br + C_2H_4 \rightarrow BrCH_2CH_2O_2$ | $k = 2.25 \cdot 10^{-13} exp(-277/T)$ | Atkinson et al. (2006) |
| H115 | $BrCH_2CH_2O_2 + NO \rightarrow BrCH_2CHO + HO_2 + NO_2$ | $k = 9.70 \cdot 10^{-12}$ | Atkinson et al. (2008) |
| H116 | $BrCH_2CH_2O_2 + CH_3O_2 \rightarrow BrCH_2CHO + 0.8\ HCHO + 0.2\ CH_3OH + 1.4\ HO_2$ | $k = 2.00 \cdot 10^{-12}$ | Bräuer et al. (2013) |
| H117 | $BrCH_2CHO + OH \rightarrow BrCH_2CO_3 + H_2O$ | $k = 2.05 \cdot 10^{-12}$ | Atkinson et al. (2008) |
| H118 | $BrCH_2CO_3 + HO_2 \rightarrow$ $0.15\ BrCH_2COOH + 0.15\ O_3 + 0.41\ BrCH_2C(O)OOH + 0.44\ BrCH_2O_2 + 0.44\ OH$ | $k = 5.20 \cdot 10^{-13} exp(980/T)$ | MCMv3.2, Rickard et al. (21.10.2013) |
| H119 | $BrCH_2CO_3 + NO \rightarrow BrCH_2O_2 + NO_2$ | $k = 7.50 \cdot 10^{-12} exp(290/T)$ | MCMv3.2, Rickard et al. (21.10.2013) |
| H120 | $BrCH_2CO_3 + CH_3O_2 \rightarrow 0.7\ BrCH_2O_2 + 0.3\ BrCH_2COOH + 0.7\ HO_2 + HCHO$ | $k = 1.00 \cdot 10^{-11}$ | Bräuer et al. (2013) |
| H121 | $BrCH_2COOH + OH \rightarrow BrCH_2O_2 + H_2O$ | $k = 1.90 \cdot 10^{-12} exp(190/T)$ | MCMv3.2, Rickard et al. (21.10.2013) |
| H122 | $BrCH_2C(O)OOH + OH \rightarrow BrCH_2CO_3 + H_2O$ | $k = 3.79 \cdot 10^{-12}$ | MCMv3.2, Rickard et al. (21.10.2013) |
| H123 | $BrCH_2O_2 + HO_2 \rightarrow BrCH_2OOH$ | $k = 4.28 \cdot 10^{-13} exp(820/T)$ | MCMv3.2, Rickard et al. (21.10.2013) |
| H124 | $BrCH_2O_2 + NO \rightarrow BrCHO + HO_2 + NO_2$ | $k = 4.05 \cdot 10^{-12} exp(360/T)$ | MCMv3.2, Rickard et al. (21.10.2013) |
| H125 | $BrCH_2O_2 + NO_3 \rightarrow BrCHO + HO_2 + NO_2$ | $k = 2.30 \cdot 10^{-12}$ | MCMv3.2, Rickard et al. (21.10.2013) |
| H126 | $BrCH_2O_2 + CH_3O_2 \rightarrow 1.4\ HO_2 + BrCHO + 0.8\ HCHO + 0.2\ CH_3OH$ | $k = 2.00 \cdot 10^{-12}$ | Bräuer et al. (2013) |
| H127 | $BrCH_2OOH + OH \rightarrow BrCH_2O_2 + H_2O$ | $k = 1.90 \cdot 10^{-12} exp(190/T)$ | MCMv3.2, Rickard et al. (21.10.2013) |
| H128 | $BrCH_2OOH + OH \rightarrow BrCHO + OH + H_2O$ | $k = 5.79 \cdot 10^{-12}$ | MCMv3.2, Rickard et al. (21.10.2013) |
| H129 | $BrCHO + NO_3 \rightarrow CO + Br + HNO_3$ | $k = 1.40 \cdot 10^{-12} exp(-1860/T)$ | Atkinson et al. (2008) |
| H130 | $BrCHO + OH \rightarrow CO + Br + H_2O$ | $k = 1.16 \cdot 10^{-12}$ | Atkinson et al. (2008) |
| H131 | $Br + C_3H_6 \rightarrow CH_3CH(O_2)CH_2Br$ | $k = 3.60 \cdot 10^{-12}$ | Atkinson et al. (2006) |
| H132 | $CH_3CH(O_2)CH_2Br + NO \rightarrow CH_3COCH_2Br + HO_2 + NO_2$ | $k = 2.70 \cdot 10^{-12} exp(360/T)$ | Atkinson et al. (2008) |
| H133 | $CH_3CH(O_2)CH_2Br + CH_3O_2 \rightarrow CH_3COCH_2Br + 0.8\ HCHO + 0.2\ CH_3OH + 1.4\ HO_2$ | $k = 4.00 \cdot 10^{-14}$ | Bräuer et al. (2013) |
| H134 | $CH_3COCH_2Br + OH \rightarrow CH_3COCHBrO_2$ | $k = 8.80 \cdot 10^{-12} exp(-1320/T)$ | Atkinson et al. (2008) |
| H135 | $CH_3COCHBrO_2 + NO \rightarrow CH_3CO_3 + BrCHO + NO_2$ | $k = 8.00 \cdot 10^{-12}$ | Atkinson et al. (2008) |
| H136 | $CH_3COCHBrO_2 + CH_3O_2 \rightarrow 0.4\ CH_3COC(O)Br + 0.6\ CH_3CO_3 + 0.6\ BrCHO + 0.8\ HO_2 + 0.8\ HCHO + 0.2\ CH_3OH$ | $k = 2.00 \cdot 10^{-12}$ | Bräuer et al. (2013) |

| Nr. | Reaction | Rate constant[a] | Comment |
|---|---|---|---|
| H137 | $I + O_3 \rightarrow IO$ | $k = 2.10 \cdot 10^{-11} \exp(-830/T)$ | Atkinson et al. (2007) |
| H138 | $I_2 + OH \rightarrow I + HOI$ | $k = 2.10 \cdot 10^{-10}$ | Atkinson et al. (2007) |
| H139 | $IO + HO_2 \rightarrow HOI$ | $k = 1.40 \cdot 10^{-11} \exp(540/T)$ | Atkinson et al. (2007) |
| H140 | $IO + IO \rightarrow 0.38\ OIO + 0.46\ I_2O_2 + 0.6\ I + 0.05\ I_2$ | $k = 5.40 \cdot 10^{-11} \exp(180/T)$ | Sander et al. (2006) |
| H141 | $OIO + OH \rightarrow HIO_3$ | $k = 2.20 \cdot 10^{-10} \exp(243/T)$ | von Glasow et al. (2002) |
| H142 | $IO + O_3 \rightarrow 0.83\ I + 0.17\ OIO$ | $k = 1.20 \cdot 10^{-15}$ | (Larin et al., 1999) |
| H143 | $IO + OIO \rightarrow I_2O_3$ | $k = 1.00 \cdot 10^{-10}$ | (Gómez Martín et al., 2007) |
| H144 | $I_2O_3 \rightarrow IO + OIO$ | $k = 2.78 \cdot 10^{-11}$ | (Kaltsoyannis and Plane, 2008) |
| H145 | $OIO + OIO \rightarrow I_2O_4$ | $k = 1.00 \cdot 10^{-10}$ | (Saunders and Plane, 2005) |
| H146 | $I_2O_4 \rightarrow OIO + OIO$ | $k = 1.67 \cdot 10^{+00}$ | (Kaltsoyannis and Plane, 2008) |
| H147 | $I_2 + O_3 \rightarrow IO + I$ | $k = 4.02 \cdot 10^{-15} \exp(-2050/T)$ | (Vikis and Macfarlane, 1985) |
| H148 | $I_2O_2 \rightarrow 0.995\ OIO + 0.995\ I + 0.01\ IO$ | $k = 1.00 \cdot 10^{+01}$ | (Kaltsoyannis and Plane, 2008) |
| H149 | $I_2 + NO_3 \rightarrow I + INO_3$ | $k = 1.50 \cdot 10^{-12}$ | Atkinson et al. (2007) |
| H150 | $IO + NO \rightarrow I + NO_2$ | $k = 7.15 \cdot 10^{-12} \exp(300/T)$ | Atkinson et al. (2007) |
| H151 | $IO + NO_2 \rightarrow INO_3$ | TROE | Atkinson et al. (2007) |
| H152 | $INO_3 \rightarrow IO + NO_2$ | $k = [M]*4.40 \cdot 10^{-05} \exp(12060/T)$ | Atkinson et al. (2007) |
| H153 | $IO + CH_3O_2 \rightarrow I + HO_2 + HCHO$ | $k = 2.00 \cdot 10^{-12}$ | (Dillon et al., 2006) |
| H154 | $IO + ClO \rightarrow 0.8\ I + 0.55\ OClO + 0.25\ Cl + 0.2\ ICl$ | $k = 4.70 \cdot 10^{-12} \exp(280/T)$ | Atkinson et al. (2007) |
| H155 | $IO + BrO \rightarrow 0.8\ OIO + Br + 0.2\ I$ | $k = 1.50 \cdot 10^{-11} \exp(510/T)$ | Atkinson et al. (2007) |

**Photolysis reactions**

| Nr. | Reaction | Rate constant[a] | Comment |
|---|---|---|---|
| H156 | $Cl_2 \rightarrow Cl + Cl$ | $J = 3.827 \cdot 10^{-03} \cos(\chi)^{0.543} \exp(-0.244/\cos(\chi))$ | Bräuer et al. (2013) |
| H157 | $ClO \rightarrow Cl + O(^3P)$ | $J = 4.755 \cdot 10^{-04} \cos(\chi)^{1.258} \exp(-0.588/\cos(\chi))$ | Bräuer et al. (2013) |
| H158 | $OClO \rightarrow ClO + O(^3P)$ | $J = 1.332 \cdot 10^{-01} \cos(\chi)^{0.416} \exp(-0.244/\cos(\chi))$ | Bräuer et al. (2013) |
| H159 | $HOCl \rightarrow Cl + OH$ | $J = 4.615 \cdot 10^{-04} \cos(\chi)^{0.656} \exp(-0.240/\cos(\chi))$ | Bräuer et al. (2013) |
| H160 | $ClNO_2 \rightarrow Cl + NO_2$ | $J = 6.219 \cdot 10^{-04} \cos(\chi)^{0.774} \exp(-0.255/\cos(\chi))$ | Bräuer et al. (2013) |
| H161 | $ClNO_3 \rightarrow Cl + NO_3$ | $J = 6.420 \cdot 10^{-05} \cos(\chi)^{0.648} \exp(-0.217/\cos(\chi))$ | Bräuer et al. (2013) |
| H162 | $ClNO_3 \rightarrow ClO + NO_2$ | $J = 1.393 \cdot 10^{-05} \cos(\chi)^{1.052} \exp(-0.243/\cos(\chi))$ | Bräuer et al. (2013) |
| H163 | $CC(=O)C(OO)CCl \rightarrow ClCH_2CHO + CH_3CO_3 + OH$ | $J = 7.649 \cdot 10^{-05} \cos(\chi)^{0.682} \exp(-0.279/\cos(\chi))$ | Bräuer et al. (2013) |
| H164 | $ClCH_2CH_2OOH \rightarrow ClCH_2CHO + HO_2 + OH$ | $J = 7.649 \cdot 10^{-06} \cos(\chi)^{0.682} \exp(-0.279/\cos(\chi))$ | Bräuer et al. (2013) |
| H165 | $ClCH_2CHO \rightarrow ClCH_2O_2 + HO_2 + CO$ | $J = 4.642 \cdot 10^{-05} \cos(\chi)^{0.762} \exp(-0.353/\cos(\chi))$ | Bräuer et al. (2013) |

| Nr. | Reaction | Rate constant[a] | Comment |
|---|---|---|---|
| H166 | $ClCH_2C(O)OOH \rightarrow ClCH_2O_2 + OH$ | $J = 7.649 \cdot 10^{-06}\cos(\chi)^{0.682}\exp(-0.279/\cos(\chi))$ | Bräuer et al. (2013) |
| H167 | $ClCH_2OOH \rightarrow ClCHO + HO_2 + OH$ | $J = 7.649 \cdot 10^{-06}\cos(\chi)^{0.682}\exp(-0.279/\cos(\chi))$ | Bräuer et al. (2013) |
| H168 | $CH_3CH(O)CH_2Cl \rightarrow CH_3O_2 + ClCH_2CO_3$ | $J = 5.804 \cdot 10^{-06}\cos(\chi)^{1.092}\exp(-0.377/\cos(\chi))$ | Bräuer et al. (2013) |
| H169 | $CH_3CH(O)CHClOOH \rightarrow ClCHO + CH_3CO_3 + OH$ | $J = 7.649 \cdot 10^{-06}\cos(\chi)^{0.682}\exp(-0.279/\cos(\chi))$ | Bräuer et al. (2013) |
| H170 | $ClCHO \rightarrow HO_2 + CO + Cl$ | $J = 4.642 \cdot 10^{-05}\cos(\chi)^{0.762}\exp(-0.353/\cos(\chi))$ | Bräuer et al. (2013) |
| H171 | $CH_3CH(Cl)CHO \rightarrow CH_3CH(Cl)O_2 + HO_2 + CO$ | $J = 2.879 \cdot 10^{-05}\cos(\chi)^{1.067}\exp(-0.358/\cos(\chi))$ | Bräuer et al. (2013) |
| H172 | $CH_3CH(Cl)OOH \rightarrow CH_3CHO + Cl + OH$ | $J = 7.649 \cdot 10^{-06}\cos(\chi)^{0.682}\exp(-0.279/\cos(\chi))$ | Bräuer et al. (2013) |
| H173 | $CH_3C(O)Cl \rightarrow CH_3CO_3 + Cl$ | $J = 5.804 \cdot 10^{-06}\cos(\chi)^{1.092}\exp(-0.377/\cos(\chi))$ | Bräuer et al. (2013) |
| H174 | $ClCOCH_2OOH \rightarrow ClCOCH_2O_2 + OH$ | $J = 7.649 \cdot 10^{-06}\cos(\chi)^{0.682}\exp(-0.279/\cos(\chi))$ | Bräuer et al. (2013) |
| H175 | $Br_2 \rightarrow Br + Br$ | $J = 4.773 \cdot 10^{-02}\cos(\chi)^{0.193}\exp(-0.213/\cos(\chi))$ | Bräuer et al. (2013) |
| H176 | $BrO \rightarrow Br + O(^3P)$ | $J = 6.368 \cdot 10^{-02}\cos(\chi)^{0.605}\exp(-0.269/\cos(\chi))$ | Bräuer et al. (2013) |
| H177 | $HOBr \rightarrow Br + OH$ | $J = 3.464 \cdot 10^{-03}\cos(\chi)^{0.441}\exp(-0.214/\cos(\chi))$ | Bräuer et al. (2013) |
| H178 | $BrNO_2 \rightarrow Br + NO_2$ | $J = 7.443 \cdot 10^{-03}\cos(\chi)^{0.355}\exp(-0.236/\cos(\chi))$ | Bräuer et al. (2013) |
| H179 | $BrNO_3 \rightarrow 0.29\ Br + 0.29\ NO_3 + 0.71\ BrO + 0.71\ NO_2$ | $J = 2.194 \cdot 10^{-04}\cos(\chi)^{0.492}\exp(-0.215/\cos(\chi))$ | Bräuer et al. (2013) |
| H180 | $BrCl \rightarrow Br + Cl$ | $J = 1.650 \cdot 10^{-02}\cos(\chi)^{0.297}\exp(-0.224/\cos(\chi))$ | Bräuer et al. (2013) |
| H181 | $BrCH_2CHO \rightarrow BrCH_2O_2 + HO_2 + CO$ | $J = 4.642 \cdot 10^{-05}\cos(\chi)^{0.762}\exp(-0.353/\cos(\chi))$ | Bräuer et al. (2013) |
| H182 | $BrCH_2C(O)OOH \rightarrow BrCH_2O_2 + OH$ | $J = 7.649 \cdot 10^{-06}\cos(\chi)^{0.682}\exp(-0.279/\cos(\chi))$ | Bräuer et al. (2013) |
| H183 | $BrCH_2OOH \rightarrow BrCHO + OH + HO_2$ | $J = 7.649 \cdot 10^{-06}\cos(\chi)^{0.682}\exp(-0.279/\cos(\chi))$ | Bräuer et al. (2013) |
| H184 | $BrCHO \rightarrow HO_2 + CO + Br$ | $J = 4.642 \cdot 10^{-05}\cos(\chi)^{0.762}\exp(-0.353/\cos(\chi))$ | Bräuer et al. (2013) |
| H185 | $CH_3COCH_2Br \rightarrow 0.7\ CO + 0.7\ Br + 0.7\ CH_3CO_3 + 0.3\ BrCH_2CO_3 + 0.3\ CH_3O_2$ | $J = 3.523 \cdot 10^{-04}\cos(\chi)^{0.885}\exp(-0.283/\cos(\chi))$ | Bräuer et al. (2013) |
| H186 | $CH_3COC(O)Br \rightarrow CO + Br + CH_3CO_3$ | $J = 1.853 \cdot 10^{-04}\cos(\chi)^{0.583}\exp(-0.225/\cos(\chi))$ | Bräuer et al. (2013) |
| H187 | $CHBr_3 \rightarrow 3\ Br + CO + HO_2$ | $J = 2.228 \cdot 10^{-06}\cos(\chi)^{1.471}\exp(-0.230/\cos(\chi))$ | Bräuer et al. (2013) |
| H188 | $I_2 \rightarrow I + I$ | $J = 2.165 \cdot 10^{-01}\cos(\chi)^{0.125}\exp(-0.185/\cos(\chi))$ | Bräuer et al. (2013) |
| H189 | $IO \rightarrow I + O(^3P)$ | $J = 2.640 \cdot 10^{-03}\cos(\chi)^{0.240}\exp(-0.240/\cos(\chi))$ | Bräuer et al. (2013) |
| H190 | $OIO \rightarrow 0.96\ I + 0.04\ IO + 0.04\ O(^3P)$ | $J = 4.054 \cdot 10^{-02}\cos(\chi)^{0.119}\exp(-0.185/\cos(\chi))$ | Bräuer et al. (2013) |
| H191 | $HOI \rightarrow I + OH$ | $J = 1.469 \cdot 10^{-02}\cos(\chi)^{0.342}\exp(-0.236/\cos(\chi))$ | Bräuer et al. (2013) |
| H192 | $INO_3 \rightarrow 0.85\ I + 0.85\ NO_3 + 0.15\ IO + 0.15\ NO_2$ | $J = 6.599 \cdot 10^{-02}\cos(\chi)^{0.530}\exp(-0.243/\cos(\chi))$ | Bräuer et al. (2013) |
| H193 | $ICl \rightarrow I + Cl$ | $J = 3.403 \cdot 10^{-02}\cos(\chi)^{0.179}\exp(-0.207/\cos(\chi))$ | Bräuer et al. (2013) |
| H194 | $IBr \rightarrow I + Br$ | $J = 1.000 \cdot 10^{-01}\cos(\chi)^{0.149}\exp(-0.197/\cos(\chi))$ | Bräuer et al. (2013) |
| H195 | $C_3H_7I \rightarrow I + C_3H_7O_2$ | $J = 3.731 \cdot 10^{-05}\cos(\chi)^{1.292}\exp(-0.217/\cos(\chi))$ | Bräuer et al. (2013) |
| H196 | $CH_2I_2 \rightarrow 2\ I + 2\ HO_2$ | $J = 1.496 \cdot 10^{-02}\cos(\chi)^{0.801}\exp(-0.265/\cos(\chi))$ | Bräuer et al. (2013) |

| Nr. | Reaction | Rate constant[a] | Comment |
|---|---|---|---|
| H197 | $CH_3I \rightarrow I + CH_3O_2$ | $J = 1.206 \cdot 10^{-05} \cos(\chi)^{1.254} \exp(-0.231/\cos(\chi))$ | Bräuer et al. (2013) |
| H198 | $ClCH_2I \rightarrow I + ClCH_2O_2$ | $J = 6.910 \cdot 10^{-04} \cos(\chi)^{1.057} \exp(-0.238/\cos(\chi))$ | Bräuer et al. (2013) |
| H199 | $BrCH_2I \rightarrow I + BrCH_2O_2$ | $J = 4.261 \cdot 10^{-04} \cos(\chi)^{0.976} \exp(-0.250/\cos(\chi))$ | Bräuer et al. (2013) |

(a) $k^{2nd}$ in $cm^3$ molecules$^{-1}$ s$^{-1}$; $k^{1st}$ in s$^{-1}$; J in s$^{-1}$

**Table S7    Parameters for pressure dependent reactions.**

| | Reaction | TYPE | $k_0$[a] | $k_\infty$[a] | $F_C$ |
|---|---|---|---|---|---|
| H5 | $Cl + NO_2 \rightarrow ClNO_2$ | TROE | $1.80 \cdot 10^{-31} *(T/298)^{-2.0}$ | $1.00 \cdot 10^{-10} *(T/298)^{-1.0}$ | 0.6 |
| H6 | $ClO + NO_2 \rightarrow ClNO_3$ | TROE | $1.60 \cdot 10^{-31} *(T/298)^{-3.4}$ | $7.00 \cdot 10^{-11}$ | 0.4 |
| H44 | $Cl + C_2H_2 \rightarrow 0.26\ ClCHO + 0.21\ Cl + 0.53\ HCl + 0.21\ GLYOXAL + 1.32\ CO + 0.79\ HO_2$ | TROE | $6.10 \cdot 10^{-30} *(T/298)^{-3.0}$ | $2.00 \cdot 10^{-10}$ | 0.6 |
| H45 | $Cl + C_2H_4 \rightarrow ClCH_2CH_2O_2$ | TROE | $1.85 \cdot 10^{-29} *(T/298)^{-3.3}$ | $6.00 \cdot 10^{-10}$ | 0.4 |
| H54 | $ClCH_2CO_3 + NO_2 \rightarrow ClPAN$ | TROE | $2.70 \cdot 10^{-28} *(T/298)^{7.1}$ | $1.20 \cdot 10^{-11} *(T/298)^{0.9}$ | 0.3 |
| H60 | $ClPAN \rightarrow ClCH_2CO_3 + NO_2$ | TROE | $4.90 \cdot 10^{-03} \exp(-12100/T)$ | $5.40 \cdot 10^{+16} \exp(-13830/T)$ | 0.3 |
| H82 | $CH_3CH(Cl)CO_3 + NO_2 \rightarrow CH_3ClPAN$ | TROE | $2.70 \cdot 10^{-28} *(T/298)^{7.1}$ | $1.20 \cdot 10^{-11} *(T/298)^{0.9}$ | 0.3 |
| H83 | $CH_3ClPAN \rightarrow CH_3CH(Cl)CO_3 + NO_2$ | TROE | $4.90 \cdot 10^{-03} \exp(-12100/T)$ | $5.40 \cdot 10^{+16} \exp(-13830/T)$ | 0.3 |
| H102 | $Br + NO_2 \rightarrow BrNO_2$ | TROE | $4.20 \cdot 10^{-31} *(T/298)^{-2.4}$ | $2.70 \cdot 10^{-11}$ | 0.55 |
| H104 | $BrO + NO_2 \rightarrow BrNO_3$ | TROE | $4.70 \cdot 10^{-31} *(T/298)^{-3.1}$ | $1.80 \cdot 10^{-11}$ | 0.4 |
| H151 | $IO + NO_2 \rightarrow INO_3$ | TROE | $7.70 \cdot 10^{-31} (T/300)^{-5.0}$ | $1.60 \cdot 10^{-11}$ | 0.6 |

(a) $k^{2nd}$ in $cm^3$ molecules$^{-1}$ s$^{-1}$; $k^{1st}$ in s$^{-1}$

Rate constants calculated with TROE formula: $k(T) = \dfrac{k_0(T)[M]}{1 + \frac{k_0(T)[M]}{k_\infty(T)}} \times F_C^{\left\{1 + log_{10}\left(\frac{k_0(T)[M]}{k_\infty(T)}\right)^2\right\}^{-1}}$

**Table S8    Implemented phase transfers in the CAPRAM-HM3.0red**

② reactions that run in the cloud mode 'sub#1', ③ reactions that run in the aerosol mode 'sub#2', ● already included in CAPRAM3.0red

| | Species | $K_{H\,(298\,K)}$[(a)] | $-\Delta H/R$[(b)] | $\alpha$ | $D_{g\,(298\,K)}$[(c)] | Comment |
|---|---|---|---|---|---|---|
| H200③● | $Cl_2$ | $9.15\cdot10^{-2}$ | 2490 | 0.08 | 1.28 | Bräuer et al. (2013) |
| H201 | Cl | $2.00\cdot10^{-1}$ | | 0.05 | 1.82 | Bräuer et al. (2013) |
| H202②● | HCl | $1.10\cdot10^{0}$ | 2020 | 0.1026 | 1.89 | Bräuer et al. (2013) |
| H203③ | HOCl | $6.60\cdot10^{2}$ | 5862 | 0.5 | 1.51 | Bräuer et al. (2013) |
| H204③● | $ClNO_2$ | $2.40\cdot10^{-2}$ | | 0.01 | 1.27 | Bräuer et al. (2013) |
| H205③ | $ClNO_3$ | $2.10\cdot10^{5}$ | 8700 | 0.1 | 1.18 | Bräuer et al. (2013) |
| H206 | ClCHO | $3.00\cdot10^{3}$ | 7216 | 0.02 | 1.23 | Bräuer et al. (2013) |
| H207③● | $Br_2$ | $7.60\cdot10^{-1}$ | 4100 | 0.08 | 1.00 | Bräuer et al. (2013) |
| H208 | Br | $1.20\cdot10^{0}$ | | 0.05 | 1.29 | Bräuer et al. (2013) |
| H209③ | HBr | $1.30\cdot10^{0}$ | 10239 | 0.0481 | 1.26 | Bräuer et al. (2013) |
| H210③ | HOBr | $9.30\cdot10^{1}$ | 5862 | 0.5 | 1.16 | Bräuer et al. (2013) |
| H211③ | $BrNO_3$ | $2.10\cdot10^{5}$ | 8700 | 0.8 | 1.01 | Bräuer et al. (2013) |
| H212③ | BrCl | $9.40\cdot10^{-1}$ | -5600 | 0.33 | 1.05 | Bräuer et al. (2013) |
| H213 | $BrCH_2CO_3$ | $6.69\cdot10^{2}$ | 5893 | 0.019 | 0.84 | Bräuer et al. (2013) |
| H214② | $BrCH_2COOH$ | $1.52\cdot10^{5}$ | 9300 | 0.0322 | 0.84 | Bräuer et al. (2013); Sander (2015) |
| H215 | BrCHO | $7.40\cdot10^{1}$ | | 0.02 | 1.02 | Bräuer et al. (2013) |
| H216 | $I_2$ | $3.00\cdot10^{0}$ | 4431 | 0.0126 | 0.86 | Bräuer et al. (2013) |
| H217③ | HOI | $4.50\cdot10^{2}$ | 5862 | 0.5 | 1.08 | Bräuer et al. (2013) |
| H218 | $HIO_3$ | $2.10\cdot10^{5}$ | 8700 | 0.0126 | 0.98 | Bräuer et al. (2013) |
| H219③ | $INO_3$ | $2.10\cdot10^{5}$ | 8700 | 0.123 | 0.96 | Bräuer et al. (2013) |
| H220③ | $I_2O_2$ | $1.00\cdot10^{4}$ | | 0.123 | 0.80 | Bräuer et al. (2013); Sander (2015) |
| H221③ | ICl | $1.10\cdot10^{2}$ | 5600 | 0.0126 | 0.98 | Bräuer et al. (2013) |
| H222③ | IBr | $2.40\cdot10^{1}$ | 5600 | 0.0126 | 0.88 | Bräuer et al. (2013) |

(a) in M atm[-1]; (b) in K; (c) in m² s[-1]

**Table S9    Implemented aqueous-phase reactions in the CAPRAM-HM3.0red**

② reactions that run in the cloud mode 'sub#1', ③ reactions that run in the aerosol mode 'sub#2', ● already included in CAPRAM3.0red

| | Reaction | $k_{298}$[(a)] | $E_A/R$[(b)] | Comment |
|---|---|---|---|---|
| H223● | $Cl_2^- + H_2O_2 \rightarrow 2\ Cl^- + H^+ + HO_2$ | $6.20 \cdot 10^5$ | 3340 | Jacobi et al. (1999) |
| H224②● | $Cl_2^- + H_2O \rightarrow H^+ + Cl^- + ClOH^-$ | $2.34 \cdot 10^1$ | | Buxton et al. (1998) |
| H225② | $HOCl + HO_2 \rightarrow Cl + H_2O + O_2$ | $7.50 \cdot 10^6$ | | Bräuer et al. (2013) |
| H226 | $HOCl + OH \rightarrow ClO + H_2O$ | $2.00 \cdot 10^9$ | | Bräuer et al. (2013) |
| H227● | $Cl_2^- + HSO_3^- \rightarrow 2\ Cl^- + H^+ + SO_3^-$ | $1.70 \cdot 10^8$ | 400 | Jacobi (1996) |
| H228③ | $HOCl + HSO_3^- \rightarrow Cl^- + H^+ + HSO_4^{2-}$ | $7.60 \cdot 10^8$ | | Herrmann (2003) |
| H229 | $Cl^- + HSO_5^- \rightarrow HOCl + SO_4^{2-}$ | $1.80 \cdot 10^{-3}$ | 7352 | Fortnum et al. (1960) |
| H230● | $Cl_2^- + Fe^{2+} \rightarrow 2\ Cl^- + Fe^{3+}$ | $1.00 \cdot 10^7$ | 3030 | Thornton and Laurence (1973) |
| H231②● | $Cl^- + FeO_2^+ \rightarrow Fe^{3+} + ClOH^- + OH^- - H_2O$ | $1.00 \cdot 10^2$ | | Jacobsen et al. (1998) |
| H232● | $Cl_2^- + Mn^{2+} \rightarrow MnCl_2^+$ | $2.00 \cdot 10^7$ | 4090 | Laurence and Thornton (1973) |
| H233● | $MnCl_2^+ \rightarrow 0.588\ Cl_2^- + 0.588\ Mn^{2+} + 0.824\ Cl^- + 0.412\ Mn^{3+}$ | $5.10 \cdot 10^5$ | | Deguillaume et al. (2010); Laurence and Thornton (1973) |
| H234 | $2\ ClO \rightarrow Cl^- + ClO_3^- + 2\ H^+$ | $2.50 \cdot 10^9$ | | Klaning and Wolff (1985) |
| H235 | $OH + ClO_3^- \rightarrow ClO + O_2 + OH^-$ | $1.00 \cdot 10^6$ | | Buxton and Subhani (1972) |
| H236 | $Cl_2 + H_2O_2 \rightarrow 2\ H^+ + 2\ Cl^- + O_2$ | $1.83 \cdot 10^2$ | 5387 | Connick (1947) |
| H237③ | $ClNO_3 \rightarrow HOCl + HNO_3$ | $1.62 \cdot 10^6$ | 2800 | Shi et al. (2001) |
| H238② | $Cl_2^- + HC_2O_4^- \rightarrow 2\ Cl^- + H^+ + C_2O_4^-$ | $1.30 \cdot 10^6$ | | Bräuer et al. (2013) |
| H239② | $Cl_2^- + C_2O_4^{2-} \rightarrow 2\ Cl^- + C_2O_4^-$ | $4.00 \cdot 10^6$ | | Bräuer et al. (2013) |
| H240② | $ClCHO \rightarrow CO + H^+ + Cl^-$ | $1.00 \cdot 10^4$ | | Prager et al. (2001) |
| H241 | $Br + H_2O_2 \rightarrow H^+ + Br^- + HO_2$ | $4.00 \cdot 10^9$ | | Sutton et al. (1965) |
| H242② | $Br_2^- + HO_2 \rightarrow Br^- + 0.5\ Br_2 + 0.5\ H_2O_2 + 0.5\ O_2$ | $8.80 \cdot 10^9$ | | Sutton and Downes (1972) |
| H243 | $BrO + BrO \rightarrow BrO_2^- + HOBr + H^+$ | $2.80 \cdot 10^9$ | | Klaning and Wolff (1985) |
| H244 | $HOBr + OH \rightarrow BrO + H_2O$ | $2.00 \cdot 10^9$ | | Klaning and Wolff (1985) |
| H245② | $HOBr + HO_2 \rightarrow Br + H_2O + O_2$ | $1.00 \cdot 10^9$ | | Bräuer et al. (2013) |
| H246② | $HOBr + H_2O_2 \rightarrow H^+ + Br^- + H_2O + O_2$ | $3.50 \cdot 10^6$ | | Young (1950) |
| H247③ | $HOBr + HSO_3^- \rightarrow H+ + Br^- + HSO_4^-$ | $5.00 \cdot 10^9$ | | Bräuer et al. (2013) |
| H248 | $Br^- + HSO_5^- \rightarrow HOBr + SO_4^{2-}$ | $1.00 \cdot 10^0$ | 5338 | Fortnum et al. (1960) |

| | Reaction | $k_{298}^{(a)}$ | $E_A/R^{(b)}$ | Comment |
|---|---|---|---|---|
| H249 | $Br^- + NO_3 \rightarrow Br + NO_3^-$ | $3.80 \cdot 10^9$ | | Zellner et al. (1996) |
| H250 | $Br_2^- + Fe^{2+} \rightarrow 2\ Br^- + Fe^{3+}$ | $3.60 \cdot 10^6$ | 3330 | Thornton and Laurence (1973) |
| H251● | $Br_2^- + Mn^{2+} \rightarrow MnBr_2^+$ | $6.30 \cdot 10^6$ | 4330 | Thornton and Laurence (1973) |
| H252● | $MnBr_2^+ \rightarrow 0.577\ Br_2^- + 0.577\ Mn^{2+} + 0.846\ Br^- + 0.423\ Mn^{3+}$ | $5.20 \cdot 10^5$ | | Thornton and Laurence (1973); Deguillaume et al. (2010) |
| H253 | $BrO_3^- + SO_4^- \rightarrow BrO + O_2 + SO_4^{2-}$ | $1.40 \cdot 10^6$ | | Zuo and Katsumura (1998) |
| H254 | $Br + O_3 \rightarrow BrO + O_2$ | $1.50 \cdot 10^8$ | | Von Gunten and Oliveras (1998) |
| H255 | $BrO_3^- + HSO_3^- \rightarrow BrO_2^- + SO_4^{2-} + H^+$ | $2.70 \cdot 10^{-2}$ | | Szirovicza and Boga (1998) |
| H256 | $BrO_3^- + OH \rightarrow BrO + O_2 + OH^-$ | $5.00 \cdot 10^6$ | | Amichai et al. (1969) |
| H257③ | $BrNO_3 \rightarrow HOBr + HNO_3$ | $1.00 \cdot 10^9$ | | Hanson et al. (1996) |
| H258 | $BrO_3^- + HC_2O_4^- \rightarrow BrO_2^- + 2\ CO_2 + H_2O$ | $7.47 \cdot 10^{-4}$ | | Pelle et al. (2004) |
| H259② | $BrCHO \rightarrow CO + H^+ + Br^-$ | $1.00 \cdot 10^4$ | | Bräuer et al. (2013) |
| H260② | $CH_2BrCO_3 + H_2O \rightarrow CH_2BrCOOH + HO_2$ | $3.55 \cdot 10^5$ | | Bräuer et al. (2013) |
| H261 | $Br_2^- + HCOO^- \rightarrow 2\ Br^- + COOH$ | $4.90 \cdot 10^3$ | | Jacobi et al. (1996) |
| H262③ | $Br^- + HOCl \rightarrow BrCl + H_2O - H^+$ | $1.30 \cdot 10^6$ | | Kumar and Margerum (1987) |
| H263② | $BrO_2^- + HOCl \rightarrow 0.85\ ClO_3^- + 0.93\ HOBr + 0.08\ ClO_2^- + 0.07\ BrO_3^- + 0.92\ Cl^- + 0.92\ H^+ - 0.85\ HOCl$ | $1.60 \cdot 10^2$ | | Nicoson et al. (2003) |
| H264 | $I^- + O_3 \rightarrow HOI + O_2$ | $2.17 \cdot 10^9$ | 8790 | Magi et al. (1997) |
| H265② | $IO + IO \rightarrow HOI + HIO_3 + H^+ - H_2O - H_2O_2$ | $1.50 \cdot 10^9$ | | Buxton et al. (1986) |
| H266③ | $HOI + HSO_3^- \rightarrow H^+ + I^- + HSO_4^-$ | $5.00 \cdot 10^9$ | | Pechtl and von Glasow (2007) |
| H267 | $HOI + OH \rightarrow IO + H_2O$ | $7.00 \cdot 10^9$ | | Buxton and Mulazzani (2007) |
| H268③ | $INO_3 \rightarrow HOI + HNO_3$ | $1.62 \cdot 10^6$ | 2800 | Hoffmann et al. (2019b) |
| H269 | $I_2O_2 + H^+ \rightarrow HIO_3 + HOI + H^+$ | $3.20 \cdot 10^4$ | | Valkai and Horvath (2016) |
| H270 | $IO_3^- + OH \rightarrow IO + O_2 + OH^-$ | $1.08 \cdot 10^5$ | | Mezyk (1996) |

(a) $k_{298}^{2nd}$ in $l^1\ mol^{-1}\ s^{-1}$; $k_{298}^{1st}$ in $s^{-1}$; (b) in K

**Table S10  Implemented aqueous-phase equilibrium reactions in the CAPRAM-HM3.0red**

② reactions that run in the cloud mode 'sub#1', ③ reactions that run in the aerosol mode 'sub#2', • already included in CAPRAM3.0red

| | Reaction | $K^{(a)}$ | $k_{f, 298}^{(b)}$ | $E_A/R^{(c)}$ | $k_{b, 298}^{(b)}$ | $E_A/R^{(c)}$ | References |
|---|---|---|---|---|---|---|---|
| H271②• | $Cl + Cl^- \rightleftharpoons Cl_2^-$ | $1.4 \cdot 10^5$ | $8.50 \cdot 10^9$ | | $6.00 \cdot 10^4$ | | Buxton et al. (1998) |
| H272③• | $Cl_2 + H_2O \rightleftharpoons H^+ + Cl^- + HOCl$ | $1.90 \cdot 10^{-5} e^{-4500/T}$ | $4.00 \cdot 10^{-1}$ | 8000 | $2.10 \cdot 10^4$ | 3500 | Wang and Margerum (1994) |
| H273③• | $HCl \rightleftharpoons H^+ + Cl^-$ | $1.72 \cdot 10^6 \, e^{6890/T}$ | $5.00 \cdot 10^{11}$ | -6890 | $2.90 \cdot 10^5$ | | Marsh and McElroy (1985); Graedel and Weschler (1981) |
| H274②• | $Cl^- + OH \rightleftharpoons ClOH^-$ | $7.00 \cdot 10^{-1}$ | $4.30 \cdot 10^9$ | | $6.10 \cdot 10^9$ | | Jayson et al. (1973) |
| H275② | $Cl + OH^- \rightleftharpoons ClOH^-$ | $7.83 \cdot 10^8$ | $1.80 \cdot 10^{10}$ | | $2.30 \cdot 10^1$ | | Klaning and Wolff (1985) |
| H276②• | $ClOH^- + H^+ \rightleftharpoons Cl + H_2O$ | $5.10 \cdot 10^6$ | $2.1 \cdot 10^{10}$ | | $4.10 \cdot 10^3$ | | Jayson et al. (1973) |
| H277②• | $ClOH^- + Cl^- \rightleftharpoons Cl_2^- + OH^-$ | $2.20 \cdot 10^{-4}$ | $1.00 \cdot 10^4$ | | $4.50 \cdot 10^7$ | | Grigor'ev et al. (1987) |
| H278②• | $Cl^- + SO_4^- \rightleftharpoons Cl + SO_4^{2-}$ | $1.20 \cdot 10^0$ | $2.52 \cdot 10^8$ | | $2.10 \cdot 10^8$ | | Buxton et al. (1999b) |
| H279②• | $Cl^- + NO_3 \rightleftharpoons Cl + NO_3^-$ | $3.40 \cdot 10^0 \, e^{-4300/T}$ | $3.40 \cdot 10^8$ | 4300 | $1.00 \cdot 10^8$ | | Buxton et al. (1999a) |
| H280 | $HOCl + NO_2^- \rightleftharpoons ClNO_2 + OH^-$ | $3.97 \cdot 10^{-4}$ | $1.99 \cdot 10^7$ | | $5.00 \cdot 10^{10}$ | | Lahoutifard et al. (2002) |
| H281③ | $Cl_2 + SO_4^{2-} \rightleftharpoons Cl^- + HOCl + HSO_4^-$ | $1.14 \cdot 10^{-3}$ | $3.20 \cdot 10^1$ | | $2.80 \cdot 10^3$ | | Wang and Margerum (1994) |
| H282③• | $Cl^- + NO_2^+ \rightleftharpoons ClNO_2$ | $1.44 \cdot 10^8$ | $3.90 \cdot 10^{10}$ | | $2.70 \cdot 10^2$ | | Behnke et al. (1997) |
| H283②• | $Br + Br^- \rightleftharpoons Br_2^-$ | $6.32 \cdot 10^5$ | $1.20 \cdot 10^{10}$ | | $1.90 \cdot 10^4$ | | Merenyi and Lind (1994) |
| H284③ | $Br_2 + H_2O \rightleftharpoons H^+ + Br^- + HOBr$ | $1.06 \cdot 10^{-10} e^{-7500/T}$ | $1.70 \cdot 10^0$ | 7500 | $1.60 \cdot 10^{10}$ | | Beckwith et al. (1996) |
| H285③ | $HBr \rightleftharpoons H^+ + Br^-$ | $1.00 \cdot 10^9$ | $5.00 \cdot 10^{11}$ | | $5.00 \cdot 10^2$ | | Lax (1969) |
| H286②• | $Br^- + OH \rightleftharpoons BrOH^-$ | $3.33 \cdot 10^2$ | $1.10 \cdot 10^{10}$ | | $3.30 \cdot 10^7$ | | Zehavi and Rabani (1972) |
| H287② | $Br + OH^- \rightleftharpoons BrOH^-$ | $3.10 \cdot 10^3$ | $1.30 \cdot 10^{10}$ | | $4.20 \cdot 10^6$ | | Zehavi and Rabani (1972); Klaning and Wolff (1985) |
| H288②• | $BrOH^- + H^+ \rightleftharpoons Br + H_2O$ | $1.80 \cdot 10^{12}$ | $4.40 \cdot 10^{10}$ | | $2.45 \cdot 10^{-2}$ | | Zehavi and Rabani (1972); Klaning and Wolff (1985) |
| H289②• | $BrOH^- + Br^- \rightleftharpoons Br_2^- + OH^-$ | $7.00 \cdot 10^1$ | $1.90 \cdot 10^8$ | | $2.70 \cdot 10^6$ | | Zehavi and Rabani (1972); de Violet (1981) |
| H290 | $HOBr + HOBr \rightleftharpoons H^+ + Br^- + BrO_2^-$ | $6.70 \cdot 10^{-12}$ | $2.00 \cdot 10^{-5}$ | | $3.00 \cdot 10^6$ | | Field and Foersterling (1986) |
| H291 | $HOBr + BrO_2^- \rightleftharpoons H^+ + Br^- + BrO_3^-$ | $1.70 \cdot 10^0$ | $3.20 \cdot 10^0$ | | $2.00 \cdot 10^0$ | | Field and Foersterling (1986) |
| H292② | $CH_2BrCOOH \rightleftharpoons CH_2BrCOO^- + H^+$ | $1.75 \cdot 10^{-5} \, e^{46/T}$ | $8.75 \cdot 10^5$ | -46 | $5.00 \cdot 10^{10}$ | | Bräuer et al. (2013) |
| H293③ | $Br_2 + SO_4^{2-} + H_2O \rightleftharpoons HOBr + Br^- + HSO_4^-$ | $6.15 \cdot 10^{-6}$ | $2.28 \cdot 10^4$ | | $3.70 \cdot 10^9$ | | Beckwith et al. (1996) |
| H294③ | $BrCl \rightleftharpoons HOBr + H^+ + Cl^- - H_2O$ | $1.80 \cdot 10^{-5}$ | $1.00 \cdot 10^5$ | | $5.60 \cdot 10^9$ | | Wang et al. (1994) |
| H295③ | $BrCl^- \rightleftharpoons Br^- + Cl$ | $1.60 \cdot 10^{-7}$ | $1.90 \cdot 10^3$ | | $1.20 \cdot 10^{10}$ | | Donati (2002) |

② reactions that run in the cloud mode 'sub#1', ③ reactions that run in the aerosol mode 'sub#2', ● already included in CAPRAM3.0red

| | Reaction | $K^{(a)}$ | $k_{f, 298}^{(b)}$ | $E_A/R^{(c)}$ | $k_{b, 298}^{(b)}$ | $E_A/R^{(c)}$ | References |
|---|---|---|---|---|---|---|---|
| H296③ | $BrCl^- \rightleftharpoons Br + Cl^-$ | $6.10 \cdot 10^{-4}$ | $6.10 \cdot 10^4$ | | $1.00 \cdot 10^8$ | | Donati (2002) |
| H297③ | $BrCl^- + Br^- \rightleftharpoons Br_2^- + Cl^-$ | $1.86 \cdot 10^3$ | $8.00 \cdot 10^9$ | | $4.30 \cdot 10^6$ | | Ershov (2004) |
| H298③ | $BrCl^- + Cl^- \rightleftharpoons Br^- + Cl_2^-$ | $2.75 \cdot 10^{-8}$ | $1.10 \cdot 10^2$ | | $4.00 \cdot 10^9$ | | Ershov (2004) |
| H299③ | $Br_2Cl^- \rightleftharpoons BrCl + Br^-$ | $5.60 \cdot 10^{-5}$ | $4.30 \cdot 10^5$ | | $7.70 \cdot 10^9$ | | Wang et al. (1994) |
| H300③ | $Br_2Cl^- \rightleftharpoons Br_2 + Cl^-$ | $7.60 \cdot 10^{-1}$ | $3.80 \cdot 10^4$ | | $5.00 \cdot 10^4$ | | Wang et al. (1994); Matthew and Anastasio (2006) |
| H301③ | $BrCl_2^- \rightleftharpoons BrCl + Cl^-$ | $1.70 \cdot 10^{-1}$ | $1.70 \cdot 10^5$ | | $1.00 \cdot 10^6$ | | Ershov (2004) |
| H302③ | $BrCl_2^- \rightleftharpoons Br^- + Cl_2$ | $1.50 \cdot 10^{-6}$ | $9.00 \cdot 10^3$ | | $6.00 \cdot 10^9$ | | Ershov (2004) |
| H303 | $I_2 + OH^- \rightleftharpoons I_2OH^-$ | $5.00 \cdot 10^0$ | $1.00 \cdot 10^{10}$ | | $2.00 \cdot 10^9$ | | Buxton and Mulazzani (2007) |
| H304 | $I_2OH^- \rightleftharpoons HOI + I^-$ | $8.30 \cdot 10^0$ | $2.49 \cdot 10^9$ | | $3.00 \cdot 10^8$ | | Buxton and Mulazzani (2007) |
| H305 | $HOI + H^+ + I^- \rightleftharpoons I_2 + H_2O$ | $1.47 \cdot 10^{12}$ | $4.40 \cdot 10^{12}$ | | $3.00 \cdot 10^0$ | | Eigen and Kustin (1962) |
| H306② | $HIO_3 \rightleftharpoons H^+ + IO_3^-$ | $1.70 \cdot 10^{-1}$ | $8.50 \cdot 10^9$ | | $5.00 \cdot 10^{10}$ | | Lide et al. (1995) |
| H307③ | $HOI + H^+ + Cl^- \rightleftharpoons ICl$ | $1.20 \cdot 10^4$ | $2.90 \cdot 10^{10}$ | | $2.40 \cdot 10^6$ | | Wang et al. (1989) |
| H308③ | $HOI + H^+ + Br^- \rightleftharpoons IBr$ | $5.10 \cdot 10^6$ | $4.10 \cdot 10^{12}$ | | $8.00 \cdot 10^5$ | | De Barros Faria et al. (1993) |
| H309③ | $ICl + Br^- \rightleftharpoons IBr + Cl^-$ | $3.30 \cdot 10^3$ | $1.65 \cdot 10^{14}$ | | $5.00 \cdot 10^{10}$ | | Wagman et al. (1982) |

(a) in $M^{m-n}$, n order of reaction of forward reaction, m order of reaction of backward reaction; (b) $k_{298}^{2nd}$ in $l^1$ $mol^{-1}$ $s^{-1}$, $k_{298}^{1st}$ in $s^{-1}$; (c) in K

**Table S11    Measured values of HCl and BrO in marine environments.**

| HCl | BrO* | Location | Comment | Reference |
|---|---|---|---|---|
| daily average: 133 – 675 ppt | | Bermuda | | Keene and Savoie (1999) |
| range: 30-250 ppt | | Hawaii | | Pszenny et al. (2004) |
| median: 351 ppt | | Appledore Island | | Keene et al. (2007) |
| daily median: 82-682 ppt | | North to South Atlantic | | Keene et al. (2009) |
| median: 206 ppt | | Cape Verde | range: 26 – 613 ppt | Sander et al. (2013) |
| | max. 1-3.6 ppt | Canary Island | in remote ocean below detection limit | Leser et al. (2003) |
| | average 2.3 ppt | Mace Head | Coastal region | Saiz-Lopez et al. (2004) |
| | average max. 2.5 ± 1.1 ppt | Cape Verde | | Read et al. (2008) |
| | < 0.5 ppt | Eastern tropical Pacific | MBL: below detection limit | Volkamer et al. (2015) |
| | 0.03 ± 0.26 ppt | Western tropical Pacific | clean MBL outflow | Chen et al. (2016) |
| | 0.17-1.64 ppt | Western Pacific | between 0.5 – 7 km height | Le Breton et al. (2017) |

DL – Detection Limit; * for a more detailed overview on measurements before 2003 see Sander et al. (2003)